# Optimal Activation Functions for the Random Features Regression Model

**Jianxin Wang** [*]
Department of Electrical and Computer Engineering
Rice University
jw162@rice.edu

**José Bento**
Department of Computer Science
Boston College
bentoayr@bc.edu

## Abstract

The asymptotic mean squared test error and sensitivity of the Random Features Regression model (RFR) have been recently studied. We build on this work and identify in closed-form the family of Activation Functions (AFs) that minimize a combination of the test error and sensitivity of the RFR under different notions of functional parsimony. We find scenarios under which the optimal AFs are linear, saturated linear functions, or expressible in terms of Hermite polynomials. Finally, we show how using optimal AFs impacts well established properties of the RFR model, such as its double descent curve, and the dependency of its optimal regularization parameter on the observation noise level.

## 1 Introduction

For many neural network (NN) architectures, the test error does not monotonically increase as a model's complexity increases but can go down with the training error both at low and high complexity levels. This phenomenon, the *double descent curve*, defies intuition and has motivated new frameworks to explain it. Explanations have been advanced involving linear regression with random covariates (Belkin et al., 2020; Hastie et al., 2022), kernel regression (Belkin et al., 2019b; Liang & Rakhlin, 2020), the *neural tangent kernel* model (Jacot et al., 2018), and the *Random Features Regression* (RFR) model (Mei & Montanari, 2022). These frameworks allow queries beyond the generalization power of NNs. For example, they have been used to study networks' robustness properties (Hassani & Javanmard, 2022; Tripuraneni et al., 2021).

One aspect within reach and unstudied to this day is finding optimal Activation Functions (AFs) for these models. It is known that AFs affect a network's approximation accuracy and efforts to optimize AFs have been undertaken. Previous work has justified the choice of AFs empirically, e.g., Ramachandran et al. (2017), or provided numerical procedures to learn AF parameters, sometimes jointly with models' parameters, e.g. Unser (2019). See Rasamoelina et al. (2020) for commonly used AFs and Appendix C for how AFs have been previously derived.

We derive for the first time closed-form optimal AFs such that an explicit objective function involving the asymptotic test error and sensitivity of a model is minimized. Setting aside empirical and principled but numerical methods, all past principled and analytical approaches to design AFs focus on non accuracy related considerations, e.g. Milletarí et al. (2019). We focus on AFs for the RFR model and expand its understanding. We preview a few surprising conclusions extracted from our main results:

1. The optimal AF can be linear, in which case the RFR model is a linear model. For example, if no regularization is used for training, and for low complexity models, a linear AF is often preferred if we want to minimize test error. For high complexity models a non-linear AF is often better;

2. A linear optimal AF can destroy the double descent curve behaviour and achieve small test error with much fewer samples than e.g. a ReLU;

3. When, apart from the test error, the sensitivity of a model becomes important, optimal AFs that without sensitivity considerations were linear can become non-linear, and vice-versa;

4. Using an optimal AF with an arbitrary regularization during training can lead to the same, or better, test error as using a non-optimal AF, e.g. ReLU, and optimal regularization.

---

[*]Work done during undergrad at Boston College.

## 1.1 Problem set up

We consider the effect of AFs on finding an approximation $f$ to a square-integrable function $f_d$ on the $d$-dimensional sphere $\mathbb{S}^{d-1}(\sqrt{d})$, the function $f_d$ having been randomly generated. The approximation $f$ is to be learnt from training data $\mathcal{D} = \{x_i, y_i\}_{i=1}^n$ where $x_i \in \mathbb{S}^{d-1}(\sqrt{d})$, the variables $\{x_i\}_{i=1}^n$ are i.i.d. uniformly sampled from $\mathbb{S}^{d-1}(\sqrt{d})$, and $y_i = f_d(x_i) + \epsilon_i$, where the noise variables $\{\epsilon_i\}_{i=1}^n$ are i.i.d. with $\mathbb{E}(\epsilon_i) = 0, \mathbb{E}(\epsilon_i^2) = \tau^2$, and $\mathbb{E}(\epsilon_i^4) < \infty$.

The approximation $f$ is defined according to the RFR model. The RFR model can be viewed as a two-layer NN with random first-layer weights encoded by a matrix $\Theta \in \mathbb{R}^{N \times d}$ with $i$th row $\theta_i \in \mathbb{R}^d$ satisfying $\|\theta_i\| = \sqrt{d}$, with $\{\theta_i\}$ i.i.d. uniform on $\mathbb{S}^{d-1}(\sqrt{d})$, and with to-be-learnt second-layer weights encoded by a vector $a = [a_i]_{i=1}^N = \mathbb{R}^N$. Unless specified otherwise, the norm $\|\cdot\|$ denotes the Euclidean norm. The RFR model defines $f_{a,\Theta} : \mathbb{S}^{d-1}(\sqrt{d}) \mapsto \mathbb{R}$ such that

$$f_{a,\Theta}(x) = \sum_{i=1}^N a_i \sigma(\langle \theta_i, x \rangle / \sqrt{d}). \tag{1}$$

where $\sigma(\cdot)$ is the AF that is the target of our study and $\langle x, y \rangle$ denotes the inner product between vectors $x$ and $y$. When clear from the context, we write $f_{a,\Theta}$ as $f$, omitting the model's parameters. The optimal weights $a^\star$ are learnt using ridge regression with regularization parameter $\lambda \geq 0$, namely,

$$a^\star = a^\star(\lambda, \mathcal{D}) = \arg \min_{a \in \mathbb{R}^N} \left\{ \frac{1}{n} \sum_{j=1}^n \left( y_j - \sum_{i=1}^N a_i \sigma(\langle \theta_i, x_j \rangle / \sqrt{d}) \right)^2 + \frac{N\lambda}{d} \|a\|^2 \right\}. \tag{2}$$

**We will tackle this question**: *What is the simplest $\sigma$ that leads to the best approximation of $f_d$?*

We quantify the simplicity of an AF $\sigma$ with its norm in different functional spaces. Namely, either

$$\|\sigma\|_1 \triangleq \mathbb{E}(|\sigma'(Z)|), \qquad \text{or} \qquad (3) \qquad \|\sigma\|_2 \triangleq \sqrt{\mathbb{E}((\sigma'(Z))^2)}, \tag{4}$$

where $\sigma'$ is the derivative of $\sigma$ and the expectations are with respected to a normal random variable $Z$ with zero mean and unit variance, i.e. $Z \sim \mathcal{N}(0, 1)$. For a comment on these choices please read Appendix A. We quantify the quality with which $f = f_{a^\star, \Theta}$ approximates $f_d$ via $L$, a linear combination of the mean squared error and the sensitivity of $f$ to perturbations in its input. For $\alpha \in [0, 1]$, $x$ uniform on $\mathbb{S}^{d-1}(\sqrt{d})$, we define

$$L \triangleq (1 - \alpha)\mathcal{E} + \alpha\mathcal{S}, \quad (5) \quad \text{where } \mathcal{E} \triangleq \mathbb{E}((f(x) - f_d(x))^2), \quad (6) \quad \text{and } \mathcal{S} \triangleq \|\mathbb{E} \nabla_x f(x)\|^2. \tag{7}$$

See Appendix B for a comment on our choice for sensitivity.

Like in Mei & Montanari (2022); D'Amour et al. (2020), we operate in the *asymptotic proportional regime* where $n, d, N \to \infty$, and have constant ratios between them, namely, $N/d \to \psi_1$ and $n/d \to \psi_2$. In this asymptotic setting, it does not matter if in defining (6) and (7), in addition to taking the expectation with respect to the test data $x$, independently of $\mathcal{D}$, we also take expectations over $\mathcal{D}$ and the random features in RFR. This is because when $n, d, N \to \infty$ with the ratios defined above, $\mathcal{E}$ and $\mathcal{S}$ will concentrate around their means (Mei & Montanari, 2022; D'Amour et al., 2020).

Mathematically, denoting by $\|\sigma\|$ either (4) or (3), our goal is to study the solutions of the problem

$$\min_{\sigma^\star} \|\sigma^\star\| \text{ subject to } \sigma^\star \in \arg \min_\sigma L(\sigma). \tag{8}$$

Notice that the outer optimization only affects the selection of optimal AF in so far as the inner optimization does not uniquely define $\sigma^\star$, which, as we will later see, it does not.

To the best of our knowledge, no prior theoretical work exists on how optimal AFs affect performance guarantees. We review literature review on non-theoretical works on the design of AFs, and a work studying the RFR model for purposes other than the design of AFs in Appendix C.

## 2 Background on the asymptotic properties of the RFR model

Here we will review recently derived closed-form expressions for the asymptotic mean squared error and sensitivity of the RFR model, which are the starting point of our work. First, however, we explain the use-inspired reasons for our setup. Our assumptions are the same as, or very similar to, those of published theoretical papers, e.g. Jacot et al. (2018); Yang et al. (2021); Ghorbani et al. (2021); Mel & Pennington (2022).

1. **Data on a sphere**: Normalization of input data is a best practice when learning with NNs (Huang et al., 2020). Assuming that input data lives on a sphere is one type of normalization.
2. **Random features**: The seminal work of Rahimi & Recht (2007a) showed the success of using random features on real datasets. For a recent review on their use see Cao et al. (2018).
3. **Asymptotic setting**: Mei & Montanari (2022) empirically showed that the convergence to the asymptotic regime is relatively fast, even with just a few hundreds of dimensions. Most real world applications involve larger dimensions $d$, lots of data $n$, and lots of neurons $N$.
4. **Shallow architecture**: For a finite input dimension $d$, the RFR model can learn arbitrary functions as the number of features $N$ grows large (Bach, 2017; Rahimi & Recht, 2007b; Ghorbani et al., 2021). Existing proof techniques make it very hard yet to extend our type of analysis to more than two layers or complex architectures. A few papers consider models with depth > 2 but do not tackle our problem and have other heavy restrictions on the model, e.g. Pennington et al. (2018).
5. **Regularization**: Using regularization during training to control the weights' magnitude is common. It can help convergence speed and generalization error (Goodfellow et al., 2016). For a review on different types of regularization for learning with NNs see Kukačka et al. (2017).

We make the following assumptions, which we assume hold in the theorems in this section.

**Assumption 1.** *We assume that the AF $\sigma$ is weakly differentiable with weak derivative $\sigma'$, it satisfies $|\sigma(u)|, |\sigma'(u)| \leq c_0 e^{c_1|u|} \forall u \in \mathbb{R}$ for some constants $0 < c_0, c_1 < \infty$, and that it also satisfies*

$$\mu_0 = \mathbb{E}\{\sigma(Z)\}, \quad \mu_1 = \mathbb{E}\{Z\sigma(Z)\}, \quad \mu_2 = \mathbb{E}\{\sigma(Z)^2\}, \quad \mu_\star^2 = \mu_2 - \mu_0^2 - \mu_1^2, \quad \zeta = \mu_1/\mu_\star, \quad (9)$$

*for some $\mu_0, \mu_1, \mu_2 \in \mathbb{R}$, where the expectations are with respect to $Z \sim \mathcal{N}(0, 1)$.*

**Assumption 2.** *We assume that $N = N(d)$ and $n = n(d)$ such that the following limits exist in $(0, \infty)$: $\lim_{d\to\infty} N(d)/d = \psi_1$ and $\lim_{d\to\infty} n(d)/d = \psi_2$.*

**Assumption 3.** *We assume that $y_i = f_d(x_i) + \epsilon_i$, where $\{\epsilon_i\}_{i\leq n} \sim_{i.i.d.} \mathbb{P}_\epsilon$ are independent of $\{x_i\}_{i\leq n}$, with $\mathbb{E}(\epsilon_1) = 0$, $\mathbb{E}(\epsilon_1^2) = \tau^2$, $\mathbb{E}(\epsilon_1^4) < \infty$, expectations with respect to $\{\epsilon_i\}$. Furthermore,*

$$f_d(x) = \beta_{d,0} + \langle \beta_{d,1}, x \rangle + f_d^{NL}(x), \quad (10)$$

*where $\beta_{d,0} \in \mathbb{R}$, $\beta_{d,1} \in \mathbb{R}^d$ are deterministic with $\lim_{d\to\infty} \beta_{d,0}^2 = F_0^2$, $\lim_{d\to\infty} \|\beta_{d,1}\|_2^2 = F_1^2 > 0$. The non-linear $f_d^{NL}$ is a centered Gaussian process indexed by $x \in \mathbb{S}^{d-1}(\sqrt{d})$, with covariance*

$$\mathbb{E}_{f_d^{NL}}\{f_d^{NL}(x_1) f_d^{NL}(x_2)\} = \Sigma_d(\langle x_1, x_2 \rangle/d), \quad (11)$$

*where $\Sigma_d(\cdot)$ satisfies $\mathbb{E}_{x\sim\text{Unif}(\mathbb{S}^{d-1}(\sqrt{d}))}\{\Sigma_d(x_1/\sqrt{d})\} = 0$, $\mathbb{E}_{x\sim\text{Unif}(\mathbb{S}^{d-1}(\sqrt{d}))}\{\Sigma_d(x_1/\sqrt{d})x_1\} = 0$, where $x_1$ is the 1st component of $x$. We define the Signal to Noise Ratio (SNR) $\rho$ by*

$$\rho = F_1^2/(F_\star^2 + \tau^2), \quad \text{where } F_\star^2 \triangleq \lim_{d\to\infty} \Sigma_d(1). \quad (12)$$

Informally, $\mu_\star$ quantifies how non-linear the AF is (cf. Lemma 3.1), $\psi_1$ quantifies the complexity of the RFR model relative of the dimension $d$, $\psi_2$ quantifies the amount of data used for training relative to $d$, $\tau^2$ is the variance of the observation noise, $F_1$ is the magnitude of the linear component of our target function $f_d$, which is controlled by $\beta_{d,1}$, $F_\star$ is the magnitude of the non-linear component $f_d^{NL}$ in the target function, and $\rho$ is the ratio between the magnitude of the linear component and the magnitude of all of the sources of randomness in the noisy function $f_d + \epsilon$. Recall that all of our results will be derived in the asymptotic regime when $d \to \infty$.

Our contributions are divided into two parts, Section 3.1 and Section 3.2. The theorems' statements in Section 3.2 quickly get prohibitively complex as they are stated more generally, with lots of special cases having to be discussed. Hence, in Section 3.2 we display our analysis on the following three different important regimes: $R_1$: *Ridgeless limit* regime, when $\lambda \to 0^+$; $R_2$: *Highly overparameterized limit*, when $\psi_1 \to \infty$; $R_3$: *Large sample limit*, when $\psi_2 \to \infty$. Section 3.1's results are general and not restricted to these regimes. In the context of the RFR model, these regimes were introduced and discussed in Mei & Montanari (2022). For what follows we define $\bar{\lambda} \triangleq \lambda/\mu_\star^2$. Any "$\lim_{d\to\infty} X = Y$" should be interpreted as $X$ converging to $Y$ in probability with respect to the training data $\mathcal{D}$, the random features $\Theta$, and the random target $f_d$ as $d \to \infty$.

## 2.1 ASYMPTOTIC MEAN SQUARED TEST ERROR OF THE RFR MODEL

The following theorems are a specialization of a more general theorem, Theorem 12 Mei & Montanari (2022), which we include in the Appendix G for completeness.

**Theorem 1** (Theorem 3 Mei & Montanari (2022)). *The asymptotic test error* (6) *for regime $R_1$ equals*

$$\mathcal{E}_{R_1}^{\infty} \equiv \lim_{\lambda \to 0^+} \lim_{d \to \infty} \mathcal{E} = F_1^2 \mathscr{B}_{\text{rless}}(\zeta, \psi_1, \psi_2) + (\tau^2 + F_\star^2) \mathscr{V}_{\text{rless}}(\zeta, \psi_1, \psi_2) + F_\star^2, \tag{13}$$

*where* $\mathscr{B}_{\text{rless}}(\zeta, \psi_1, \psi_2) \equiv \mathscr{E}_{1,\text{rless}}/\mathscr{E}_{0,\text{rless}}$, $\mathscr{V}_{\text{rless}}(\zeta, \psi_1, \psi_2) \equiv \mathscr{E}_{2,\text{rless}}/\mathscr{E}_{0,\text{rless}}$, *and the functions* $\mathscr{E}_{0,\text{rless}}, \mathscr{E}_{1,\text{rless}}$ *and* $\mathscr{E}_{2,\text{rless}}$ *are polynomials that are functions of* $\zeta^2, \psi_1, \psi_2$ *and* $\chi$, *where* $\chi$ *is a function of* $\psi \equiv \min\{\psi_1, \psi_2\}$ *and* $\zeta^2$. *See Appendix D for details.*

**Remark 1.** As a function of $\psi_1$, $\mathcal{E}_{R_1}^{\infty}$ has a discontinuity at $\psi_1 = \psi_2$ called the *interpolation threshold*. For $\psi_2$ high enough, and for $\psi_1 < \psi_2$, $\mathcal{E}_{R_1}^{\infty}$ decreases, reaches a minimum and then explodes approaching $\psi_2$. However, past $\psi_2$, $\mathcal{E}_{R_1}^{\infty}$ decreases again with $\psi_1$. This *double descent* behavior has been observed/studied in many settings, including Mei & Montanari (2022) and references therein.

**Theorem 2** (Theorem 4 Mei & Montanari (2022)). *The asymptotic test error* (6) *for regime $R_2$ equals*

$$\mathcal{E}_{R_2}^{\infty} \equiv \lim_{\psi_1 \to \infty} \lim_{d \to \infty} \mathcal{E} = F_1^2 \mathscr{B}_{\text{wide}}(\zeta, \psi_2, \overline{\lambda}) + (\tau^2 + F_\star^2) \mathscr{V}_{\text{wide}}(\zeta, \psi_2, \overline{\lambda}) + F_\star^2, \tag{14}$$

*where* $\mathscr{B}_{\text{wide}}$ *and* $\mathscr{V}_{\text{wide}}$ *are defined in Appendix E*

**Theorem 3** (Theorem 5 Mei & Montanari (2022)). *The asymptotic test error* (6) *for regime $R_3$ equals*

$$\mathcal{E}_{R_3}^{\infty} \equiv \lim_{\psi_2 \to \infty} \lim_{d \to \infty} \mathcal{E} = F_1^2 \mathscr{B}_{\text{lsamp}}(\zeta, \psi_1, \lambda/\mu_\star^2) + F_\star^2 \tag{15}$$

*where* $\mathscr{B}_{\text{lsamp}}(\zeta, \psi_1, \lambda/\mu_\star^2)$ *is defined in Appendix F*

## 2.2 ASYMPTOTIC SENSITIVITY OF THE RFR MODEL

We derive a sensitivity formula for regimes $R_1, R_2, R_3$. Our theorems are a specialization (proofs in Appendix M) of the more general Theorem 13 that we include in the Appendix G for completeness.

**Theorem 4.** *The sensitivity* (7) *for regime $R_1$ equals*

$$\mathcal{S}_{R_1}^{\infty} \equiv \lim_{\lambda \to 0^+} \lim_{d \to \infty} \mathcal{S} = \zeta^2 \left( \frac{F_1^2 \mathscr{D}_{1,\text{rless}}(\zeta, \psi_1, \psi_2)}{(\chi \zeta^2 - 1) \mathscr{D}_{0,\text{rless}}(\zeta, \psi_1, \psi_2)} + \frac{(F_\star^2 + \tau^2) \mathscr{D}_{2,\text{rless}}(\zeta, \psi_1, \psi_2)}{\mathscr{D}_{0,\text{rless}}(\zeta, \psi_1, \psi_2)} \right), \tag{16}$$

*where* $\mathscr{D}_{0,\text{rless}}(\zeta, \psi_1, \psi_2)$, $\mathscr{D}_{1,\text{rless}}(\zeta, \psi_1, \psi_2)$, *and* $\mathscr{D}_{2,\text{rless}}(\zeta, \psi_1, \psi_2)$ *are polynomials found in App. H.*

**Theorem 5.** *Let $\omega_2$ equal* (32), *defined in Appendix E. The sensitivity* (7) *for regime $R_2$ equals*

$$\mathcal{S}_{R_2}^{\infty} \equiv \lim_{\psi_1 \to \infty} \lim_{d \to \infty} \mathcal{S} = \frac{\omega_2^2((F_\star^2 + \tau^2)(-1 + \omega_2) + F_1^2(-1 - \psi_2 + \omega_2(-1 + \psi_2)))}{(-1 + \omega_2)(\psi_2 - 2\omega_2\psi_2 + \omega_2^2(-1 + \psi_2))}. \tag{17}$$

**Theorem 6.** *Let $\omega_1$ equal* (35), *defined in Appendix F.. The sensitivity* (7) *for regime $R_3$ equals*

$$\mathcal{S}_{R_3}^{\infty} \equiv \lim_{\psi_2 \to \infty} \lim_{d \to \infty} \mathcal{S} = F_1^2(1 + (2/(-1 + \omega_1)) + (\psi_1/(\psi_1 - 2\omega_1\psi_1 + \omega_1^2(-1 + \psi_1)))). \tag{18}$$

## 2.3 GAUSSIAN EQUIVALENT MODELS

A string of recent work shows that the asymptotic statistics of different models, e.g. their test MSE, is equivalent to that of a Gaussian model. This equivalence is known for the setup in (Mei & Montanari, 2022), and also for other setups Hu & Lu (2020); Ba et al. (2022); Loureiro et al. (2021); Montanari & Saeed (2022). Setups differ on the loss they consider, the type of regularization, the random feature matrices used, the training procedure, the asymptotic regime studied, or on the model architecture.

In the Gaussian model equivalent to our setup, the AF constants $\mu_0, \mu_1$, and $\mu_2$ appear as parameters. For example, $\mu_\star$ appears as the magnitude of noise added to the regressor matrix entries, and the non-linear part of the target in the RFR model appears as additive mismatch noise. As such, e.g., tuning $\mu_\star$ is related to an implicit ridge regularization. However, since in the Gaussian equivalent model $\mu_\star$ also appears as an effective model mismatch noise, tuning AFs leads to a richer behaviour than just tuning $\lambda$. Furthermore, tuning the AF requires tuning more than just one parameter, while tuning regularization only one, making our contribution in Sec. 3.2 all the more valuable. In fact, one of our contributions (cf. contribution 4 in Sec. 1) is quantifying the limitation of this connection: tuning AFs can lead to strictly better performance than tuning regularization (cf. Section 3.3).

Gaussian equivalent models derive a good portion of their importance from their connection to the original models to which their equivalence is proved, and which are typically closer to real-world use of neural networks. By themselves, these Gaussian models are extremely simplistic and lack basic real-world components, such as the concept of AF that we study here. Hypothesizing an equivalence to a Gaussian models greatly facilitates analytical advances and numerous unproven conjectures have been put forth regarding how generally these equivalences can be established Goldt et al. (2022); Loureiro et al. (2021); Dhifallah & Lu (2020).

## 2.4 Advantages and limitations of studying the RFR model

It is known (Mei & Montanari, 2022) that in the asymptotic proportional regime, the RFR cannot learn the non-linear component of certain families of non-linear functions, and in fact cannot do better than linear regression on the input for these functions. Ba et al. (2022) show that a single not-small gradient step to improve the initially random weights of RFR's first layer allows surpassing linear regression's performance in the asymptotic proportional regime. However, for not-small steps, no explicit asymptotic MSE or sensitivity formulas are given that one could use to tune AFs parameters. Also, Ba et al. (2022), and others, e.g. Hu & Lu (2020), work with a slightly different class of functions than Mei & Montanari (2022), e.g. their AFs are odd functions, making comparisons not apples-to-apples. It is known that the RFR can learn non-linear functions in other regimes, e.g. $n \sim \text{poly}(d)$, and asymptotic formulas for the RFR in this setting also exist Misiakiewicz (2022). There is numerical evidence of the real-word usefulness of the RFR (Rahimi & Recht, 2007b).

Linear regression also exhibits a double descent curve in the asymptotic proportional regime (Hastie et al., 2019). However, under e.g. overparameterization this curve exhibits a minimizer at a finite $\psi_2$, while empirical evidence for real networks shows that the error decreases monotonically as $\psi_2 \to \infty$. Therefore, linear regression is not as good as the RFR to explain observed double-descent phenomena. Furthermore, linear regression does not deal with AFs, which is our object of study. Finally, even in a setting where the RFR cannot learn certain non-linear functions with zero MSE, it remains an important question to study how much tuning AF can help improve the MSE and how this affects properties like the double descent curve.

## 3 Main results

We will find the simplest AFs that lead to the best trade-off between approximation accuracy and sensitivity for the RFR model. Mathematically, we will solve (8). From the theorems in Section 2 we know that $\mathcal{E}$ and $\mathcal{S}$, and hence $L = (1 - \alpha)\mathcal{E} + \alpha\mathcal{S}$, only depend on the AF via $\mu_0, \mu_1, \mu_2$. Therefore, we will proceed in two steps. In Section 3.1, we will fix $\mu_0, \mu_1, \mu_2$, and find $\sigma$ with associated values $\mu_0, \mu_1, \mu_2$ that has minimal norm, either (4) or (3). In Section 3.2, we will find values of $\mu_0, \mu_1, \mu_2$ that minimize $L = (1 - \alpha)\mathcal{E} + \alpha\mathcal{S}$. Together, these specify optimal AFs for the RFR model.

It is the case that properties of the RFR model other than the test error and sensitivity also only depend on the AF via $\mu_0, \mu_1, \mu_2$. One example is the robustness of the RFR model to disparities between the training and test data distribution (Tripuraneni et al., 2021). Although we do not focus on these other properties, the results in Section (3.1) can be used to generate optimal AFs for them as well, as long as, similar to in Section 3.2, we can obtain $\mu_0, \mu_1, \mu_2$ that optimize these other properties.

We made the decision to, as often as possible, simplify expressions by manipulating them to expose the signal to noise ratio $\rho = F_1^2/(\tau^2 + F_\star^2)$, $F_1 > 0$, rather than using the variables $F_1, \tau$, and $F_\star$. The only downside is that conclusions in the regime $\tau = F_\star = 0$ require a bit more of effort to be extracted, often been readable in the limit $\rho \to \infty$.

The complete proofs of our main results can be found in Appendix M and their main ideas below. The proofs of Section 3.2 are algebraically heavy and we provide a Mathematica file to symbolically check expressions of theorem statements and proofs in the supplementary material.

### 3.1 Optimal activation functions given fixed $\mu_0, \mu_1$, and $\mu_2$

Since one of our goals is knowing when an optimal AF is linear we start with the following lemma.

**Lemma 3.1.** *The AF $\sigma$ is linear (almost surely) if and only if $\mu_\star^2 \triangleq \mu_2 - \mu_1^2 - \mu_0^2 = 0$.*

We now state results for the norms (4) and (3). The problem we will solve under both norms is similar. Let $Z \sim \mathcal{N}(0, 1)$. We consider solving the following functional problem, where $i = 1$ or $2$,

$$\min_{\sigma} \|\sigma\|_i \text{ subject to } \mathbb{E}(\sigma(Z)) = \mu_0, \mathbb{E}(Z\sigma(Z)) = \mu_1, \mathbb{E}(\sigma(Z)^2) = \mu_2, \text{ with } Z \sim \mathcal{N}(0, 1). \quad (19)$$

If $i = 2$, we seek solutions over the Gaussian-weighted Lebesgue space of twice weak-differentiable functions that have $\mathbb{E}((\sigma(Z))^2)$ and $\mathbb{E}((\sigma'(Z))^2)$ defined and finite. If $i = 1$, we seek solutions over the Gaussian-weighted Lebesgue space of weak-differentiable functions that have $\mathbb{E}((\sigma(Z))^2)$ and $\mathbb{E}(|\sigma'(Z)|)$ defined and finite. The derivative $\sigma'$ is to be understood in a weak sense.

Since $\sigma$ is a one-dimensional function, the requirement of existence of weak derivative implies that there exists a function $v$ that is absolute continuous and that agrees with $\sigma$ almost everywhere (Rudin et al., 1976). Therefore, any specific solution we propose should be understood as an equivalent class of functions that agree with $v$ up to a set of measure zero with respect to the Gaussian measure.

**Theorem 7.** *The minimizers of* (19) *for* $i = 2$, *i.e.* $\|\sigma\|^2 = \mathbb{E}((\sigma'(Z))^2)$, *are*

$$\sigma(x) = ax^2 + bx + c, \text{ where } a = \pm\mu_\star/\sqrt{2}, b = \mu_1, \text{ and } c = \mu_0 - a. \tag{20}$$

In Theorem 7, if $\mu_\star = 0$ there is only one minimizer, a linear function. If $\mu_\star > 0$, there are exactly two minimizers, both quadratic functions. Note that both minimizers satisfy the growth constraints of Assumption 1, and hence can be used within the analysis of the RFR model. We note that quadratic AFs have been empirically studied in the past, e.g. Wuraola & Patel (2018).

**Theorem 8.** *One minimizer of* (19) *for* $i = 1$, *i.e.* $\|\sigma\| = \mathbb{E}(|\sigma'(Z)|)$, *is*

$$\sigma(x) = \mu_0 + b \max\{\min\{x, -s\}, s\}, \tag{21}$$

*where* $b = \frac{\mu_1}{erf(s/\sqrt{2})}$, *erf is the Gauss error function, and* $s \in \mathbb{R}$ *is the unique solution to the equation* $\zeta^2 \triangleq \mu_1^2/\mu_\star^2 = g(s)$ *if* $\mu_\star \neq 0$, *and* $s = +\infty$ *if* $\mu_\star = 0$, *where g is specified in Appendix I.*

When $\|\sigma\| = \mathbb{E}(|\sigma'(Z)|)$, we can characterize the complete solution family to (19). These are AFs of the form $\sigma(x) = a + b \max\{s, \min\{t, x\}\}$, where $a, b, s$, and $t$ are chosen such that the constraints in (19) hold. It is possible to explicitly write $a$ and $b$ as a function of $\mu_0, \mu_1, s, t$, and express $s, t$ as the solution of $E(s, t) = \mu_1^2/\mu_\star^2$, where $E(\cdot, \cdot)$ has explicit form. In this case, for each $\mu_0, \mu_1, \mu_2$ there are an infinite number of optimal AFs since $E(s, t) = \mu_1^2/\mu_\star^2$ has an infinite number of solutions. ReLU's are included in this family as $t \to \infty$. The involved lengthy expressions do not bring any new insights, so we state and prove only Thr. 8, which is a specialization of the general theorem to $s = -t$.

*Proofs' main ideas:* We give the main ideas behind the proof of Theorem 7. The proof of Theorem 8 follows similar techniques. The first-order optimality conditions imply that $-2x\sigma'(x) + 2\sigma''(x) + \lambda_1 + \lambda_2 x + \lambda_3 \sigma(x) = 0$, where the Lagrange multipliers $\lambda_1, \lambda_2$, and $\lambda_3$ must be later chosen such that $\mathbb{E}\{\sigma(Z)\} = \mu_0$, $\mathbb{E}\{Z\sigma(Z)\} = \mu_1$, and $\mathbb{E}\{\sigma^2(Z)\} = \mu_2$. Using the change of variable $\sigma(x) = \tilde{\sigma}(x/\sqrt{2}) - \lambda_1/\lambda_3 - x\lambda_2/(\lambda_3 - 1)$ we obtain $-2x\tilde{\sigma}'(x) + \tilde{\sigma}''(x) + \lambda_3\tilde{\sigma}(x) = 0$ which is the *Hermite ODE*, which is well studied in physics, e.g. it appears in the study of the quantum harmonic oscillator. The cases $\lambda_3 \in \{0, 3\}$ require special treatment. Using a finite energy/norm condition we can prove that $\lambda_3$ is quantized. In particular $\lambda_3 = 4k, k = 1, 2, ...,$ which implies that $\sigma(x) = -\lambda_1/\lambda_3 - \lambda_2 x/(\lambda_3 - 2) + cH_{2k}(x/\sqrt{2})$, where $H_i$ is the $i$th *Hermite polynomial* and $c$ a constant. The energy/norm is minimal when $k = 1$, which implies a quadratic AF. □

### 3.2 OPTIMAL ACTIVATION FUNCTION PARAMETERS

We will find AF parameters that minimize a linear combination of sensitivity and test error. We are interested in an asymptotic analytical treatment in the three regimes mentioned in Section 2. To be specific, we will compute

$$\mathcal{U}_{R_i}(\psi_1, \psi_2, \tau, \alpha, F_1, F_\star, \lambda) \equiv \arg\min_{\mu_0, \mu_1, \mu_2} \quad (1 - \alpha)\mathcal{E}_{R_i}^\infty + \alpha\mathcal{S}_{R_i}^\infty, \text{ where } i = 1, 2, \text{ or } 3. \tag{22}$$

We are not aware of previous work explicitly studying the trade-off between $\mathcal{E}$ and $\mathcal{S}$ for the RFR model. For the RFR model, the work of Mei & Montanari (2022) studies only the test error and D'Amour et al. (2020) studies a definition of sensitivity related but different from ours. Other papers have studied trade-offs between robustness and error measures related but different than ours and for other models, e.g. Tsipras et al. (2018); Zhang et al. (2019).

To simplify our exposition, we do not present results for the edge case $\alpha = 1$, for which problem (22) reduces to minimizing the sensitivity. Below we focus on the case when $\alpha \in [0, 1)$.

**Special notation:** In Theorem 9 we use the following special notation. Given two potential choices for AF parameters, say $x$ and $y$, we define $x \sqcup y$ to mean that $x$ exists and that $y$ might exist or not, and that $x \sqcup y = y$ if $y$ exists and it leads to a smaller value of $(1 - \alpha)\mathcal{E} + \alpha\mathcal{S}$ than using $x$, and otherwise $x \sqcup y = x$. Note that $x \sqcup y$ and $y \sqcup x$ make different statements about the existence of $x$ and $y$. This notation is important to interpret the results of Table 1 in Theorem 9.

**Theorem 9.** *Let* $\alpha \in [0, 1)$, $\underline{\psi} \equiv \min\{\psi_1, \psi_2\}$ *and* $\overline{\psi} \equiv \max\{\psi_1, \psi_2\}$. *We have that*

$$\mathcal{U}_{R_1} = \left\{(\mu_0, \mu_1, \mu_2) : x\mu_1^2(-1 + x + \underline{\psi}) = \mu_\star^2(\underline{\psi} + x)\right\}, \text{ where } x \text{ is as in Table 1.} \tag{23}$$

*In Table 1, $x_1, x_2$, and $x_3$ are the smallest, second smallest and third smallest roots of a 4th degree polynomial $p(x)$, specified in Appendix J, in the range $(x_L, x_R) \triangleq (-\underline{\psi}, \min\{0, 1 - \overline{\psi}\})$, if these exists. The variables $\beta_1, \beta_2, \beta_3, \alpha_L, \alpha_C, \alpha_R$, the polynomial $p(x)$, and the conditions $E_1$ and $E_2$ are defined in Appendix J.2 when $\psi_1 < \psi_2$, and when $\psi_1 > \psi_2$ these are defined in Appendix J.3.*

| | $\beta_1 \leq \overline{\psi}$ | $\beta_2 < \overline{\psi} < \beta_1$ | $\beta_3 < \overline{\psi} \leq \beta_2$ | $\overline{\psi} \leq \beta_3$ |
|---|---|---|---|---|
| $(\alpha < \alpha_L) \wedge E_1$ | $x_R$ | $x_R \sqcup x_1$ | $x_R$ | $x_R$ |
| $(\alpha < \alpha_L) \wedge E_2 \wedge (\alpha > \alpha_C)$ | $x_1$ | $x_1 \sqcup x_3$ | $x_1$ | $--$ |
| $(\alpha < \alpha_L) \wedge E_2 \wedge (\alpha < \alpha_C)$ | $x_1$ | $x_1 \sqcup x_3$ | $x_1$ | $--$ |
| $(\alpha > \alpha_L) \wedge E_1 \wedge (\alpha > \alpha_C)$ | $--$ | $x_L$ | $x_L$ | $x_L$ |
| $(\alpha > \alpha_L) \wedge E_1 \wedge (\alpha < \alpha_C)$ | $--$ | $x_R$ | $x_R$ | $x_R$ |
| $(\alpha > \alpha_L) \wedge E_2$ | $x_L$ | $x_L \sqcup x_2$ | $x_L \sqcup x_2$ | $x_L$ |

Table 1: The optimal AFs (23) depends on $x$ according to this table. Cells with "- -" never happen. The values of $x_1, x_2, x_3, \beta_1, \beta_2, \beta_3, \alpha_L, \alpha_C, \alpha_R, x_L, x_R$, and the events $E_1$ and $E_2$ are specified below.

**Remark 2.** Excluding the $\alpha = 1$ scenario, it follows directly from (23) that the optimal AF is linear if and only if $x = x_R$. With this information and Table 1, we have all the information needed to find exactly when the optimal AF is, or is not, linear. For regime $R_1$, changing $\alpha$ alone can change the optimal AF from linear to non-linear and vice-versa (see e.g. 3rd column of Table 1), which justifies the observation 3 in Section 1.

**Remark 3.** For the cases considered in Table 1, $x$ is unique. When $\alpha \in \{\alpha_R, \alpha_L, \alpha_C\}$, or when $(\psi_1 > \psi_2) \wedge (\psi_1 \in \{A, B\})$, ($A\&B$ are defined in Appendix J.3), we can lose the uniqueness of $x$. Yet, we can still explicitly characterize the sets of optimal $x$ and of optimal AFs parameters. For simplicity we omit these cases from Thr. 9.

**Remark 4.** Theorem 9's proof gives relationships among Table 1's constants that imply that (1) no two rows/columns simultaneously hold and (2) in some cases some cells might not hold. See App. J.

We do not consider $\psi_1 = \psi_2$ in Theorem 9 because it implies $(1 - \alpha)\mathcal{E}_{R_1}^\infty + \alpha\mathcal{S}_{R_1}^\infty$ is not defined. Note that $\psi_1 = \psi_2$ has been called the *interpolation threshold* in the context of studying the generalization properties of the RFR model under vanishing regularization (Mei & Montanari, 2022). See Remark 1.

When $x \in \{x_L, x_R\}$ we can compute the optimal value of the objective explicitly. For example, if $\psi_1 < \psi_2$ and $x = x_L$ the optimal value of the objective is $\frac{(1-\alpha)(\psi_2(F_1^2 + F_\star^2) + \psi_1 \tau^2)}{\psi_2 - \psi_1}$. If $\psi_1 > \psi_2$ and $x = x_R$ the optimal value of the objective is $\frac{\alpha F_1^2(\psi_1 - \psi_2) + F_\star^2((\alpha-1)\psi_2 - \alpha) + \alpha(\psi_1 - 1)\tau^2 - \psi_1 \tau^2}{\psi_1 - \psi_2}$ if $\psi_1 < 1$ and $\frac{F_1^2(\alpha(2\psi_1 - 1)(\psi_1 - \psi_2) + (\psi_1 - 1)\psi_2) + F_\star^2((\alpha-1)\psi_2 - \alpha\psi_1) - \psi_1 \tau^2}{\psi_1 - \psi_2}$ if $\psi_1 \geq 1$. This follows by substitution.

**Theorem 10.** *Let $\alpha \in [0, 1)$. We have that,*

$$\mathcal{U}_{R_2} = \left\{ (\mu_0, \mu_1, \mu_2) : \mu_1^2(-1 + 2\psi_2 - x)(-1 + x) = -2(\mu_\star^2 + \lambda\psi_2)(1 + x) \right\}, \tag{24}$$

*where $x$ is the unique solution to $p(x) = 0$ in the range $x \in (-1, \min\{1, -1 + 2\psi_2\})$, where $p(x) \triangleq p_0 + p_1 x + p_2 x^2 + p_3 x^3 + p_4 x^4$ with coefficients described in Appendix K.*

**Remark 5.** The only way to get $\mu_\star = 0$, and hence a linear optimal AF is if $x$ simultaneously satisfies $\mu_1^2(-1 + 2\psi_2 - x)(-1 + x) = -2\lambda\psi_2(1 + x)$ and $p(x) = 0$. Since the first equation does not depend on $\alpha$, but the zeros of $p(x) = 0$ change continuously with $\alpha$, only very special choices of parameters lead to linear AFs. In general regime $R_2$ does not have optimal linear AFs.

**Theorem 11.** *Let $\alpha \in [0, 1)$. We have that*

$$\mathcal{U}_{R_3} = \begin{cases} \{(\mu_0, \mu_1, \mu_2) : \mu_\star = 0 \wedge \mu_1 = \infty\}, & \text{if } \alpha = 0 \vee (\psi_1 = 1 \wedge 0 < \alpha \leq \frac{1}{4}) \\ \left\{(\mu_0, \mu_1, \mu_2) : \mu_\star = 0 \wedge \mu_1^2 = \frac{-4\alpha^2\lambda + 3\alpha\lambda + \sqrt{\alpha}\lambda}{16\alpha^2 - 8\alpha + 1}\right\}, & \text{if } \psi_1 = 1 \wedge \alpha > \frac{1}{4} \\ \{(\mu_0, \mu_1, \mu_2) : \mu_\star = 0 \wedge \mu_1^2(-1 + 2\psi_1 - x)(-1 + x) + 2\lambda\psi_1(1 + x) = 0\}, & \text{if } \psi_1 \neq 1 \end{cases} \tag{25}$$

*where $x$ is the unique solution to $p(x) = 0$ in the range $x \in (-1, \min\{1, -1 + 2\psi_2\})$, where $p(x)$ is define like in Theorem 10 but with $\rho \to \infty$ and with $\psi_2$ replaced by $\psi_1$.*

**Remark 6.** The optimal AF is always linear and independent of the noise variables $F_\star$ and $\tau$.

**Remark 7.** When $\psi_1 = 1, \alpha \leq \frac{1}{4}$ there is no optimal AF inside our AF search space since no AF can satisfy $\mu_1 = \infty$. Rather, there exists a sequence of valid AFs with decreasing $L$ whose $\mu_1 \to \infty$.

**Remark 8.** We can compute the optimal objective in closed-form in some scenarios. When $\alpha = 0$ the optimal objective is $F_\star^2$. When $\psi_1 = 1 \wedge 0 < \alpha \leq \frac{1}{4}$, the optimal objective approaches $\alpha F_1^2 + (1 - \alpha)F_\star^2$ as $\mu_1 \to \infty$. When $\psi_1 = 1 \wedge \alpha > \frac{1}{4}$, the optimal objective is $F_1^2(4\sqrt{\alpha} - 1 - 3\alpha) + F_\star^2(1 - \alpha)$.

*Proofs' main ideas:* We give the main ideas behind the proof of Theorem 10. The proof of Theorems 9 and 11 follows similar techniques but require more care. The objective $L$ only depends on AF

parameters via $\omega_2 = \omega_2(\psi_2, \mu_1^2, \mu_\star^2)$. We use the Möbius transformation $x = (1 + \omega_2)/(\omega_2 - 1)$ such that the infinite range $\omega_2 \in [-\infty, 0]$ gets mapped to the finite range $x \in [-1, 1]$. We then focus the rest of the proof on minimizing $L = L(x)$ over the range of $x$. First we show that given that $\mu_1^2, \mu_\star^2 \geq 0, \psi_1 > 0$, the range of $x$ can be reduced to $x \in [x_L, x_R] \triangleq [-1, \min\{1, -1 + 2\psi_2\}]$. Then we compute $dL/dx$ and $d^2L/dx^2$, which turn out to be rational functions of $x$. We then show that if $x \in [-1, \min\{1, -1 + 2\psi_2\}]$ then $d^2L/dx^2 > 0$, so $L$ is strictly convex. We also show that $dL/dx < 0$ at $x_L$ and $dL/dx > 0$ at $x_R$, thus $x_R$ and $x_L$ cannot be minimizers. Finally, we show that the zeros of the numerator $p(x)$ of the rational function $dL/dx$ differ from the denominator's zeros. So the optimal $x$ is the unique solution to $p(x)$ in $[x_L, x_R]$. $\qquad\square$

## 3.3 Important observations

Together, Sections 3.1 and 3.2 explicitly and fully characterize the solutions of (8) in the ridgeless, overparametrized, and large sample regimes. A few important observations follow from our theory. In Appendix L we discuss more on this topic and include details on the observations below.

**Observation 1**: In regime $R_1$, and if $\alpha = 0, \psi_1 < \psi_2$, the optimal AF is linear. This follows from Theorem 9 and Remark 2. Indeed, expressions simplify and we get that $p(x) = \rho\psi_2(\psi_1 - (x + \psi_1)^2)^2$ if $\psi_1 \neq 1$ or $p(x) = -(2 + x)^2\rho\psi_2$ if $\psi_1 = 1$, which implies that $x_1$ does not exist (since it would be outside of $(-x_L, x_R)$). Hence, the first row of Table 1 always gives $x = x_R$ and the optimal AF is linear. Also, when $\alpha = 0, \psi_1 < \psi_2$, we can explicitly compute the optimal objective (see paragraph before Theorem 10). If furthermore $\tau = F_\star = 0$, we can show that also when $\psi_1 > \psi_2$, $x_1$ does not exist and $x = x_R$, therefore the optimal objective and AF when $\psi_1 > \psi_2$ have the same formula as when $\psi_1 < \psi_2$. Hence, if $\alpha = \tau = F_\star = 0, \psi_1 < \psi_2$, from the formula one can conclude that choosing an optimal linear AF destroys the double descent curve if $\psi_2 > 1$[1], the test error becoming exactly zero for $\psi_1 \geq 1$. This contrasts with choosing a non-linear, sub-optimal, AF which will exhibit a double descent curve. This justifies observation 1 (low complexity $\psi_1 < \psi_2$) and observation 2 in Sec. 1. Fig. 1-(A,B) illustrates this and details the high-complexity ($\psi_1 > \psi_2$) observation.

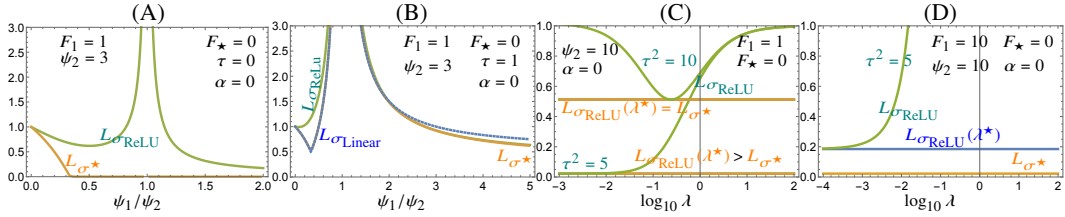

Figure 1: (A) Consider the regime $R_1$. In a noiseless setting, if $\psi_2 > 1$, the evolution of $L$ versus $\psi_1$, when an optimal linear AF $\sigma^\star$ is used, can achieve 0 test error for $\psi_1 \geq 1$. However, if a non-linear $\sigma_{\text{ReLU}}$ is used, we observe the typical double descent curve. (B) Consider the regime $R_1$. If there is observation noise $\tau > 0$, the evolution of $L$ versus $\psi_1$ with a linear AF $\sigma_{\text{linear}}$ is only optimal for $\psi_1 < \psi_2$. For $\psi_1 > \psi_2$, $L$ is optimal for a linear AF until $\psi_1 < C$ ($C = 5$ for the parameters here). For $\psi_1 > C$ a non-linear AF $\sigma_\star$, here close to but different from a ReLU, achieves minimal $L$. (C) Consider the regime $R_2$. When a ReLU is used (green curves), the evolution of $L$ versus $\lambda$ for both low and high Signal to Noise Ratio (SNR) $\rho$ is only optimal for a special choice of $\lambda$, achieving the minimum $L_{\sigma_{\text{ReLU}}}(\lambda^\star)$. However, also for the same low and high SNR settings, when an optimal (non-linear) AF is used (orange curves), we obtain the same, or slightly better, $L_{\sigma^\star}$ regardless of any careful choice for $\lambda$. For low SNR ($\tau^2 = 10$) we have $L_{\sigma^\star} = L_{\sigma_{\text{ReLU}}}(\lambda^\star) = 0.512$ and for high SNR ($\tau^2 = 5$) we get $L_{\sigma_{\text{ReLU}}}(\lambda^\star) = 0.0220 > L_{\sigma^\star} = 0.0217$. (D) In a situation just like in (C) but with even higher SNR, the difference between the minimum $L$ that can be achieved with a particular choice of $\lambda$ (blue line ordinate value $L_{\sigma_{\text{ReLU}}}(\lambda^\star)$) and the value of $L$ with any choice of $\lambda$ but with an optimal (non-linear) AF (orange line ordinate value $L_{\sigma^\star}$) becomes clearly visible. Both (C) and (D) show that optimally tuning AFs can be different from optimally tuning regularization. Tuning AFs is always better or equal to tuning $\lambda$, showing the limits of the connection between AFs and implicit regularization when Gaussian equivalence holds (cf. Section 2.3). We include inside of each plot the parameters used. See Appendix L.1 for how to reproduce this figure.

**Observation 2**: In regime $R_2$, looking at Theorem 2 and Theorem 5, one sees that both $\mathcal{E}$ and $\mathcal{S}$, and hence the objective $L$ (cf. (5)), only depend on the optimal AF parameters via $\omega_2$. In particular, we can solve (22) by searching for the $\omega_2$ that achieves the smallest objective. Given the definition of $\omega_2 = \omega_2(\psi_2, \zeta^2, \mu_\star, \lambda)$ in (32), fixing $\lambda$ and changing $\zeta$ or $\mu_\star$ always allows one to span a larger range of values for $\omega_2$ than fixing the AF's parameters $\zeta, \mu_\star$ and changing $\lambda$. In particular, a tedious calculation shows that in the first case the achievable range for $\omega_2$ is $[\frac{\psi_2}{\min\{0^-, -1+\psi_2\}}, 0]$ which contains the range

---

[1]If $\psi_2 < 1$ the optimal AF is still linear but the explosion at the interpolation threshold $\psi_1 = \psi_2$ remains.

in the second case which is $\left[\frac{1}{2}\left(\zeta^2\left(-\psi_2\right)-\sqrt{\zeta^2\left(\psi_2\left(\zeta^2\left(\psi_2-2\right)+2\right)+\zeta^2+2\right)+1}+\zeta^2+1\right),0\right]$. This implies that while for a fixed AF one needs to tune $\lambda$ during learning for best performance, if an optimal AF is used, regardless of $\lambda$, we always achieve either equal or better performance. This justifies the observation 4 made in Section 1. This is illustrated in Figure 1-(C,D).

In Appendix N we have experiments involving real data that show consistency with these observations. The supplementary material has code to generate Fig. 1 and the figures in Appendix N for real data.

## 4    CONCLUSION AND FUTURE WORK

We found optimal Activation Functions (AFs) for the Random Features Regression model (RFR) and characterized when these are linear, or non-linear. We connected the best AF to use with the regime in which RFR operates: e.g. using a linear AF can be optimal even beyond the interpolation threshold; in some regimes optimal AFs can replace, and outperform, regularization tuning.

We reduced the gap between the practice and theory of AFs' design, but parts remain to be closed. For example, we could only obtain explicit equations for optimal AFs under two functional norms in the optimization problem from which we extract them. One could explore other norms in the future. One could also explore adding higher order moment restrictions to the AF since some of these higher order constraints appear in the theoretical analysis of neural models (Ghorbani et al., 2021).

One open problem is determining, both numerically and analytically, how generic our observations are. One could numerically compute optimal AFs for several models, target functions, and regimes beyond the ones we considered here, and determine how the conditions under which the optimal AF is, or not, linear compare with the conditions we presented. We suspect that the choice of target function affects our conclusions. In fact, even for our current results, the amount of non-linearity in our target function affects our conclusions. In particular, it can affect the optimal AF being linear or not linear (this is visible in Theorem 9 in its dependency on $\rho$, cf. (12), via the polynomial $p(x)$).

Another future direction would be to study optimal AFs when the first layer in our model is also trained, even if with just one gradient step. For this model there are asymptotic expressions for the test error (Ba et al., 2022) to which one could apply a similar analysis as in this paper. One could also study the RFR model under different distributions for the random weights, including the *Xavier distribution* (Glorot & Bengio, 2010) and the *Kaiming distribution* (He et al., 2015). Some experimental results are included in Appendix O. Regarding the use of different distributions, we note the following: The key technical contribution in Mei & Montanari (2022) is the use of random matrix theory to show that the spectral distribution of the Gramian matrix obtained from the Regressor matrix in the ridge regression is asymptotically unchanged if the Regressor matrix is recomputed by replacing the application of an AF element wise by the addition of Gaussian i.i.d. noise. Because many universality results exist in random matrix theory, we expect that for other choices of random weights distributions, exactly the same asymptotic results would hold. The first thing to try to prove would be similar results for from well-known random matrix ensembles. We note that Gerace et al. (2020) provides very strong numerical evidence that this is true for other matrix ensembles, and stronger results are known in what concerns just spectral equivalence (Benigni & Pé ché, 2021).

One could study both the RFR and other models in regimes other than the asymptotic proportional regime. The work of Misiakiewicz (2022) is a good lead since it provides asymptotic error formulas derived for our setup but when $n \sim \text{poly}(d), d \to \infty$. In this regime, the RFR can learn non-linear target functions with zero MSE (Misiakiewicz, 2022; Ghorbani et al., 2021). These formulas are equivalent to the ones in Mei & Montanari (2022) after a renormalization of parameters, a reparametrisation that depend on a problem's constants, as noted in Lu & Yau (2022). It is unclear if this reparametrisation would change the high-level observations from our work but we expect it to change their associated low-level details, like Table 1's thresholds. It would be interesting to make these same investigations for more realistic neural architectures, such as Belkin et al. (2019a) and Nakkiran et al. (2019), for which phenomena such as the double descent curve is well documented.

Finally, it would be interesting to design AFs for an RFR model that optimizes a combination of test error and robustness to test/train distribution shifts and adversarial attacks. The starting point would be Tripuraneni et al. (2021); Hassani & Javanmard (2022) (cf Appendix C). The results of Hassani & Javanmard (2022) would need to be generalized from a ReLU to general AFs before one could optimize the AFs' parameters.

ACKNOWLEDGMENTS

We thank Song Mei for valuable discussions regarding the asymptotic properties of the Random Feature Regression model. We thank Piotr Suwara for his help regarding Hermite-type differential equations and their solutions.

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

## A    COMMENT ON THE CHOICE OF METRICS

A few reason for us choosing norms (3) and (4) in our setup are the following.

- We use the L1 and L2 norms because these are two of the most widely used functional norms.
- We use these norms on a Gaussian-weighted space because the dependency of performance on the activation functions (AFs) from prior work that we build involves Gaussian-weighted measures. To be specific, both the sensitivity and error discussed in Section 2.1 and Section 2.2 depend on the AF only via $\mu_0, \mu_1$, and $\mu_2$, defined in (9), and these in turn are defined using a Gaussian distribution. In other words, Gaussian-weighted spaces are the natural space in our high-dimensional setting.
- We focus on the derivative of the AF to impose the notion that the AF cannot have sudden changes, i.e. needs to be simple.

We are aware that other choices for functional norms are possible, e.g. taking higher-order derivatives and/or higher-order moments of the AFs, and we plan to investigate them in future work as mentioned in Section 4.

## B    COMMENT ON THE CHOICES OF SENSITIVITY

This appendix pertains the choice of definition for sensitivity in (7).

We want to relate $\mathbb{E}(\|\nabla_x f_{a^\star, \Theta}(x)\|^2)$ with $\|\mathbb{E}(\nabla_x f_{a^\star, \Theta}(x))\|^2$. The expectations can be taken just with respect to the test data $x$ since in the asymptotic regime these quantities concentrate around their expected values with respect to the other random variables.

The gradient $\nabla_x f_{a^\star, \Theta}(x)$ equals $a^{\star T}(R\Theta/\sqrt{d})$, where $R = \mathrm{diag}(\sigma'(\Theta x/\sqrt{d}))$, and for some vector $v$, $\mathrm{diag}(v)$ is a diagonal matrix with diagonal equal to $v$. We can write

$$\mathbb{E}\|\nabla_x f_{a^\star, \Theta}(x)\|^2 = \mathbb{E}\sum_k (\sum_i a_i^\star R_{ii}\Theta_{ik}/\sqrt{d})^2 = \sum_{i,j,k} a_i^\star a_j^\star \mathbb{E}(R_{ii}R_{jj})\Theta_{ik}\Theta_{jk}/d. \qquad (26)$$

Following D'Amour et al. (2020), Appendix E.5, when $d \to \infty$, we use the fact that $\langle \theta_i x \rangle/\sqrt{d} \to Z$, where $Z \sim \mathcal{N}(0,1)$, and hence that $\mathbb{E}\,R_{ii} = \mathbb{E}\,\sigma'(\langle \theta_i, x \rangle/\sqrt{d}) = \mathbb{E}(\sigma'(Z)) + o(1) = \mu_1 + o(1)$, to compute $\mathbb{E}(R_{ii}R_{jj})$.

We need to consider two scenarios. If $i \neq j$, then $\mathbb{E}\,R_{ii}R_{jj} = \mathbb{E}\,R_{ii}\,\mathbb{E}\,R_{jj} = \mu_1^2$. If $i = j$, then $\mathbb{E}\,R_{ii}R_{jj} = \mathbb{E}(\sigma'(Z))^2$. Let us define $\mu_3 = \mathbb{E}(\sigma'(Z)^2)$. Replacing the formulas for $\mathbb{E}\,R_{ii}R_{jj}$ in (26) we get that

$$\mathbb{E}\|\nabla_x f_{a^\star, \Theta}(x)\|^2 = \mu_1^2\|a^{\star T}\Theta\|^2/d + (\mu_3 - \mu_1^2)\sum_{i,k} a_i^{\star 2}\Theta_{i,k}^2/d. \qquad (27)$$

The first term of the r.d.s. of (27) is exactly eq. (19) on D'Amour et al. (2020) for which we are given asymptotic expression and which we use as the basis of Theorems 4-6, which are themselves a specialization of Theorem 13. The second term is non-negative since by Jensen's inequality $\mu_3 = \mathbb{E}(\sigma'(Z)^2) \geq \mathbb{E}(\sigma'(Z))^2 = \mu_1^2$. Therefore, $\mathbb{E}(\|\nabla_x f_{a^\star, \Theta(x)}\|^2) \geq \|\mathbb{E}(\nabla_x f_{a^\star, \Theta(x)})\|^2$.

The results we present, especially those in Section 3.2, are already extremely complex to state and to interpret, even with us essentially optimizing only two parameters, $\mu_1$ and $\mu_2$ ($\mu_0$ can be assumed 0 without loss of generality). Stating results on optimizing an extra parameter $\mu_3$ would make our exposition even more complex, with many more special cases. Only in the specific case where we choose the objective (4) for the outer optimization problem (8), it clear that $\mu_3$ is determined from $\mu_0, \mu_1$ and $\mu_2$, and hence it is clear that there is no added complexity in the number of parameters to optimize. However, we still need to find an asymptotically formula for the second term in (27).

At the same time, while the first term in (27) can be expressed as the trace of a product of random matrices, making it easier to use random matrix theory to get asymptotic formulas for it, the second term does not easily yield to a similar type of analysis.

## C   MORE ON RELATED WORK

First attempts to optimize AFs include Poli (1996), Weingaertner et al. (2002), and Khan et al. (2013), where genetic and evolutionary algorithms were used to learn how to numerically combine different AFs from a library into the same network. More recently, Ramachandran et al. (2017) used reinforcement learning to empirically discover AFs that minimize test accuracy. Their search was done over AFs that were a combination of basic units. This work produced the *Swish* AF. Similarly, Goyal et al. (2020) defined AFs as the weighted sum of a pre-defined basis and searched for optimal weights via training. Unser (2019) provided a theoretical foundation to simultaneously learn a NN's weights and continuous piecewise-linear AFs. They showed that learning in their framework is compatible with learning in current existing deep-ReLU, parametric ReLU, APL (adaptive piecewise-linear) and MaxOut architectures. Tavakoli et al. (2021) parameterized continuous piece-wise linear AFs and numerically learnt their parameters to improve both accuracy and robustness to adversarial perturbations. They numerically compared the performance of their SPLASH framework with that of using ReLUs, leaky-ReLUs (Maas et al., 2013), PReLUs (He et al., 2015), tanh units, sigmoid units, ELUs (Clevert et al., 2015), maxout units (Goodfellow et al., 2013), Swish units, and APL units (Agostinelli et al., 2014). Similarly, Zhou et al. (2021) parameterized AFs as piece-wise linear units and learnt the AFs parameters to optimize different tasks. Banerjee et al. (2019) proposed an empirical method to learn variations of ReLUs. Bubeck et al. (2020) studied 2-layer NNs and gave a condition on the Lipschitz constant of a polynomial AF for the network to perfectly fit data. They related this condition to the model's parameter-size and robustness and numerically related the number of ReLUs in the model to its robustness.

Several papers proposed new AFs and empirically studied their performance without systematically tuning them. Milletarí et al. (2019) identified ReLU and Swish as naturally arising components of a statistical mechanics model. Rozsa & Boult (2019) introduced a "tent"-shaped AF that improves robustness without adversarial training, while not hurting the accuracy on non-adversarial examples. Zhou et al. (2020) proposed an AF called SRS that can overcome the non-zero mean, negative missing, and unbounded output in ReLUs. Their work was purely empirical. Wuraola & Patel (2018) developed the SQuared Natural Law AF. Nicolae (2018) proposed the Piece-wise Linear Unit AF.

The RFR model was introduced by Rahimi & Recht (2007a) as a way to project input data into a low dimensional random features space and it has since then been studied considerably. A great part of the literature has drawn connections between the expressive power of NNs and that of the RFR model, often via the study of Gaussian processes. For example, Williams (1996) did this in the context of shallow but infinitely wide NN and the works Garriga-Alonso et al. (2019); Novak et al. (2019); de G. Matthews et al. (2018); Hazan & Jaakkola (2015) did this for deep networks. Daniely et al. (2016); Daniely (2017) connected the RFR model to training a NN with gradient descent.

In addition to Mei & Montanari (2022); D'Amour et al. (2020), already discussed, other papers studied the approximation properties of the RFR model. Ghorbani et al. (2021) studied both the RFR model and the neural tangent kernel model and provided conditions under which these models can fit polynomials in the raw features up to a maximum degree. These conditions were provided under two regimes, when $n \to \infty$ and $N, d$ large but finite, or when $N \to \infty$ and $n, d$ large but finite. Their results hold under weak assumptions on the AFs. Tripuraneni et al. (2021) used the RFR model to compute how robust the test error is to distribution shifts between training and test data. This was done in a high-dimensional asymptotic limit when random features and training data are normal distributed. The derivations hold for a generic AF that satisfies some mild assumptions similar to the assumptions in this paper. Hassani & Javanmard (2022) characterized the role of overparametrization on the adversarial robustness for the RFR model under an asymptotic regime when learning a linear function with normal-distributed random weights and normal samples. Their AF was a shifted ReLU.

Finally, a few papers have studied the behavior of models similar to the RFR but within a different context. For example, Taheri et al. (2021) and Bean et al. (2013) seek to compute the optimal loss function under similar asymptotic regimes of large data sets.

## D   DETAILS REGARDING THEOREM 1

The definition of the functions $\mathscr{E}_{0,\text{rless}}$, $\mathscr{E}_{1,\text{rless}}$, $\mathscr{E}_{2,\text{rless}}$ and $\chi$ is as follows.

$$\mathcal{E}_{0,\text{rless}}(\zeta,\psi_1,\psi_2,\chi) \equiv -\chi^5\zeta^6 + 3\chi^4\zeta^4 + (\psi_1\psi_2 - \psi_2 - \psi_1 + 1)\chi^3\zeta^6 - 2\chi^3\zeta^4 - 3\chi^3\zeta^2$$
$$+ (\psi_1 + \psi_2 - 3\psi_1\psi_2 + 1)\chi^2\zeta^4 + 2\chi^2\zeta^2 + \chi^2 + 3\psi_1\psi_2\chi\zeta^2 - \psi_1\psi_2,$$
$$\mathcal{E}_{1,\text{rless}}(\zeta,\psi_1,\psi_2,\chi) \equiv \psi_2\chi^3\zeta^4 - \psi_2\chi^2\zeta^2 + \psi_1\psi_2\chi\zeta^2 - \psi_1\psi_2, \text{ and}$$
$$\mathcal{E}_{2,\text{rless}}(\zeta,\psi_1,\psi_2,\chi) \equiv \chi^5\zeta^6 - 3\chi^4\zeta^4 + (\psi_1 - 1)\chi^3\zeta^6 + 2\chi^3\zeta^4 + 3\chi^3\zeta^2 + (-\psi_1 - 1)\chi^2\zeta^4 - 2\chi^2\zeta^2 - \chi^2,$$
$$\text{where } \chi = \chi(\zeta,\psi) \equiv -\left(\sqrt{(\psi\zeta^2 - \zeta^2 - 1)^2 + 4\zeta^2\psi} + \psi\zeta^2 - \zeta^2 - 1\right) \bigg/ \left(2\zeta^2\right). \tag{28}$$

Note that all functions are a function of the square of $\zeta$, i.e. $\zeta^2$. Note also that $\mathcal{E}_{0,\text{rless}}$, $\mathcal{E}_{1,\text{rless}}$, $\mathcal{E}_{2,\text{rless}}$ are polynomials of their respective variables.

## E   DETAILS REGARDING THEOREM 2

$$\mathcal{B}_{\text{wide}}(\zeta,\psi_2,\overline{\lambda}) \equiv (\psi_2\omega_2 - \psi_2)/((\psi_2 - 1)\omega_2^3 + (1 - 3\psi_2)\omega_2^2 + 3\psi_2\omega_2 - \psi_2), \tag{29}$$
$$\mathcal{V}_{\text{wide}}(\zeta,\psi_2,\overline{\lambda}) \equiv (\omega_2^3 - \omega_2^2)/((\psi_2 - 1)\omega_2^3 + (1 - 3\psi_2)\omega_2^2 + 3\psi_2\omega_2 - \psi_2), \tag{30}$$
$$\text{where} \tag{31}$$
$$\omega_2 \equiv -\left(\sqrt{(\psi_2\zeta^2 - \zeta^2 - \overline{\lambda}\psi_2 - 1)^2 + 4\psi_2\zeta^2(\overline{\lambda}\psi_2 + 1)} + \psi_2\zeta^2 - \zeta^2 - \overline{\lambda}\psi_2 - 1\right) \bigg/ \left(2(\overline{\lambda}\psi_2 + 1)\right). \tag{32}$$

## F   DETAILS REGARDING THEOREM 3

$$\mathcal{B}_{\text{lsamp}}(\zeta,\psi_1,\overline{\lambda}) \equiv (((\omega_1^3 - \omega_1^2)/\zeta^2) + \psi_1\omega_1 - \psi_1)/((\psi_1 - 1)\omega_1^3 + (1 - 3\psi_1)\omega_1^2 + 3\psi_1\omega_1 - \psi_1), \tag{33}$$
$$\text{where} \tag{34}$$
$$\omega_1 \equiv -\left(\sqrt{(\psi_1\zeta^2 - \zeta^2 - \overline{\lambda}\psi_1 - 1)^2 + 4\psi_1\zeta^2(\overline{\lambda}\psi_1 + 1)} + \psi_1\zeta^2 - \zeta^2 - \overline{\lambda}\psi_1 - 1\right) \bigg/ \left(2(\overline{\lambda}\psi_1 + 1)\right). \tag{35}$$

## G   GENERAL THEOREMS FOR THE ASYMPTOTIC MEAN SQUARED TEST ERROR AND SENSITIVITY OF THE RFR MODEL

This appendix is referenced at the start of Section 2.1 and at the start of Section 2.2.

**Theorem 12** (Theorem 2 Mei & Montanari (2022)). *If assumptions 1, 2, 3 hold, and for any value of the regularization parameter $\lambda > 0$, the asymptotic test error* (6) *for the RFR satisfies*

$$\mathcal{E} \xrightarrow{p} \mathcal{E}^\infty \equiv F_1^2 \mathcal{B}(\zeta,\psi_1,\psi_2,\lambda/\mu_\star^2) + (\tau^2 + F_\star^2)\mathcal{V}(\zeta,\psi_1,\psi_2,\lambda/\mu_\star^2) + F_\star^2, \tag{36}$$

*where $\xrightarrow{p}$ denotes convergence in probability when $d \to \infty$ with respect to the training data $\mathcal{D}$, the random features $\Theta$, and the random target function $f_d$, and where $\mathcal{B}(\zeta,\psi_1,\psi_2,\overline{\lambda}) \equiv \mathcal{E}_1(\zeta,\psi_1,\psi_2,\overline{\lambda})/\mathcal{E}_0(\zeta,\psi_1,\psi_2,\overline{\lambda})$, $\mathcal{V}(\zeta,\psi_1,\psi_2,\overline{\lambda}) \equiv \mathcal{E}_2(\zeta,\psi_1,\psi_2,\overline{\lambda})/\mathcal{E}_0(\zeta,\psi_1,\psi_2,\overline{\lambda})$, and the functions $\mathcal{E}_0$, $\mathcal{E}_1$, $\mathcal{E}_2$ are defined as follows,*

$$\mathcal{E}_0(\zeta,\psi_1,\psi_2,\overline{\lambda}) \equiv -\chi^5\zeta^6 + 3\chi^4\zeta^4 + (\psi_1\psi_2 - \psi_2 - \psi_1 + 1)\chi^3\zeta^6 - 2\chi^3\zeta^4 - 3\chi^3\zeta^2$$
$$+ (\psi_1 + \psi_2 - 3\psi_1\psi_2 + 1)\chi^2\zeta^4 + 2\chi^2\zeta^2 + \chi^2 + 3\psi_1\psi_2\chi\zeta^2 - \psi_1\psi_2,$$
$$\mathcal{E}_1(\zeta,\psi_1,\psi_2,\overline{\lambda}) \equiv \psi_2\chi^3\zeta^4 - \psi_2\chi^2\zeta^2 + \psi_1\psi_2\chi\zeta^2 - \psi_1\psi_2,$$
$$\mathcal{E}_2(\zeta,\psi_1,\psi_2,\overline{\lambda}) \equiv \chi^5\zeta^6 - 3\chi^4\zeta^4 + (\psi_1 - 1)\chi^3\zeta^6 + 2\chi^3\zeta^4 + 3\chi^3\zeta^2 + (-\psi_1 - 1)\chi^2\zeta^4 - 2\chi^2\zeta^2 - \chi^2,$$

*where $\chi(\psi_1,\psi_2,\overline{\lambda}) \equiv \nu_1(i(\psi_1\psi_2\overline{\lambda})^{1/2}) \cdot \nu_2(i(\psi_1\psi_2\overline{\lambda})^{1/2})$, and $\nu_1$ and $\nu_2$ are two functions specified in Def. 1 in Mei & Montanari (2022).*

**Theorem 13** (Eq. (19) D'Amour et al. (2020)). *If assumptions 1, 2 and 3 hold, and for any value of the regularization parameter $\overline{\lambda} \equiv \lambda/\mu_\star^2 > 0$, the asymptotic sensitivity for the RFR, namely equation (7) with $f = f_{a,\Theta}$ where $f_{a,\Theta}$ is defined as in equations (1) and (2), satisfies*

$$\mathcal{S} \xrightarrow{p} \mathcal{S}^\infty \equiv \mu_1^2 \left( \frac{F_1^2}{\mu_\star^2} \cdot \frac{\mathscr{D}_1(\zeta, \psi_1, \psi_2, \overline{\lambda})}{(\chi \zeta^2 - 1)\mathscr{D}_0(\zeta, \psi_1, \psi_2, \overline{\lambda})} + \frac{\tau^2 + F_\star^2}{\mu_\star^2} \cdot \frac{\mathscr{D}_2(\zeta, \psi_1, \psi_2, \overline{\lambda})}{\mathscr{D}_0(\zeta, \psi_1, \psi_2, \overline{\lambda})} \right), \tag{37}$$

*where $\xrightarrow{p}$ denotes convergence in probability when $d \to \infty$ with respect to the training data $X$, $\varepsilon$, the random features $\Theta$, and the random target function $f_d$, and where*

$$\mathscr{D}_0(\zeta, \psi_1, \psi_2, \overline{\lambda}) = \chi^5 \zeta^6 - 3\chi^4 \zeta^4 + (\psi_1 + \psi_2 - \psi_1 \psi_2 - 1)\chi^3 \zeta^6 + 2\chi^3 \zeta^4 + 3\chi^3 \zeta^2 \tag{38}$$
$$+ (3\psi_1\psi_2 - \psi_2 - \psi_1 - 1)\chi^2 \zeta^4 - 2\chi^2 \zeta^2 - \chi^2 - 3\psi_1\psi_2\chi\zeta^2 + \psi_1\psi_2 \,,$$

$$\mathscr{D}_1(\zeta, \psi_1, \psi_2, \overline{\lambda}) = \chi^6 \zeta^6 - 2\chi^5 \zeta^4 - (\psi_1\psi_2 - \psi_1 - \psi_2 + 1)\chi^4 \zeta^6 + \chi^4 \zeta^4 \tag{39}$$
$$+ \chi^4 \zeta^2 - 2(1 - \psi_1\psi_2)\chi^3 \zeta^4 - (\psi_1 + \psi_2 + \psi_1\psi_2 + 1)\chi^2 \zeta^2 - \chi^2 \,,$$

$$\mathscr{D}_2(\zeta, \psi_1, \psi_2, \overline{\lambda}) = -(\psi_1 - 1)\chi^3 \zeta^4 - \chi^3 \zeta^2 + (\psi_1 + 1)\chi^2 \zeta^2 + \chi^2 \,, \tag{40}$$

*and where $\chi$ is defined in (28).*

**Remark 9.** Eq. (19) in D'Amour et al. (2020) is expressed in terms of the asymptotic limit of $\|a^\star \Theta\|^2$. See Appendix B for the connection between this representation and equation (7) in our definition of $\mathcal{S}$.

## H    DETAILS REGARDING THEOREM 4

$$\mathscr{D}_{0,\text{rless}}(\zeta, \psi_1, \psi_2) = \chi^5 \zeta^6 - 3\chi^4 \zeta^4 + (\psi_1 + \psi_2 - \psi_1 \psi_2 - 1)\chi^3 \zeta^6 + 2\chi^3 \zeta^4 + 3\chi^3 \zeta^2 \tag{41}$$
$$+ (3\psi_1\psi_2 - \psi_2 - \psi_1 - 1)\chi^2 \zeta^4 - 2\chi^2 \zeta^2 - \chi^2 - 3\psi_1\psi_2\chi\zeta^2 + \psi_1\psi_2 \,,$$

$$\mathscr{D}_{1,\text{rless}}(\zeta, \psi_1, \psi_2) = \chi^6 \zeta^6 - 2\chi^5 \zeta^4 - (\psi_1\psi_2 - \psi_1 - \psi_2 + 1)\chi^4 \zeta^6 + \chi^4 \zeta^4 \tag{42}$$
$$+ \chi^4 \zeta^2 - 2(1 - \psi_1\psi_2)\chi^3 \zeta^4 - (\psi_1 + \psi_2 + \psi_1\psi_2 + 1)\chi^2 \zeta^2 - \chi^2 \,,$$

$$\mathscr{D}_{2,\text{rless}}(\zeta, \psi_1, \psi_2) = -(\psi_1 - 1)\chi^3 \zeta^4 - \chi^3 \zeta^2 + (\psi_1 + 1)\chi^2 \zeta^2 + \chi^2 \,, \tag{43}$$

where $\chi$ is defined as in (28), and $\psi \equiv \min\{\psi_1, \psi_2\}$.

## I    DETAILS REGARDING THEOREM 8

The formula for $g(s)$ is as follows:

$$g(s) = \left( e^{s^2/2} \text{erf}^2 \left( s/\sqrt{2} \right) \right) \Big/ \left( e^{s^2/2} \left( 1 - \text{erf}\left( s/\sqrt{2} \right) \right) \left( s^2 + \text{erf}\left( s/\sqrt{2} \right) \right) - \sqrt{2/\pi} s \right). \tag{44}$$

## J    DETAILS REGARDING THEOREM 9

### J.1    RELATIONSHIP AMONG THE CONSTANTS IN TABLE 1

This appendix is referenced in Remark 4.

The following relationships can be derived either from the proof of Theorem 9, or from a direct computation based on the formulas given in Theorem 9. They do not add critical information to what is proved in Theorem 9 but they give general rules to exclude some cells in Table 1.

**Remark 10.** Let $\psi_1 < \psi_2$. If $\psi_2 < \min\{2\psi_1, \psi_1 + 1\}$, then $\alpha_L < \alpha_C < \alpha_R$. In this case, the 4th and 5th rows of Table 1 never happen. If $\psi_2 > \min\{2\psi_1, \psi_1 + 1\}$, then $\alpha_R < \alpha_C < \alpha_L$. If $\psi_2 = \min\{2\psi_1, \psi_1 + 1\}$, then $\alpha_L = \alpha_C = \alpha_R$. We always have that $\alpha_L, \alpha_C, \alpha_R \in [0, 1]$, and $0 \le \beta_3 < \beta_2 < \beta_1$. Since $E_1$ and $E_2$ cannot simultaneously hold, no two rows can simultaneously hold. Since $\beta_3 < \beta_2 < \beta_1$, no two columns can simultaneously hold. Remark 10 follows from Lemma M.13 used in the proof of Theorem 9.

**Remark 11.** Let $\psi_1 > \psi_2$. We always have that $\alpha_L \in [0, 1]$. It also holds that $\alpha_C \in [0, 1]$ if and only if $\psi_1 \leq \psi_2/(1 - \rho|1 - \psi_2|)$ when $0 < \rho \leq \frac{1}{|\psi_2 - 1|}$ and $\psi_1 \geq \psi_2/(1 - \rho|1 - \psi_2|)$ when $\rho \geq \frac{1}{|\psi_2 - 1|}$ . We have that $\alpha_R \in [0, 1]$ if and only if $\psi_1 \leq \psi_2 + \frac{1}{2}\rho(\psi_2 - 1)^2 \min\{\psi_2, 1\}$ and

$$\psi_1 \geq \frac{-1 + \rho(\psi_2 - 1)^2(2\psi_2 + \min\{2\psi_2 - 1, 1\}) + \psi_2(2 + \max\{\psi_2, 2 - \psi_2\})}{2(\max\{\psi_2, 1\} + \rho(\psi_2 - 1)^2)}.$$

Note that if $\psi_1 > \psi_2 = 1$ then $\alpha_R$ is not defined, see statement of Theorem 9. If we had attempted to use the formulas above we would have gotten that $\alpha \in [0, 1]$ if and only if $1 \leq \psi_1 \leq 1$, this last condition never being met when $\psi_1 > \psi_2 = 1$. Also, $A \leq B$ always, where $A$ and $B$ are given in (67) and (68). Furthermore, $B < \psi_2$ if and only if $1/\rho > \gamma \triangleq \min\{1, \max\{0, 2\psi_2 - 1\}\}$. Since $A \leq B$, it follows that $E_1$ and $E_2$ cannot simultaneously hold, and hence no two rows can simultaneously hold. We always have that $\beta_3 < \beta_2 < \beta_1$, therefore no two columns can simultaneously hold. Remark 11 follows from Lemma M.14 used in the proof of Theorem 9.

### J.2 STATEMENT OF THEOREM 9 WHEN $\psi_1 < \psi_2$

**Theorem (9 continued).** *If $\psi_1 < \psi_2$ then $E_1 = (\alpha < \alpha_R)$, $E_2 = (\alpha > \alpha_R)$,*

$$\beta_1 = \min\{\psi_1 - 4, -3\psi_1\} - 8\sqrt{|1 - \psi_1|} \max\{1/\psi_1, \psi_1^{3/2}\} + 8 \max\{1/\psi_1, \psi_1^2\}, \tag{45}$$

$$\beta_2 = \psi_1(\psi_1 + 2)/(\psi_1 + 1), \tag{46} \qquad \beta_3 = \psi_1 + |1 - \psi_1|\min\{\psi_1, 1/\psi_1\}, \tag{47}$$

$$\alpha_L = \psi_2/(\psi_2 + 1 + \rho^{-1}), \tag{48} \qquad \alpha_C = \psi_2/(2\psi_2 + 1 - \psi_1 + \max\{0, 1 - \psi_1\} + \rho^{-1}), \tag{49}$$

$$\alpha_R = \psi_2/(3\psi_2 + 1 - 2\min\{1 + \psi_1, 2\psi_1\} + \rho^{-1}). \tag{50}$$

*Furthermore, the polynomial $p(x)$ is defined as follows.*

*If $\psi_1 \neq 1$ then $p(x) \triangleq p_0 + p_1 x + p_2 x^2 + p_3 x^3 + p_4 x^4 + p_5 x^5$ where*

$$p_0 = -(\psi_1 - 1)^2\psi_1^2 \left(\rho(\alpha - 4\alpha\psi_1 + (3\alpha - 1)\psi_2) + \alpha\right), \tag{51}$$

$$p_1 = 2(\psi_1 - 1)\psi_1 \left(\rho(\alpha(\psi_1(9\psi_1 - 6\psi_2 - 4) + \psi_2) + 2\psi_1\psi_2) - 2\alpha\psi_1\right), \tag{52}$$

$$p_2 = 2\psi_1 \left(\rho(\alpha + 4\alpha\psi_1(4\psi_1 - 3) + \psi_2(4\alpha - 9\alpha\psi_1 + 3\psi_1 - 1)) - \alpha(3\psi_1 - 1)\right), \tag{53}$$

$$p_3 = 4\psi_1 \left(\rho(\alpha(7\psi_1 - 2) - 3\alpha\psi_2 + \psi_2) - \alpha\right), \tag{54}$$

$$p_4 = \rho(\alpha(12\psi_1 - 1) - 3\alpha\psi_2 + \psi_2) - \alpha, \tag{55}$$

$$p_5 = 2\alpha\rho, \tag{56}$$

*If $\psi_1 = 1$ then $p(x) \triangleq q_0 + q_1 x + q_2 x^2 + q_3 x^3$ where*

$$q_0 = \rho(-10\alpha + 10\alpha\psi_2 - 4\psi_2) + 4\alpha, \tag{57} \qquad q_1 = \rho(-20\alpha + 12\alpha\psi_2 - 4\psi_2) + 4\alpha, \tag{59}$$

$$q_2 = \rho(-11\alpha + 3\alpha\psi_2 - \psi_2) + \alpha, \tag{58} \qquad \textit{and } q_3 = -2\alpha\rho. \tag{60}$$

### J.3 STATEMENT OF THEOREM 9 WHEN $\psi_1 > \psi_2$

**Theorem** (9 continued). *If* $\psi_1 > \psi_2$ *and* $\psi_2 \neq 1$ *then* $E_1 = (\psi_1 < B) \vee ((\alpha < \alpha_R) \wedge (B < \psi_1 < A))$, $E_2 = (\psi_1 > A) \vee ((\alpha > \alpha_R) \wedge (B < \psi_1 < A))$,

$$\beta_1 : r(\beta_1) = 0, \text{ where } r \text{ is a 4th degree polynomial described in Appendix J.4,} \tag{61}$$

$$\beta_2 = \psi_2 + (\alpha\psi_2/(\psi_2 + 1 + \rho^{-1})), \tag{62}$$

$$\beta_3 = \psi_2 + (\min\{\psi_2, 1/\psi_2\}\alpha|\psi_2 - 1|^3/((\psi_2 - 1)^2 + \rho^{-1}(\psi_2 + 3))), \tag{63}$$

$$\alpha_L = (2\psi_1 - \psi_2)/(2\psi_1 - \psi_2 + 1 + \rho^{-1}), \tag{64}$$

$$\alpha_C = (\rho\psi_1 - (\psi_1 - \psi_2)/|1 - \psi_2|)/(\rho(\max\{0, 1 - \psi_2\} + 2\psi_1 - \psi_2) + 1), \tag{65}$$

$$\alpha_R = \frac{2(\psi_1 - \psi_2)\max\{1, \psi_2\} - \rho(\psi_2 - 1)^2 \psi_2}{(\psi_2 - 1)^2(\rho(2\min\{1, \psi_2\} - 2\psi_1 + \psi_2) - \rho - 1)}, \tag{66}$$

$$A = \psi_2 + (\rho\min\{1, \psi_2\}(\psi_2 - 1)^2/2), \text{ and} \tag{67}$$

$$B = \frac{\rho(\psi_2 - 1)^2(2\psi_2 + 1 + 2\min\{\psi_2 - 1, 0\}) + 2\psi_2 - 1 + 2\psi_2\max\{\psi_2, 1\} - \psi_2^2}{2\rho(\psi_2 - 1)^2 + 2\max\{1, \psi_2\}}. \tag{68}$$

*If* $\psi_1 > \psi_2$ *and* $\psi_2 = 1$ *then* $E_1 = $ *False,* $E_2 = $ *True and eqs.* (61)-(68) *hold when reading Table 1, except* (66), *which is not defined, and* (65) *which becomes* $\alpha_C = -\infty$ *implying* $(\alpha > \alpha_C)$ *is True.*

*Furthermore, the polynomial* $p(x)$ *is defined as follows.*

$$p(x) \triangleq p_0 + p_1 x + p_2 x^2 + p_3 x^3 + p_4 x^4 + p_5 x^5, \text{ where} \tag{69}$$

$$p_0 = \psi_2^2 \left( \rho(\psi_2 - 1)^2(2\alpha\psi_1 - (3\alpha + 1)\psi_2 + \alpha) + \left( \alpha\psi_2^2 - 2(\alpha + 1)\psi_2 + \alpha + 2\psi_1 \right) \right), \tag{70}$$

$$p_1 = 2\psi_2 \left( (\psi_2(2\alpha(\psi_2 - 1) - 1) + \psi_1) - \rho(\psi_2 - 1) \left( (7\alpha + 2)\psi_2^2 - \psi_2(4\alpha\psi_1 + 3\alpha + 1) + \psi_1 \right) \right), \tag{71}$$

$$p_2 = 2\psi_2 \left( \alpha(3\psi_2 - 1) - \rho \left( (13\alpha + 3)\psi_2^2 - 2\psi_2(3\alpha\psi_1 + 5\alpha + 1) + 2\alpha\psi_1 + \alpha + \psi_1 \right) \right), \tag{72}$$

$$p_3 = 4\psi_2 \left( \rho(2\alpha(\psi_1 + 1) - (6\alpha + 1)\psi_2) + \alpha \right), \tag{73}$$

$$p_4 = \rho(2\alpha\psi_1 - (11\alpha + 1)\psi_2 + \alpha) + \alpha, \tag{74}$$

$$p_5 = -2\alpha\rho. \tag{75}$$

### J.4 DEFINITION OF THE POLYNOMIAL $r(x)$ WHEN $\psi_1 > \psi_2$

This appendix is referenced in the statement of Theorem 9 in equation (61).

The coefficient $\beta_1$ is such that $r(\beta_1) = 0$, where $r(x) = r_0 + r_1 x + r_2 x^2 + r_3 x^3 + r_4 x^4$ with

$$\begin{aligned} r_0 = {} & \rho\psi_2^3(\rho\psi_2(\rho\psi_2(16\alpha\rho\psi_2 - 8\alpha(14\alpha\rho + \rho - 6) + \rho) + 8\alpha(\rho((8\alpha(2\alpha + 5) - 1)\rho - 28\alpha + 5) + 6)) \\ & + 16\alpha(\alpha\rho + \rho + 1)(-4\alpha\rho + \rho + 1)^2), \end{aligned}$$

$$\begin{aligned} r_1 = {} & 4\rho\psi_2^2(\rho\psi_2(2\alpha(8\alpha\rho(7 - 2(\alpha + 5)\rho) + 3(\rho - 5)\rho - 18) - \rho\psi_2(-56\alpha^2\rho + 12\alpha\rho\psi_2 - 6\alpha(\rho - 6) + \rho)) \\ & - 4\alpha(\rho + 1)((2\alpha - 3)\rho - 3)((4\alpha - 1)\rho - 1)), \end{aligned}$$

$$r_2 = 2\rho\psi_2(-8\alpha^2\rho(\rho\psi_2(7\rho\psi_2 - 20\rho + 14) + 7(\rho + 1)^2) + 12\alpha(\rho\psi_2 + \rho + 1)(\rho\psi_2) + 2(\rho + 1)^2) + 3\rho^3\psi_2^2),$$

$$r_3 = -8\alpha\rho(\rho\psi_2 + \rho + 1)(\rho\psi_2(2\rho\psi_2 - 3\rho + 4) + 2(\rho + 1)^2) - 4\rho^4\psi_2^2, \text{ and}$$

$$r_4 = \rho^4\psi_2.$$

## K    DETAILS REGARDING THEOREM 10

The coefficients of $p(x)$ are given below

$$p_0 = 8\psi_2\rho^{-1}+(\alpha+4\psi_2(-1+2\psi_2)(-1+2\alpha)), \quad (76) \quad p_1 = 8\psi_2\rho^{-1}+4((1-4\psi_2)\alpha+2\psi_2^2), \quad (77)$$

$$p_2 = -2(-3\alpha + \psi_2(2+4\alpha)), \quad (78) \quad p_3 = 4\alpha, \quad (79) \text{ and } \quad p_4 = \alpha. \quad (80)$$

## L    IMPORTANT OBSERVATIONS

This appendix is referenced in Section 3.3.

### L.1    REPRODUCING FIGURE 1

The plots in Figure 1 come directly from our theory. They involve no experiments and can be obtained using many standard mathematical computing software tools. Nonetheless, since it does require some effort to code our equations, we include code to generate Figure 1 in the following Github link: `https://github.com/Jeffwang87/RFR_AF`. This code is also available in the supplementary zip file provided. To generate the plots in Figure 1, run the file named `RunMeToGenerateFigure_1.nb`. It runs using Wolfram Mathematica V12. We ran it using a MacBook Pro with 2.6 GHz 6-Core Intel Core i7 and 32 GB 2667 MHz DDR4. In this machine it takes about 1 second to run.

### L.2    SOME DETAILS ABOUT OBSERVATION 1 IN SECTION 3.3

When $\alpha = 0$ and $\psi_1 < \psi_2$, and using Theorem 9, we can explicitly compute the objective $L = \mathcal{E}_{R_1} = (F_\star^2\psi_2 + F_1^2\max\{1-\psi_1,0\}\psi_2 + \psi_1\tau^2)/(\psi_2-\psi_1)$ if $\psi_1 < \psi_2$. This follows from making $\alpha = 0$ in the expressions in the paragraph before Theorem 10. Observe that, as function of $\psi_1$, the test error $\mathcal{E}_{R_1}$ achieves a minimum of $(F_\star^2\psi_2 + \tau^2)/(-1+\psi_2)$ at $\psi_1 = 1$ if $\psi_2 > 1 + 1/\rho$ and a minimum of $(F_1^2\psi_2 + F_\star^2\psi_2)/\psi_2$ at $\psi_1 = 0$ otherwise. In particular, if $\tau = F_\star = 0^+$, then the condition $\psi_2 > 1 + 1/\rho$ becomes $\psi_2 > 1$.

When $\alpha = 0$ and $\psi_1 > \psi_2$ the polynomial that determines $x_1$ becomes

$$p(x) = \psi_2\left(-4\psi_2x^3 - 2(\psi_1+\psi_2(3\psi_2-2))x^2 - x^4 - 2(\psi_2-1)(\psi_1+\psi_2(2\psi_2-1))x - (\psi_2-1)^2\psi_2^2\right). \quad (81)$$

One can show that in the range $\psi_1 > \psi_2$ and $x \in (x_L, x_R) = (-\psi_2, \min\{0, 1-\psi_2\})$, we have $p(x) < 0$ and hence it has no roots. Therefore, $x_1$ does not exist. Therefore, if furthermore we have $\tau = F_\star = 0$, then condition $E_2$ is never true, so $x$ must be read from the first row in Table 12, which implies that $x = x_R$, and the optimal AF is linear also when $\psi_1 > \psi_2$.

## M    PROOFS

This appendix is referenced in Section 3.

### M.1    PROOF OF SENSITIVITY PROPERTIES OF SECTION 2.2

*Proof of Theorem 4.* The proof follows directly from Theorem 13 by taking the limit when $\lambda \to +0$. □

*Proof of Theorem 5.* The proof follows directly from Theorem 13 by taking the limit when $\psi_1 \to +\infty$. □

*Proof of Theorem 6.* The proof follows directly from Theorem 13 by taking the limit when $\psi_2 \to +\infty$. □

## M.2    Proof of necessary and sufficient condition for linearity of Section 3.1

*Proof of Lemma 3.1 .* If $\sigma(x)$ is linear function, a direct calculation shows that $\mu_\star = 0$. On the other hand, since $0 \leq \mathbb{E}(\sigma(Z) - \mu_0 - \mu_1 Z)^2 = \mu_\star^2$, we have that $\mu_\star = 0$ implies that $\sigma(Z) = \mu_0 + \mu_1 Z$ almost surely. Since the support of the probability density function of $Z$ is $\mathbb{R}$, it follows that $\sigma(x) = \mu_0 + \mu_1 x$ except on a set of measure zero in $\mathbb{R}$. $\qquad\square$

## M.3    Proof of Theorem 7

To prove Theorem 7, we first need to state and prove a series of intermediary results.

**Lemma M.1.** *A necessary condition for optimally of $\sigma(x)$ is that*

$$-2x\sigma'(x) + 2\sigma''(x) + \lambda_1 + \lambda_2 x + \lambda_3 \sigma(x) = 0, \tag{82}$$

*where $\lambda_1$, $\lambda_2$, and $\lambda_3$ satisfy the following constraints,*

$$\lambda_1 + \lambda_3 \mu_0 = 0, \tag{83}$$

$$-2\mu_1 + \lambda_2 + \lambda_3 \mu_1 = 0, \tag{84}$$

$$-2\mathbb{E}((\sigma'(Z))^2) + \lambda_1 \mu_0 + \lambda_2 \mu_1 + \lambda_3 \mu_2 = 0, \tag{85}$$

*and that*

$$\lim_{x \to +\infty} (\sigma'(x))^2 e^{\frac{-x^2}{2}} = \lim_{x \to -\infty} (\sigma'(x))^2 e^{\frac{-x^2}{2}} = 0. \tag{86}$$

**Remark 12.** Note that since (82) is a second order ODE, its solutions are parametrized by two constants in addition to being parametrized by $\lambda_1, \lambda_2, \lambda_3$. These five constants are set by our three constraints (19) together with two boundary conditions from our variational problem, which are $\lim_{x \to \pm\infty} \sigma'(x) e^{\frac{-x^2}{2}} = 0$. These last two we replace (see proof) by (86).

**Remark 13.** The lemma implies that knowing $\mu_0, \mu_1, \mu_2$ and the objective value $\mathbb{E}((\sigma'(Z))^2)$ is enough to determine $\lambda_1, \lambda_2, \lambda_3$, even without solving the ODE.

*Proof of Lemma M.1.*
*Derivation of equation* (82): A Lagragian for (19)-(19) is

$$\mathbb{E}((\sigma'(Z))^2 + \lambda_1(\mu_0 - \sigma(Z)) + \lambda_2(\mu_1 - Z\sigma(Z)) + \lambda_3((1/2)(\mu_2 - \sigma(Z))^2)), \tag{87}$$

which can be written as $\int_{-\infty}^{\infty} L(z, \sigma, \sigma')\mathrm{d}z$ where, if we define $p(z) = \frac{1}{\sqrt{2\pi}}e^{\frac{-z^2}{2}}$, the Lagrangian density $L$ is

$$L(z, \sigma, \sigma') = p(z)(\sigma'^2 + \lambda_1(\mu_0 - \sigma) + \lambda_2(\mu_1 - z\sigma) + \lambda_3(1/2)(\mu_2 - \sigma^2)). \tag{88}$$

We use the Euler-Lagrange equation with free boundary conditions Gelfand et al. (2000) to get the necessary condition (82). To do so, we compute in sequence

$$\frac{\partial L}{\partial \sigma} = -p(z)(\lambda_1 + \lambda_2 z + \lambda_3 \sigma), \tag{89}$$

$$\frac{\partial L}{\partial \sigma'} = p(z)(2\sigma'), \tag{90}$$

$$\frac{\mathrm{d}}{\mathrm{d}z}\frac{\partial L}{\partial \sigma'} = p'(z)(2\sigma') + p(z)(2\sigma'') = p(z)(-2z\sigma' + 2\sigma''). \tag{91}$$

These lead to

$$0 = \frac{\mathrm{d}}{\mathrm{d}z}\frac{\partial L}{\partial \sigma'} - \frac{\partial L}{\partial \sigma} = p(z)(-2z\sigma' + 2\sigma'' + \lambda_1 + \lambda_2 z + \lambda_3 \sigma) \text{ and} \tag{92}$$

$$0 = \lim_{z \to \pm\infty} \frac{\partial L}{\partial \sigma'} = \lim_{z \to \pm\infty} p(z)\sigma'(z), \tag{93}$$

where the last condition follows from the fact that there is no boundary condition on $\sigma$. Since $p(z) > 0$, equation (92) implies (82).

*Derivation of equations* (86): Since we are working with necessary conditions, we can choose not list (93) in our lemma. Rather, we include condition (86), which a consequence of the fact that $\mathbb{E}((\sigma'(Z))^2)$ must be finite. Indeed, $\mathbb{E}((\sigma'(Z))^2) = \int_{-\infty}^{\infty} p(z)(\sigma'(z))^2 < \infty$ implies that $p(z)(\sigma'(z))^2$ must vanish at $\pm\infty$. Although not necessary for this proof, note that, since $p(z)$ goes to zero as $z \to \pm\infty$, equations (86) imply equations (93).

*Derivation of equations* (83)-(85):

Equation (82) must hold for all $x$. Hence, we can replace in it $x$ by a standard normal random variable $Z$ and compute the expected value of both of its sides. This leads to

$$-2\,\mathbb{E}(Z\sigma'(Z)) + 2\,\mathbb{E}(\sigma''(Z)) + \lambda_1 + \lambda_3\mu_0 = 0. \tag{94}$$

We can also multiply (82) by $x$, replace $x$ by a standard normal random variable $Z$ and compute the expected value of both of its sides. This leads to,

$$-2\,\mathbb{E}(Z^2\sigma'(Z)) + 2\,\mathbb{E}(Z\sigma''(Z)) + \lambda_2 + \lambda_3\mu_1 = 0. \tag{95}$$

Finally, we multiply (82) by $\sigma(x)$, replace $x$ by a standard normal random variable $Z$ and compute the expected value of both of its sides. This leads to,

$$-2\,\mathbb{E}(Z\sigma(Z)\sigma'(Z)) + 2\,\mathbb{E}(\sigma(Z)\sigma''(Z)) + \lambda_1\mu_0 + \lambda_2\mu_1 + \lambda_3\mu_2 = 0. \tag{96}$$

Using integration by parts, and the fact that for $p(z) = \frac{1}{\sqrt{2\pi}}e^{\frac{-z^2}{2}}$ we have that $p'(z) = -zp(z)$ and $p''(z) = (1 - z^2)p(z)$, we derive the following useful relationships

$$\mathbb{E}(\sigma''(Z)) = \mathbb{E}(Z\sigma'(Z)), \tag{97}$$

$$\mathbb{E}(Z\sigma''(Z)) = \mathbb{E}(Z^2\sigma'(Z)) - \mathbb{E}(\sigma'(Z)), \tag{98}$$

$$\mathbb{E}(\sigma'(Z)) = \mu_1, \tag{99}$$

$$\mathbb{E}(\sigma(Z)\sigma''(Z)) = \mathbb{E}(Z\sigma(Z)\sigma'(Z)) - \mathbb{E}((\sigma'(Z))^2). \tag{100}$$

To derive these relationships via integration by parts we made use of the following relationships

$$\lim_{z\to\pm\infty} \sigma'(z)\sigma(z)p(z) = \lim_{z\to\pm\infty} \sigma'(z)zp(z) = \lim_{z\to\pm\infty} \sigma'(z)p(z) = \lim_{z\to\pm\infty} \sigma(z)p(z) = 0, \tag{101}$$

which can be proved from $\mathbb{E}((\sigma(Z))^2), \mathbb{E}((\sigma'(Z))^2) < \infty$. Also, from $\mathbb{E}((\sigma(Z))^2), \mathbb{E}((\sigma'(Z))^2) < \infty$ and $\mu_1 \in \mathbb{R}$, we can show that each of the expected values in the right-hand-side of (97)-(100) is well-defined. Hence, the left-hand-side of (97)-(100) is well defined.

To finish the proof we replace (97) in (94) to get $\lambda_1 + \lambda_3\mu_0 = 0$, which is equation (83). Then we replace (98) and (99) in (95) to get $-2\mu_1 + \lambda_2 + \lambda_3\mu_1 = 0$, which is equation (84). Lastly, we replace (100) in (96) to get $-2\,\mathbb{E}((\sigma'(Z))^2) + \lambda_1\mu_0 + \lambda_2\mu_1 + \lambda_3\mu_2 = 0$, which is equation (85). $\qquad\square$

**Lemma M.2.** *The solutions of* (82) *are of the form*

$$\sigma(x) = \bar{\sigma}(x) + c_1 H_{\frac{\lambda_3}{2}}\left(\frac{x}{\sqrt{2}}\right) + c_2 F_{\{\frac{-\lambda_3}{4}\},\{\frac{1}{2}\}}\left(\frac{x^2}{2}\right), \tag{102}$$

*where*

$$\bar{\sigma}(x) = \frac{\lambda_2 x}{2} - \frac{\lambda_1 x^2}{4}F_{\{1,1\},\{\frac{3}{2},2\}}\left(\frac{x^2}{2}\right), \text{ if } \lambda_3 = 0 \tag{103}$$

$$\bar{\sigma}(x) = -\frac{\lambda_1}{2} - \frac{\lambda_2 x}{2} + \frac{1}{2}\sqrt{\frac{\pi}{2}}\lambda_2 e^{\frac{x^2}{2}} erf\left(\frac{x}{\sqrt{2}}\right) - \frac{1}{4}\lambda_2 x^3 F_{\{1,1\},\{\frac{3}{2},2\}}\left(\frac{x^2}{2}\right), \text{ if } \lambda_3 = 2 \tag{104}$$

$$\bar{\sigma}(x) = -\frac{\lambda_1}{\lambda_3} - \frac{\lambda_2 x}{\lambda_3 - 2}, \text{ if } \lambda_3 \notin \{0, 2\} \tag{105}$$

*where erf is the Gauss error function , $F_{\{a_k\},\{b_k\}}$ is the generalized hypergeometric function with parameters $\{a_k\}, \{b_k\}$, and $H_n$ is the Hermite polynomial, extended to a possibly non-integer n.*

**Remark 14.** By an extension of $H_n(x)$ to non-integer we mean that $H_n(x) = F_{\{-\frac{1}{2}(n-1)\},\{\frac{3}{2}\}}(x^2)$, which is defined for non-integer $n$.

*Proof of Lemma M.2.* Since (82) is a second order ODE, its solutions are spanned by any particular solution $\bar{\sigma}(x)$ plus a linear combination of two solutions to its associated homogeneous ODE. The homogenous ODE associate with (82) is $-2x\sigma'(x) + 2\sigma''(x) + \lambda_3\sigma(x) = 0$. The change of variable $\sigma(x) = \tilde{\sigma}(x/\sqrt{2})$ allows us to get $-2x\tilde{\sigma}'(x) + \tilde{\sigma}''(x) + \lambda_3\tilde{\sigma}(x) = 0$, which is a well-known ODE called the *Hermite differential equation*. This implies that the homogeneous solutions of our ODE are spanned by $H_{\frac{\lambda_3}{2}}\left(\frac{x}{\sqrt{2}}\right)$ and $F_{\{\frac{-\lambda_3}{4}\},\{\frac{1}{2}\}}\left(\frac{x^2}{2}\right)$. The particular solutions $\bar{\sigma}(x)$ can be confirmed by direct substitution. $\square$

**Lemma M.3.** *Let $\sigma$ be of the form (102). If $\lambda_3 = 0$, then $\mathbb{E}((\sigma'(Z))^2) < \infty$ implies that $\sigma(x)$ is of the form*

$$\sigma(x) = c + \frac{x\lambda_2}{2}, \tag{106}$$

*for some $c$, and that $\mathbb{E}((\sigma'(Z))^2) = \frac{\lambda_2^2}{4}$.*

*Proof of Lemma M.3.* If $\lambda_3 = 0$ then based on Lemma M.2 we can re-write that

$$\sigma(x) = \sqrt{\frac{\pi}{2}}c_1\text{erfi}\left(\frac{x}{\sqrt{2}}\right) + c_2 - \frac{1}{4}x\left(\lambda_1 xF\left(\frac{x^2}{2}\right) - 2\lambda_2\right), \tag{107}$$

for some $c_1, c_2$, where erfi is the inverse of erf. From this it follows that,

$$\sigma'(x) = \frac{1}{4}\left(e^{\frac{x^2}{2}}\left(4c_1 - \sqrt{2\pi}\lambda_1\text{erf}\left(\frac{x}{\sqrt{2}}\right)\right) + 2\lambda_2\right). \tag{108}$$

The objective $\mathbb{E}((\sigma'(Z))^2) < \infty$ implies that $\lim_{x\to\pm\infty}(\sigma'(x))^2 e^{-\frac{x^2}{2}} = 0$. Since $\lim_{x\to\pm\infty}\text{erf}(x) = \pm 1$, we have that $\lim_{x\to\pm\infty}(\sigma'(x))^2 e^{-\frac{x^2}{2}} = 0$ implies that $4c_1 = \pm\sqrt{2\pi}\lambda_1$ for both $\pm$ simultaneously. This implies that $c_1 = \lambda_1 = 0$.

Substituting $c_1 = \lambda_1 = 0$ in (107) and simplifying we get $\sigma(x) = c_2 + \frac{\lambda_2 x}{2}$. From this expression one can then directly compute $\mathbb{E}((\sigma'(Z))^2)$. $\square$

**Lemma M.4.** *Let $\sigma$ be of the form (102). If $\lambda_3 = 2$, then $\mathbb{E}((\sigma'(Z))^2) < \infty$ implies that $\sigma(x)$ is of the form*

$$\sigma(x) = cx - \frac{\lambda_1}{2}, \tag{109}$$

*for some $c$, and that $\mathbb{E}((\sigma'(Z))^2) = c^2$.*

*Proof of Lemma M.4.* If $\lambda_3 = 2$ then based on Lemma M.2 we can re-write that

$$\sigma(x) = \frac{1}{4}\Bigg(-x\Big(\lambda_2\Big(x^2 F_{\{1,1\},\{\frac{3}{2},2\}}\left(\frac{x^2}{2}\right) + 2\Big)$$

$$+ 2\sqrt{2}\left(\sqrt{\pi}c_2\text{erfi}\left(\frac{x}{\sqrt{2}}\right) - 2c_1\right)\Big) + e^{\frac{x^2}{2}}\left(\sqrt{2\pi}\lambda_2\text{erf}\left(\frac{x}{\sqrt{2}}\right) + 4c_2\right) - 2\lambda_1\Bigg) \tag{110}$$

for some $c_1$ and $c_2$, where erfi is the inverse of erf. From this it follows that

$$\sigma'(x) = \frac{2c_1 - \sqrt{\pi}c_2\text{erfi}\left(\frac{x}{\sqrt{2}}\right)}{\sqrt{2}} - \frac{1}{4}\lambda_2 x^2 F_{\{1,1\},\{\frac{3}{2},2\}}\left(\frac{x^2}{2}\right). \tag{111}$$

The objective $\mathbb{E}((\sigma'(Z))^2) < \infty$ implies that $\lim_{x\to\pm\infty}(\sigma'(x))^2 e^{-\frac{x^2}{2}} = 0$. Since $\lim_{x\to\pm\infty}\left(\text{erfi}\left(\frac{x}{\sqrt{2}}\right)\right)^2 = \lim_{x\to\pm\infty}\left(x^2 F_{\{1,1\},\{\frac{3}{2},2\}}\left(\frac{x^2}{2}\right)\right)^2 = \infty$, and since

$$\lim_{x\to\pm\infty}\frac{x^2 F_{\{1,1\},\{\frac{3}{2},2\}}\left(\frac{x^2}{2}\right)}{\text{erfi}\left(\frac{x}{\sqrt{2}}\right)} = \pm\pi, \tag{112}$$

it follows we only have a finite objective if $\frac{-\sqrt{\pi}c_2}{\sqrt{2}} = \pm\frac{\lambda_2\pi}{4}$ for both $\pm$ simultaneously. But this implies that $c_2 = \lambda_2 = 0$.

Substituting $c_2 = \lambda_2 = 0$ in (110), and simplifying, leads to $\sigma(x) = c_1\sqrt{2}x - \frac{\lambda_1}{2}$. The $\sqrt{2}$ factor can be absorbed by $c_1$. From this expression one can compute $\mathbb{E}((\sigma'(Z))^2)$. $\qquad\square$

**Lemma M.5.** *Let $\sigma$ be of the form* (102). *If $\lambda_3 \notin \{0, 2\}$, $c_1 = c_2 = 0$ then*

$$\sigma(x) = -\frac{\lambda_1}{\lambda_3} - \frac{\lambda_2 x}{\lambda_3 - 2}, \tag{113}$$

*and*

$$\mathbb{E}((\sigma'(Z))^2) = \left(\frac{\lambda_2}{(\lambda_3 - 2)}\right)^2. \tag{114}$$

*Proof of Lemma M.5.* This follows directly from Lemma M.2. $\qquad\square$

**Lemma M.6.** *Let $\sigma$ be of the form* (102). *If $\lambda_3 \notin \{0, 2\}$, and either $c_1$ or $c_2$ are non-zero, then $\mathbb{E}((\sigma'(Z))^2) < \infty$ implies that $\lambda_3 = 4k$ for $k \in \mathbb{Z}^+$.*

*Proof of Lemma M.6.* If $\lambda_3 \notin \{0, 2\}$, then by Lemma M.2 we have a formula for $\sigma(x)$ from which we get that

$$\sigma'(x) = \frac{c_1\lambda_3 H_{\frac{\lambda_3}{2}-1}\left(\frac{x}{\sqrt{2}}\right)}{\sqrt{2}} - \frac{1}{2}c_2\lambda_3 x F_{\{1-\frac{\lambda_3}{4}\},\{\frac{3}{2}\}}\left(\frac{x^2}{2}\right) - \frac{\lambda_2}{\lambda_3 - 2}, \tag{115}$$

where $F$ and $H$ are as defined in Lemma M.2.

The boundeness of $\mathbb{E}((\sigma'(Z))^2)$ is dependent on how fast $H_{\frac{\lambda_3}{2}-1}\left(\frac{x}{\sqrt{2}}\right)$ and $xF_{\{1-\frac{\lambda_3}{4}\},\{\frac{3}{2}\}}\left(\frac{x^2}{2}\right)$ grow as $x \to \pm\infty$.

If $c_1 = 0$ and $c_2 \neq 0$, $\mathbb{E}((\sigma'(Z))^2) < \infty$ only if $\mathbb{E}\left(\left(xF_{\{1-\frac{\lambda_3}{4}\},\{\frac{3}{2}\}}\left(\frac{x^2}{2}\right)\right)^2\right) < \infty$, which in turn is true only if $\lim_{x\to\infty}\left(xF_{\{1-\frac{\lambda_3}{4}\},\{\frac{3}{2}\}}\left(\frac{x^2}{2}\right)\right)^2 e^{\frac{-x^2}{2}} = 0$. This implies that $\lambda_3$ is even.

If $c_1 \neq 0$ and $c_2 = 0$, $\mathbb{E}((\sigma'(Z))^2) < \infty$ only if $\mathbb{E}\left(\left(H_{\frac{\lambda_3}{2}-1}\left(\frac{x}{\sqrt{2}}\right)\right)^2\right) < \infty$, which in turn is true only if $\lim_{x\to-\infty}\left(H_{\frac{\lambda_3}{2}-1}\left(\frac{x}{\sqrt{2}}\right)\right)^2 e^{\frac{-x^2}{2}} = 0$. This implies that $\lambda_3$ is even.

If $c_1, c_2 \neq 0$, $\mathbb{E}((\sigma'(Z))^2) < \infty$ only if $\lim_{x\to\infty}(\sigma'(x))^2 e^{\frac{-x^2}{2}} = 0$. But $xF_{\{1-\frac{\lambda_3}{4}\},\{\frac{3}{2}\}}\left(\frac{x^2}{2}\right)$ grows much faster than $H_{\frac{\lambda_3}{2}-1}\left(\frac{x}{\sqrt{2}}\right)$ as $x \to \infty$, hence it must be that $\lim_{x\to\infty}\left(xF_{\{1-\frac{\lambda_3}{4}\},\{\frac{3}{2}\}}\left(\frac{x^2}{2}\right)\right)^2 e^{\frac{-x^2}{2}} = 0$. This implies that $\lambda_3$ is even. $\qquad\square$

**Lemma M.7.** *Let $\lambda_3 = 4k$ for $k \in \mathbb{Z}^+$, and let $\sigma(x)$ be a solution of* (82), *then*

$$\sigma(x) = -\frac{\lambda_1}{\lambda_3} - \frac{\lambda_2 x}{\lambda_3 - 2} + cH_{2k}\left(\frac{x}{\sqrt{2}}\right) \tag{116}$$

*for some $c$ and*

$$\mathbb{E}((\sigma'(Z))^2) = \frac{4^k((2k)!)^2}{(2k-1)!}c^2 + \left(\frac{\lambda_2}{\lambda_3 - 2}\right)^2. \tag{117}$$

*Proof of Lemma M.7.* If $\lambda_3 = 4k$ then

$$F_{\{\frac{-\lambda_3}{4}\},\{\frac{1}{2}\}}\left(x^2/2\right) = \frac{(-1)^k k!}{(2k)!}H_{\lambda_3/2}(x/\sqrt{2}), \tag{118}$$

and hence the homogenous part of $\sigma(x)$ can be written as $cH_{\lambda_3/2}(x)$, from which (116) follows. From this expression it follows that

$$\sigma'(x) = \frac{c\lambda_3 H_{\frac{\lambda_3}{2}-1}\left(\frac{x}{\sqrt{2}}\right)}{\sqrt{2}} - \frac{\lambda_2}{\lambda_3 - 2}, \tag{119}$$

and from this expression the value for $\mathbb{E}((\sigma'(Z))^2)$ follows. $\qquad\square$

**Lemma M.8.** *Let $\sigma$ be of the form* (102). *If $\lambda_3 \notin \{0, 2\}$, then $\mathbb{E}((\sigma'(Z))^2) < \infty$ implies that $\sigma(x)$ is of the form*

$$\sigma(x) = -\frac{\lambda_1}{\lambda_3} - \frac{\lambda_2 x}{\lambda_3 - 2} + cH_{\lambda_3/2}\left(\frac{x}{\sqrt{2}}\right) \tag{120}$$

*for some $c$ that must be zero if $\lambda_3 \neq 4k$, $k \in \mathbb{Z}^+$, and*

$$\mathbb{E}((\sigma'(Z))^2) = \frac{4^k((2k)!)^2}{(2k-1)!}c^2 + \left(\frac{\lambda_2}{\lambda_3 - 2}\right)^2. \tag{121}$$

*Proof of Lemma M.8.* This follows directly from Lemmas M.5-M.7. $\qquad\square$

**Lemma M.9.** *If $\lambda_1, \lambda_2, \lambda_3$ satisfy* (83)-(85), *then*

$$\left(\frac{\lambda_2}{\lambda_3 - 2}\right)^2 = \mu_1^2. \tag{122}$$

*Proof of Lemma M.9.* Defining $R = \mathbb{E}((\sigma'(Z))^2)$ and solving the linear system (83)-(85) leads to

$$\lambda_1 = \frac{2\mu_0\left(R - \mu_1^2\right)}{\mu_0^2 + \mu_1^2 - \mu_2}, \lambda_2 = \frac{2\mu_1\left(\mu_0^2 - \mu_2 + R\right)}{\mu_0^2 + \mu_1^2 - \mu_2}, \lambda_3 = -\frac{2\left(R - \mu_1^2\right)}{\mu_0^2 + \mu_1^2 - \mu_2}. \tag{123}$$

Computing $\lambda_2/(\lambda_3 - 2)$ we get $-\mu_1$, from which the first relation follows.

If we solve $\lambda_3 = -\frac{2(R - \mu_1^2)}{\mu_0^2 + \mu_1^2 - \mu_2}$ for $R$ and recall that $\mu_\star^2 = \mu_2 - \mu_0^2 - \mu_1^2$, the second relation follows.

$\qquad\square$

We are now ready to prove Theorem 7, which we restate here for convenience.

**Theorem ( 7).** *The minimizers of* (19) *are*

$$\sigma(x) = ax^2 + bx + c, \tag{124}$$

*where*

$$a = \pm\frac{\mu_\star}{\sqrt{2}}, b = \mu_1, \text{ and } c = \mu_0 - a. \tag{125}$$

*Proof of Theorem 7.* We will show that any minimizer of (19) must be a function $\sigma(x) = ax^2 + bx + c$ for some $a, b, c$. From this fact, the variational problem can be reduced to a simple quadratic programming problem over $a, b, c$, from which it is straightforward to derive (124) and (125).

If $\mu_\star = 0$ then by Lemma 3.1, we know that the solution must be linear, and hence we are done.

If $\mu_\star > 0$ then by Lemma 3.1 we know that $\sigma(x)$ cannot be a linear function. Hence, from Lemma M.3 and Lemma M.4 we know that $\lambda_3$ cannot be 0 or 2. Therefore, its solution must be of the form specified by Lemma M.8 with $c \neq 0$ and $\lambda_3 = 4k$ for some $k \in \mathbb{Z}^+$.

Define $R = \mathbb{E}((\sigma'(Z))^2)$. From the constraints (19) we know that $\lambda_1, \lambda_2$ and $R$ can be written as a linear function of $\lambda_3 = 4k$. In particular, $R = \mu_1^2 + \frac{\lambda_3}{2}\mu_\star^2$. From (121) and Lemma M.9 we can write that $R = \frac{4^k((2k)!)^2}{(2k-1)!}c^2 + \mu_1^2$, which implies that $c = \pm\left(\frac{R - \mu_1^2}{\frac{4^k((2k)!)^2}{(2k-1)!}}\right)$ is also a function of $\lambda_3$. Therefore, the solution to is parametrized by $\lambda_3$ alone which must be chosen to minimize $R = \mu_1^2 + \frac{\lambda_3}{2}\mu_\star^2$. Therefore, we must choose $\lambda_3 = 4$, the smallest possible multiple of 4. This implies that $H_{\lambda_3/2}(x/\sqrt{2})$ is a quadratic function, from which it follows that $\sigma(x)$ is also a quadratic function, and hence we are done. $\qquad\square$

## M.4 Proof of Theorem 8

Before we prove Theorem 8, we will state and prove a series of intermediary results.

**Lemma M.10.** *A necessary condition for optimality of $\sigma(x)$ is that*

$$x + \lambda_1 + \lambda_2 x + \lambda_3 \sigma(x) = 0, \text{ for all } x : \sigma'(x) \neq 0, \tag{126}$$

*where $\lambda_1$, $\lambda_2$, and $\lambda_3$ must be such that,*

$$\mathbb{E}(\sigma(Z)) = \mu_0, \tag{127}$$

$$\mathbb{E}(\sigma(Z)Z) = \mu_1, \tag{128}$$

$$\mathbb{E}((\sigma(Z))^2) = \mu_2. \tag{129}$$

*Proof of Lemma M.10.* A Lagragian for (126) is

$$\mathbb{E}(|\sigma'(Z)| + \lambda_1(\mu_0 - \sigma(Z)) + \lambda_2(\mu_1 - Z\sigma(Z)) + \lambda_3((1/2)(\mu_2 - \sigma(Z))^2)), \tag{130}$$

which can be written as $\int_{-\infty}^{\infty} L(z, \sigma, \sigma') \mathrm{d}z$ where, if we define $p(z) = \frac{1}{\sqrt{2\pi}} e^{\frac{-z^2}{2}}$, the Lagrangian density $L$ is

$$L(z, \sigma, \sigma') = p(z)(|\sigma'| + \lambda_1(\mu_0 - \sigma) + \lambda_2(\mu_1 - z\sigma) + \lambda_3(1/2)(\mu_2 - \sigma^2)). \tag{131}$$

Any variation $\sigma(z) + \epsilon\eta(z)$ of an optimal $\sigma(z)$ must yield $\mathbb{E}(|\sigma'(z) + \epsilon\eta'(z)|) \geq \mathbb{E}(|\sigma'(z)|)$. In particular, this must be the case for any variation such that $\eta(z) = 0$ whenever $\sigma'(z) = 0$. If we focus on these variations, from $\mathbb{E}(|\sigma'(z) + \epsilon\eta'(z)|) \geq \mathbb{E}(|\sigma'(z)|)$ the Euler-Lagrange equation can be derived despite $|\cdot|$ not being differentiable at 0. To be specific, it must hold that

$$\frac{\mathrm{d}}{\mathrm{d}z}\frac{\partial L}{\partial \sigma'} - \frac{\partial L}{\partial \sigma} = 0 \ \forall z : \sigma'(z) \neq 0. \tag{132}$$

Since there are no fixed boundary conditions on our integration domain $(-\infty, \infty)$, it also needs to hold that $\lim_{z \to \pm\infty} \frac{\partial L}{\partial \sigma'} = \lim_{z \to \pm\infty} p(z)\sigma'(z) = 0$, which we choose not to list in our necessary conditions.

Since $\frac{\mathrm{d}}{\mathrm{d}z}\frac{\partial L}{\partial \sigma'} = 0$ if $\sigma'(z) \neq 0$, equation (132) implies (126). First order optimality conditions imply that the Lagrange multipliers must be choose such that $\mathbb{E}(\sigma(Z)) = \mu_0, \mathbb{E}(\sigma(Z)Z) = \mu_1, \mathbb{E}((\sigma(Z))^2) = \mu_2$. □

**Lemma M.11.** *If $\lambda_3 \neq 0$, the solutions of (126) are of the form*

$$\sigma(x) = -\frac{\lambda_1}{\lambda_3} - \frac{1 + \lambda_2}{\lambda_3} \min\{\max\{x, b\}, c\}, \tag{133}$$

*for some constants $b, c$ where $b < c$ if $\frac{1+\lambda_2}{\lambda_3} \leq 0$ and $b > c$ otherwise.*

*Proof of Lemma M.11.* Since $\sigma$ is a one-dimensional function, the requirement of existence of weak derivative implies that there exists $v$ absolute continuous that agrees with $\sigma$ almost everywhere. We will work with these absolute continuous representations of $\sigma$. Other solutions can differ from the absolute continuous solutions only up to a set of measure zero with respect to the Gaussian measure.

From (126) we know that wherever $\sigma'(x) \neq 0$, we have that $\sigma(x) = ax + b$ for the same fixed $a$ and $b$. Hence, since $\sigma$ is continuous, $\sigma$ must be an alternation of flat portions and portions with the same slope $a$. Because of continuity, we cannot have a flat portion interrupt portion of slope $a$ (unless $a = 0$), as illustrated in Figure 2-right, and $\sigma$ must be as in the other two cases in Figure 2. These have a form as in (133). □

**Lemma M.12.** *Let $\sigma$ be of the form (133). Then,*

$$\mathbb{E}(|\sigma'(Z)|) = \frac{1}{2}\left|\frac{1 + \lambda_2}{\lambda_3}\right| \mathbb{P}(Z \in [b, c]), \tag{134}$$

$$\mu_1 \triangleq \mathbb{E}(Z\sigma(Z)) = -\frac{1}{2}\frac{1 + \lambda_2}{\lambda_3} \mathbb{P}(Z \in [b, c]), \tag{135}$$

*where $[b, c]$ should be interpreted as $[c, b]$ if $c < b$. In particular, $\mathbb{E}(|\sigma'(Z)|) = |\mu_1|$.*

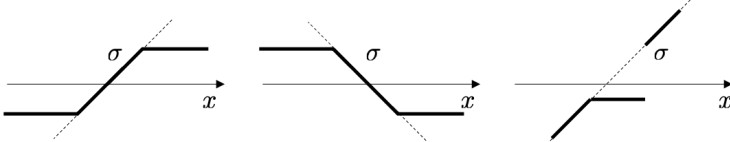

Figure 2: The first two functions are the only two possible types of continuous functions that satisfy (126). The right-most function also satisfies (126) but is not continuous.

*Proof of Lemma M.12.* From Lemma M.11, we have explicit formulas for $\sigma$ and $\sigma'$. The proof boils down to a direct calculation of the expected values, which themselves boil down to computing a few Gaussian integrals. $\square$

We are now ready to prove Theorem 8, which we restate below for convenience.

**Theorem** (8). *One minimizer of* (3) *is*

$$\sigma(x) = \mu_0 + a \min\{\max\{x, -s\}, s\} \tag{136}$$

*where* $a = \frac{\mu_1}{erf(s/\sqrt{2})}$, *erf is the Gauss error function, and* $s \in \mathbb{R}$ *is the unique solution to the equation*

$$\zeta^2 \triangleq \frac{\mu_1^2}{\mu_\star^2} = \frac{e^{\frac{s^2}{2}} erf\left(\frac{s}{\sqrt{2}}\right)^2}{e^{\frac{s^2}{2}}\left(1 - erf\left(\frac{s}{\sqrt{2}}\right)\right)\left(s^2 + erf\left(\frac{s}{\sqrt{2}}\right)\right) - \sqrt{\frac{2}{\pi}}s}, \tag{137}$$

*if* $\mu_\star \neq 0$, *and* $s = \infty$ *if* $\mu_\star = 0$.

*Proof of Theorem 8.* From Lemma M.11, we know that if $\lambda_3 \neq 0$, then any minimizer must have the form (133). From Lemma M.12 we know that all of the functions of this form have the same objective. Hence, if $\lambda_3 \neq 0$, all of the functions of the form (133) that satisfy constraints (19) are a global minimizer .

We set $\lambda_3 = 1 \neq 0$. To satisfy the three constraints (19) we have 4 remaining values to play with, namely $\lambda_1, \lambda_2, b, c$. Hence, we set $b = -c$. With a reparameterization, this leads to $\sigma$ having the form $\sigma(x) = b + a \min\{\max\{x, -s\}, s\}$, where $b$ has a new meaning. That is, $\sigma$ is flat outside of the interval $[-s/|a|, s/|a|]$, $s \geq 0$, and inside of this interval it has slope $a$. The goal is to find $a, b, s$ from the constraints (19).

From $\sigma(x) = b + a \min\{\max\{x, -s\}, s\}$ a direct computations leads to

$$\mu_0 = b, \tag{138}$$

$$\mu_1 = a\,\text{erf}\left(\frac{s}{\sqrt{2}}\right), \tag{139}$$

$$\mu_2 = a^2\left(s^2 - 1\right)\text{erfc}\left(\frac{s}{\sqrt{2}}\right) + a^2\left(1 - \sqrt{\frac{2}{\pi}}se^{-\frac{s^2}{2}}\right) + b^2, \tag{140}$$

where $\text{erfc} = 1 - \text{erf}$. We can use the first two equations to write $b$ and $a$ as a function of $\mu_0, \mu_1, s$. Substituting $a$ and $b$ with these functions in the third equation, and simplifying, leads to

$$\mu_2 = \mu_0^2 - \frac{\mu_1^2\left(-\left(s^2 - 1\right)\text{erfc}\left(\frac{s}{\sqrt{2}}\right) + \sqrt{\frac{2}{\pi}}e^{-\frac{s^2}{2}}s - 1\right)}{\text{erf}\left(\frac{s}{\sqrt{2}}\right)^2}. \tag{141}$$

Recalling that $\mu_2 = \mu_\star^2 + \mu_1^2 + \mu_0^2$, replacing this definition into the above equation, and simplifying leads to

$$\frac{\mu_1^2}{\mu_\star^2} \triangleq \zeta^2 = -\frac{e^{\frac{s^2}{2}}\text{erf}\left(\frac{s}{\sqrt{2}}\right)^2}{e^{\frac{s^2}{2}}\left(\text{erf}\left(\frac{s}{\sqrt{2}}\right) - 1\right)\left(s^2 + \text{erf}\left(\frac{s}{\sqrt{2}}\right)\right) + \sqrt{\frac{2}{\pi}}s}. \tag{142}$$

One can show that the function on the right hand side (142) is monotonic increasing in $s \in [-\infty, \infty]$ with range $[0, \infty]$, which implies that there is only one $s$ that solves (142).

$\square$

## M.5 PROOF OF THEOREM 9

This proof involves heavy algebraic computations. To aid the reader, this paper is accompanied by a Mathematica file that symbolically checks the equations both in the theorem statement as well as in the proof below. This file is in the supplementary zip file, as well as in the following Github link `https://github.com/Jeffwang87/RFR_AF`. It is called `RunMeToCheckProofOfTheorem9.nb`.

*Proof of Theorem 9.* Theorem 9 amounts a statement about the solutions of the optimization problem (22) for regime $R_1$.

Its proof amounts to studying the local minima of the objective via its first and second derivatives, both on the inside and on the boundary of the variables' domain.

We will prove the theorem for $\psi_1 < \psi_2$ and $\psi_1 > \psi_2$ separately. For $\psi_1 = \psi_2$ the objective is not defined.

In what follows, we let $L = (1 - \alpha)\mathcal{E}_{R_1}^\infty + \alpha \mathcal{S}_{R_1}^\infty$. We will use the fact that $L$ is a one-dimensional function of $\zeta^2 \in [0, +\infty]$, as can be seen from (13) and (16). We will use this and the fact that (28) defines a monotonic function between $\chi$ and $\zeta^2$, to express $L$ as a function of $\chi$ and study the solutions of the optimization problem (22)

in the variable $\chi$. Notice that (28) can be solved for $\zeta^2$ as $\zeta^2 = \frac{\mu_1^2}{\mu_\star^2} = \frac{\chi + \min\{\psi_1, \psi_2\}}{\chi(-1 + \chi + \min\{\psi_1, \psi_2\})}$ and, calling $\chi$ by $x$, this can be arranged to get (23), which connects optimal values of $\chi$ with optimal values of $\mu_0, \mu_1, \mu_2$. Henceforth, we denote $\chi$ by $x$. We note that since $\zeta^2 \in [0, +\infty]$, from (28) it follows that $x \in [x_L, x_R] \triangleq [-\min\{\psi_1, \psi_2\}, \min\{0, 1 - \min\{\psi_1, \psi_2\}\}]$. Hence, we only need to study the function $L(x)$ in this interval.

**Case 1, $\psi_1 < \psi_2$:**

For the sake of simplicity, and for the most part, we will assume that $\psi_1 \neq 1$ and omit the argument for $\psi_1 = 1$. The argument when $\psi_1 = 1$ is almost identical to the argument when $\psi_1 \neq 1$ if we work with the extended reals $[-\infty, \infty]$. Furthermore, most conclusions for $\psi_1 = 1$ can be obtained by taking the limit of $\psi_1 \to 1$ with $\psi_1 \neq 1$. The only situation where this is not the case is that the polynomial $p(x)$, referenced right after Table 1, has an expression when $\psi_1 = 1$ that cannot be obtained as the limit of its expression for $1 \neq \psi_1$. We will also assume that $\alpha > 0$. Recall that we are already assuming that $\alpha < 1$ because when $\alpha = 1$ our problem is trivial. If $\alpha = 0$ one can check that $L(x)$ is a decreasing line, hence the minimum is at $x = x_R$, and this solution can be obtained from the first row of Table 1. This solution for $\alpha = 0$ can also be obtained by studying $\alpha > 0$ and taking $\alpha \to 0$. Below thus assume that $0 < \alpha < 1$.

We start by studying the second derivative of $L$. A direct computation yields

$$\frac{\mathrm{d}^2 L}{\mathrm{d}x^2} = \tilde{L}(x) + C, \text{ where } C = \frac{2F_1^2 \alpha}{\psi_1 - \psi_2} \tag{143}$$

and $\tilde{L}(x) = -\frac{2\alpha F_1^2 \psi_1 \left(-3x^2(\psi_1 - 1) - 2x^3 + (\psi_1 - 1)^2 \psi_1\right)}{\left((x + \psi_1)^2 - \psi_1\right)^3}$.

A tedious calculation (omitted) shows that $\frac{\mathrm{d}^2 \tilde{L}}{\mathrm{d}x^2} \geq 0$ for $x \in [x_L, x_R]$ (i.e. it is convex), that $\tilde{L}(x_L) < \tilde{L}(x_R)$, and that $\frac{\mathrm{d}\tilde{L}}{\mathrm{d}x}(x = x_L) < 0$. From this it follows that, depending on the value of $\psi_2$, the concavity $\frac{\mathrm{d}^2 L}{\mathrm{d}x^2}$ is positive or negative, as illustrated in the following figure:

To be specific, starting from large $\psi_2$, i.e. small $-C$, and decreasing its value, i.e. increasing $-C$, we obtain the following four scenarios. While $\psi_2$ is large and $-C$ is bellow $A_2$, the function $L$ is convex. After $\psi_2$ reaches a value $\beta_1$ at which $-C$ touches $A_2$, the function $L$ is convex for small $x$, then concave, and then convex for large $x$. As $\psi_2$ keeps decreasing, and after it reaches a value $\beta_2$ at

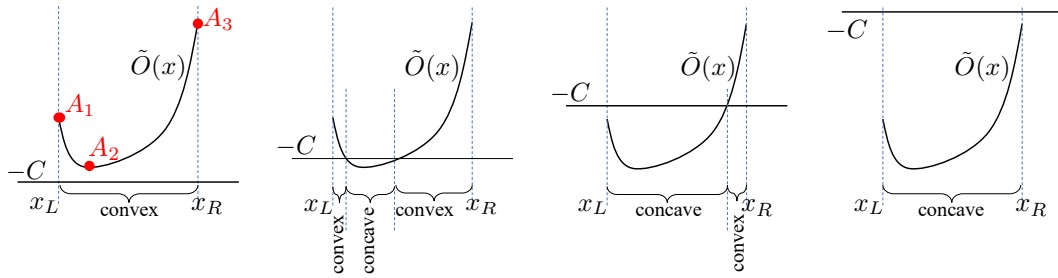

Figure 3: Depending on the value of $\psi_2$, the function $L$ is convex, convex-concave-convex, concave-convex, or concave respectively. The points A, B, and C will be referenced later in the proof. It is possible to compute closed-form expressions for the $y$-coordinate of these points.

which $-C$ touches $A_1$, the function $L$ is concave for small $x$, and then convex for large $x$. Finally, after $\psi_2$ reaches a value $\beta_3$ at which $-C$ touches $A_3$, the function $L$ is concave. Note that by definition $0 \le \beta_3 < \beta_2 < \beta_1$.

It is possible to compute closed-form expressions for the $y$-coordinate of the points $A_1, A_2, A_3$, which we denote by $A_1, A_2,$ and $A_3$. Namely, $A_1 = F_1^2 \alpha (2 + \frac{2}{\psi_1}), A_2 = F_1^2 \alpha \left( \frac{\sqrt{1-\psi_1}+1}{\psi_1} - \frac{1}{2} \right)$, if $\psi_1 \le 1$, and $A_2 = F_1^2 \alpha \left( \sqrt{\frac{\psi_1 - 1}{\psi_1}} - \frac{1}{2\psi_1} + 1 \right)$, if $\psi_1 > 1$, and $A_3 = F_1^2 \alpha \max \left\{ \frac{2}{\psi_1 - \psi_1^2}, \frac{2\psi_1}{\psi_1 - 1} \right\}$. Using the closed-form expression for $-C$, see (143), we find closed-form expressions for $\beta_1, \beta_2,$ and $\beta_3$ as $\beta_1 = \psi_2 : A_2(\psi_2) + C(\psi_2) = 0, \beta_2 = \psi_2 : A_1(\psi_2) + C(\psi_2) = 0,$ and $\beta_3 = \psi_2 : A_3(\psi_2) + C(\psi_2) = 0$. By definition of $\beta_1, \beta_2,$ and $\beta_3$, these equations have a unique solution when $\psi_2 > \psi_1 > 0$ and their explicit expressions are given in (45), (46) and (47) respectively.

Now that we have characterized the curvature of $L(x)$, we are ready to locate its global minimum. To do so, we will use the curvature of $L(x)$ together with the first-order optimally condition $\frac{dL}{dx} = 0$, and the following three extra pieces of information: the sign of the derivative of $L(x)$ at $x = x_L$; the sign of the derivative of $L(x)$ at $x = x_R$; and the sign of $L(x_L) - L(x_R)$.

A direct computation yields

$$\frac{dL}{dx} = \frac{p_0 + p_1 x + p_2 x^2 + p_3 x^3 + p_4 x^4 + p_5 x^5}{(\psi_1 - (x + \psi_1)^2)^2 (\psi_1 - \psi_2)}, \text{ if } \psi_1 \ne 1, \text{ and} \tag{144}$$

$$\frac{dL}{dx} = \frac{q_0 + q_1 x + q_2 x^2 + q_3 x^3}{(2 + x)^2 (-1 + \psi_2)}, \text{ if } \psi_1 = 1. \tag{145}$$

where the coefficients $p_0, \dots, p_5$ and $q_0 \dots, q_3$ are, apart from a multiplying constant, given in (51)-(60).

The roots of the first denominator, i.e. $-\sqrt{\psi_1} - \psi_1$ and $\sqrt{\psi_1} - \psi_1$, are not a root of the first numerator when $\psi_1 \ne 1$, and the roots of the second denominator, i.e. $-2$, are not a root of the second numerator when $\psi_2 > \psi_1 = 1$. Therefore, the first-order optimality conditions are $p(x) = 0$.

Not all solutions of $p(x) = 0$ minimize $L(x)$. To locate the minimizer, we use the sign of the derivative of $L(x)$ at $x = x_L$; the sign of the derivative of $L(x)$ at $x = x_R$; and the sign of $L(x_L) - L(x_R)$. These signs can be determined using Lemma M.13. Lemma M.13 also proves Remark 10.

**Lemma M.13.** *If $\psi_1 < \psi_2$, the following relationships hold,*

$$\left.\frac{dL}{dx}\right|_{x=x_L} = -\frac{F_1^2(\alpha + (\alpha-1)\psi_2) + \alpha\left(F_\star^2 + \tau^2\right)}{\psi_1 - \psi_2}, \tag{146}$$

$$\left.\frac{dL}{dx}\right|_{x=x_R} = \frac{F_1^2\psi_2 - \alpha\left(F_1^2(-4\psi_1 + 3\psi_2 + 1) + F_\star^2 + \tau^2\right)}{\psi_1 - \psi_2} \ if\ \psi_1 \le 1, \tag{147}$$

$$\left.\frac{dL}{dx}\right|_{x=x_R} = \frac{F_1^2(\alpha + 2\alpha\psi_1 - 3\alpha\psi_2 + \psi_2) - \alpha\left(F_\star^2 + \tau^2\right)}{\psi_1 - \psi_2} \ if\ \psi_1 > 1, \tag{148}$$

$$L(x_L) - L(x_R) = \frac{\psi_1\left(F_1^2(\alpha - 2\alpha\psi_1 + (2\alpha - 1)\psi_2) + \alpha\left(F_\star^2 + \tau^2\right)\right)}{\psi_1 - \psi_2} \ if\ \psi_1 \le 1,\ and \tag{149}$$

$$L(x_L) - L(x_R) = \frac{\alpha\left(F_\star^2 + \tau^2\right) - F_1^2(\alpha\psi_1 - 2\alpha\psi_2 + \psi_2)}{\psi_1 - \psi_2} \ if\ \psi_1 > 1. \tag{150}$$

*Also,*

$$\left.\frac{dL}{dx}\right|_{x=x_L} < 0\ if\ \alpha < \alpha_L\ and\ \left.\frac{dL}{dx}\right|_{x=x_L} \ge 0\ if\ \alpha \ge \alpha_L, \tag{151}$$

$$\left.\frac{dL}{dx}\right|_{x=x_R} < 0\ if\ \alpha < \alpha_R\ and\ \left.\frac{dL}{dx}\right|_{x=x_R} \ge 0\ if\ \alpha \ge \alpha_R,\ and \tag{152}$$

$$L(x_L) - L(x_R) > 0\ if\ \alpha < \alpha_C\ and\ L(x_L) - L(x_R) \le 0\ if\ \alpha \ge \alpha_C, \tag{153}$$

*where $\alpha_L, \alpha_R$ and $\alpha_C$ are given in* (48), (50), *and* (49) *respectively.*

*Furthermore, the following are true and help determine from which row in Table 1 to read x. If $\psi_2 < \min\{2\psi_1, \psi_1 + 1\}$, then $\alpha_L < \alpha_C < \alpha_R$. If $\psi_2 > \min\{2\psi_1, \psi_1 + 1\}$, then $\alpha_R < \alpha_C < \alpha_L$. If $\psi_2 = \min\{2\psi_1, \psi_1 + 1\}$, then $\alpha_L = \alpha_C = \alpha_R$. In particular, it follows from this that if $\psi_2 \ge \beta_1$, then $\alpha_R \le \alpha_C \le \alpha_L$ and if $\psi_2 \le \beta_3$, then $\alpha_L \le \alpha_C \le \alpha_R$.*

*Finally, $\alpha_L, \alpha_R, \alpha_C \in [0, 1]$.*

*Proof.* The derivation of (146)-(150) follows from direct substitution of $x = x_L$ or $x = x_R$ into $L$ (for which we have expressions from Theorems 1 and 4) or its derivative (in eq. (144)-(145)). The derivation of (151)-(153) follows from the observation that the equations (146)-(150) are linear in $\alpha$, and hence we can easily compute the values of $\alpha$ at which these expressions change from a negative value to a positive value. Once an explicit formula for $\alpha_L, \alpha_C$, and $\alpha_R$ is obtained, it is easy to find the criteria to decide their relative magnitude by comparing the term under parenthesis in the denominators of (48), (50), and (49). It is also easy to see from their formulas that their value is always in the range $[0, 1]$. □

To finish the proof of Theorem 9 we consider the different scenarios in Table 1. In what follows, statements about the concavity of $L$ are justified via the explanation accompanying Figure 3, and statements about the slope of $L$ are based on Lemma M.13.

**Case 1.1,** $\psi_1 < \psi_2 \wedge \psi_2 \ge \beta_1 \wedge \alpha < \alpha_L \wedge \alpha < \alpha_R$**:** The function $L$ is convex and is decreasing at $x = x_L$ and decreasing at $x = x_R$. Hence its minimum is at $x = x_R$.

**Case 1.2,** $\psi_1 < \psi_2 \wedge \psi_2 \ge \beta_1 \wedge \alpha_R < \alpha < \alpha_L$**:** The function $L$ is convex and it is decreasing at $x = x_L$ and increasing at $x = x_R$. Hence it has a unique minimizer at $x = x_1$.

When $\psi_1 < \psi_2 \wedge \psi_2 \ge \beta_1$, we can use Lemma M.13 and a tedious calculation (omitted) to show that $\alpha_R < \alpha_L$, which is why the 4th and 5th rows of the first column of Table 1 are empty. A simpler way to see that the 4th and 5th rows of the first column of Table 1 must be empty is as follows. Since $\psi_1 < \psi_2 \wedge \psi_2 \ge \beta_1$ then we know (from the argument following with Figure 3) that $L$ is convex. If $\alpha > \alpha_L$ and $\alpha < \alpha_R$ (i.e. $E_1$ holds) then by Lemma M.13 we know that $L$ is increasing at $x = x_L$ and decreasing at $x = x_R$, but this is impossible for a convex function.

**Case 1.3,** $\psi_1 < \psi_2 \wedge \psi_2 \ge \beta_1 \wedge \alpha > \alpha_L \wedge \alpha > \alpha_R$**:** The function $L$ is convex and it is increasing at both $x = x_L$ and $x = x_R$. Hence its minimum is at $x = x_L$.

**Case 1.4,** $\psi_1 < \psi_2 \wedge \beta_2 < \psi_2 < \beta_1 \wedge \alpha < \alpha_L \wedge \alpha < \alpha_R$**:** The function $L$ is first convex, then concave and then convex. It is decreasing at both $x = x_L$ and at $x = x_R$. Hence $L$ has at most one local minimum in the interior of the domain, at $x = x_1$ if it exists, and a local minimum at $x_R$, the domain being $[x_L, x_R]$. Therefore, the minimum can be expressed as $x_R \sqcup x_1$.

**Case 1.5,** $\psi_1 < \psi_2 \wedge \beta_2 < \psi_2 < \beta_1 \wedge \alpha_R < \alpha < \alpha_L$**:** The function $L$ is first convex, then concave and then convex. It is decreasing at $x = x_L$ and increasing at $x = x_R$. Hence $L$ has no local minimum at the end points of the domain $[x_L, x_R]$. Also, $L$ either has exactly one local minimum in the interior of the domain, at $x = x_1$, or, $L$ has exactly two local minimum and one local maximum (resp.) in the interior of the domain, at $x = x_1$, $x = x_3$ and $x = x_2$ respectively. Therefore, the minimum can be expressed as $x_1 \sqcup x_3$. Note that $x_1$ always exists but $x_3$ might not.

**Case 1.6,** $\psi_1 < \psi_2 \wedge \beta_2 < \psi_2 < \beta_1 \wedge \alpha_L < \alpha < \alpha_R$**:** The function $L$ is first convex, then concave and then convex. It is increasing at $x = x_L$ and decreasing at $x = x_R$. Hence, the minimum is either at the $x_L$ or $x_R$, depending whether $\alpha > \alpha_C$ or $\alpha < \alpha_C$. This case is an example where it is clear that allowing $\alpha = \alpha_C$ leads to non-uniqueness in the optimal $x$. See Remark 9.

**Case 1.7,** $\psi_1 < \psi_2 \wedge \beta_2 < \psi_2 < \beta_1 \wedge \alpha > \alpha_L \wedge \alpha > \alpha_R$**:** The function $L$ is first convex, then concave, and then convex. It is increasing at both $x = x_R$ and $x = x_L$. Hence $L$ either has no critical point in the interior of the domain, or it has two critical point in the interior of the domain, at $x = x_1$ (local maximum) and $x = x_2$ (local minimum) if they exists, and always has a local minimum at $x_L$. Therefore, the minimum can be expressed as $x_L \sqcup x_2$. Note that $x_L$ always exists, but $x_2$ not always exists.

**Case 1.8,** $\psi_1 < \psi_2 \wedge \beta_3 < \psi_2 \leq \beta_2 \wedge \alpha < \alpha_L \wedge \alpha < \alpha_R$**:** The function $L$ is first concave and convex. and is decreasing at both $x = x_L$ and $x = x_R$. Hence its minimum is at $x = x_R$.

**Case 1.9,** $\psi_1 < \psi_2 \wedge \beta_3 < \psi_2 \leq \beta_2 \wedge \alpha_R < \alpha < \alpha_L$**:** The function $L$ is first concave and convex. It is decreasing at $x = x_L$ and increasing at $x = x_R$. Hence, $L$ has exactly one local minimum in the interior of the domain, which is at $x = x_1$ and always exists, and which is also a global minimum.

**Case 1.10,** $\psi_1 < \psi_2 \wedge \beta_3 < \psi_2 \leq \beta_2 \wedge \alpha_L < \alpha < \alpha_R$**:** The function $L$ is first concave and convex. It is increasing at $x = x_L$ and decreasing at $x = x_R$. Hence, the minimum is either at the $x_L$ or $x_R$, depending whether $\alpha > \alpha_C$ or $\alpha < \alpha_C$.

**Case 1.11,** $\psi_1 < \psi_2 \wedge \beta_3 < \psi_2 \leq \beta_2 \wedge \alpha > \alpha_L \wedge \alpha > \alpha_R$**:** The function $L$ is first concave and convex. It is increasing at both $x = x_L$ and $x = x_R$. Hence $L$ either has no critical point in the interior of the domain or it has two critical point in the interior of the domain, at $x = x_1$ (local maximum) and $x = x_2$ (local minimum) if they exists, and always has a local minimum at $x_L$. Therefore, the minimum can be expressed as $x_L \sqcup x_2$. Note that $x_L$ always exists, but $x_2$ not always exists.

**Case 1.12,** $\psi_1 < \psi_2 \wedge \psi_2 \leq \beta_3 \wedge \alpha < \alpha_L \wedge \alpha < \alpha_R$**:** The function $L$ is concave. It is decreasing at both $x = x_L$ and $x = x_R$. Hence, its minimum is at $x = x_R$

When $\psi_1 < \psi_2 \wedge \psi_2 \leq \beta_3$, Lemma M.13 and a tedious calculation (omitted) shows that that $\alpha_L < \alpha_R$, which is why the 2nd and 3rd rows of the last column of Table 1 are empty. A simpler way to see that the 2nd and 3rd rows of the last column of Table 1 must be empty is as follows. Since $\psi_1 < \psi_2 \wedge \psi_2 \leq \beta_3$ we know (from the argument following Figure 3) that $L$ is concave. If $\alpha < \alpha_L$ and $\alpha > \alpha_R$ (i.e. $E_2$ holds) then by Lemma M.13 we known that $L$ is decreasing at $x = x_L$ and increasing at $x = x_R$, but this is impossible for a concave function.

**Case 1.13,** $\psi_1 < \psi_2 \wedge \psi_2 \leq \beta_3 \wedge \alpha_L < \alpha < \alpha_R$**:** The function $L$ is concave. It is increasing at $x = x_L$ and decreasing at $x = x_R$. Hence, the minimum is either at the $x_L$ or $x_R$, depending whether $\alpha > \alpha_C$ or $\alpha < \alpha_C$.

**Case 1.14,** $\psi_1 < \psi_2 \wedge \psi_2 \leq \beta_3 \wedge \alpha > \alpha_L \wedge \alpha > \alpha_R$**:** The function $L$ is concave. It is increasing at $x = x_L$ and at $x = x_R$. Hence, the minimum is at $x_L$.

**Case 2,** $\psi_1 > \psi_2$**:**

The case when $\psi_2 = 1$ can be proved by taking appropriate limits of the case when $\psi_2 \neq 1$. For now, we assume that $\psi_2 \neq 1$.

We first prove that the second derivative of $L$ is convex, just like in the case when $\psi_1 < \psi_2$. A direct computation yields

$$\frac{\mathrm{d}^2 L}{\mathrm{d}x^2} = \tilde{L}(x) + C, \ \text{ where } C = -\frac{2F_1^2 \alpha}{\psi_1 - \psi_2} \tag{154}$$

and $\tilde{L}(x) = -\frac{2\psi_2 \left( 3x^2 \left( (F_\star^2 + \tau^2) - F_1^2(\psi_2 - 1) \right) - 2x^3 F_1^2 + 6x(F_\star^2 + \tau^2)\psi_2 + \psi_2 \left( F_1^2(\psi_2 - 1)^2 + (F_\star^2 + \tau^2)(3\psi_2 + 1) \right) \right)}{\left( (x + \psi_2)^2 - \psi_2 \right)^3}$.

A tedious calculation (omitted) shows that $\frac{\mathrm{d}^2 \tilde{L}}{\mathrm{d}x^2} \geq 0$ for $x \in [x_L, x_R]$ (i.e. it is convex), that $\tilde{L}(x_L) < \tilde{L}(x_R)$, and that $\frac{\mathrm{d}\tilde{L}}{\mathrm{d}x}(x = x_L) < 0$. To do this calculation, we recommend the following. First break $\tilde{L}$ into two components. One component proportional to $F_1^2$, called $\tilde{L}_1$, and one component proportional to $(F_\star^2 + \tau^2)$, called $\tilde{L}_\star$. Then, show that both components are convex, that $\tilde{L}_\star(x_L) \leq \tilde{L}_\star(x_R)$, that $\tilde{L}_1(x_L) < \tilde{L}_1(x_R)$, that $\frac{\mathrm{d}\tilde{L}_\star}{\mathrm{d}x}(x = x_L) = 0$, and that $\frac{\mathrm{d}\tilde{L}_1}{\mathrm{d}x}(x = x_L) < 0$.

From this it follows that, depending on the value of $\psi_1$, the concavity $\frac{\mathrm{d}^2 L}{\mathrm{d}x^2}$ is positive or negative. The situation is exactly the same as in the Figure 3 but now the $x$ axis is $\psi_1$, the points $A_1$, $A_2$ and $A_3$ are different, and so are the definitions of $\beta_1, \beta_2$ and $\beta_3$. We do however have that, by definition, $0 \leq \beta_3 < \beta_2 < \beta_1$.

With a slight abuse of notation we refer to the $y$-coordinate value of this points by $A_1$, $A_2$, and $A_3$. We now have $A_1 = \frac{2\left( F_1^2(\psi_2 + 1) + (F_\star^2 + \tau^2) \right)}{\psi_2}$ and $A_3 = -\frac{2\max\{1, \psi_2\}\left( F_1^2(\psi_2 - 1)^2 + (F_\star^2 + \tau^2)(3\min\{1, \psi_2\} + \max\{1, \psi_2\}) \right)}{(\psi_2 - 1)^3 \min\{1, \psi_2\}}$. Notice that when $\psi_2 = 1$ we have that $A_3 = +\infty$, and in fact we also have $L(x_R) = +\infty$. Getting $A_2$ is a bit more complicated. From the first order condition $\frac{\mathrm{d}\tilde{L}}{\mathrm{d}x} = 0$ and the convexity of $\tilde{L}$ – recall that $\tilde{L}$ a rational function – we can extract that the $x$-coordinate of $A_2$ is the unique root of a 4th degree polynomial $h(x) = h_0 + h_1 x + h_2 x^2 + h_3 x^3 + h_4 x^4$ in the range $x \in [x_L, x_R]$, where $h_0 = \psi_2^2 \left( \rho(\psi_2 - 1)^2 + 2(\psi_2 + 1) \right)$, $h_1 = 2\psi_2 \left( \rho(\psi_2 - 1)^2 + (3\psi_2 + 1) \right)$, $h_2 = 6\psi_2$, $h_3 = 2(1 - \rho(\psi_2 - 1))$, $h_4 = -\rho$, Call this root $x_{A_2}$. We then have $A_2 = \tilde{L}(x_{A_2})$. It turns out that we can write $A_2$ directly as the solution of $g(A_2/((\tau^2 + F_\star^2)\psi_2)) = 0$, where $g(x) = g_0 + g_1 x + g_2 x^2 + g_3 x^3 + g_4 x^4$ with $g_0 = \rho^4$, $g_1 = -4(\rho\psi_2 + \rho + 1)(\rho\psi_2(2\rho\psi_2 - 3\rho + 4) + 2(\rho + 1)^2)$, $g_2 = -4\psi_2^2 \left( \rho\psi_2(7\rho\psi_2 - 20\rho + 14) + 7(\rho + 1)^2 \right)$, $g_3 = -16\psi_2^4(\rho\psi_2 + \rho + 1)$, and $g_4 = 16\psi_2^6$.

Using the expressions for $A_1$ and $A_3$, and the expression for $C$, see (154), $\beta_2$ and $\beta_3$ are defined as $\beta_2 = \psi_1 : A_1(\psi_1) + C(\psi_1) = 0$, and $\beta_3 = \psi_1 : A_3(\psi_1) + C(\psi_1) = 0$. By definition of $\beta_2$ and $\beta_3$, these equations have a unique solution when $\psi_1 > \psi_2 > 0$ and their expressions are given in (62) and (63) respectively. We can also define $\beta_1$ as the solution of $\beta_1 = \psi_1 : A_2(\psi_1) + C(\psi_1) = 0$. By definition of $\beta_1$, the solution is unique in the range $\psi_1 > \psi_2 > 0$. We can use the fact that $A_2 = \tilde{L}(x_{A_2})$ to write that $\beta_1 = \psi_2 + \frac{2F_1^2 \alpha}{\tilde{L}(x_{A_2})}$. We can also use the fact that $g(A_2/((\tau^2 + F_\star^2)\psi_2)) = 0$, which implies that $g(-C(\psi_1) / ((\tau^2 + F_\star^2)\psi_2)) = 0$ when $\psi_1 = \beta_1$, to write $\beta_1$ as the root of a 4th degree polynomial $r(x)$ that is specified in Appendix J.4, which is the way in which we decide to state Theorem 9.

Now that we have characterized the curvature of $L(x)$, we are ready to locate its global minimum. To do so, we will use the curvature of $L(x)$ together with the first-order optimally condition $\frac{\mathrm{d}L}{\mathrm{d}x} = 0$, and the following three extra pieces of information: the sign of the derivative of $L(x)$ at $x = x_L$; the sign of the derivative of $L(x)$ at $x = x_R$; and the sign of $L(x_L) - L(x_R)$.

A direct computation yields

$$\frac{\mathrm{d}L}{\mathrm{d}x} = \frac{p_0 + p_1 x + p_2 x^2 + p_3 x^3 + p_4 x^4 + p_5 x^5}{(\psi_2 - (x + \psi_2)^2)^2(\psi_1 - \psi_2)}, \tag{155}$$

where the coefficients $p_0, \ldots, p_5$ are, apart from a multiplying constant, given in (70)- (75). The roots of the denominator, i.e. $-\sqrt{\psi_2} - \psi_2$ and $\sqrt{\psi_2} - \psi_2$, are not a root of the numerator, therefore, the first order optimalit conditions are $p(x) = 0$.

Not all solutions of $p(x) = 0$ minimize $L(x)$. To locate the minimizer, we use the sign of the derivative of $L(x)$ at $x = x_L$; the sign of the derivative of $L(x)$ at $x = x_R$; and the sign of $L(x_L) - L(x_R)$. These signs can be determined using Lemma M.14. Lemma M.14 also proves Remark 11.

**Lemma M.14.** *If $\psi_1 > \psi_2$ then the following relationships hold.*

$$\left.\frac{dL}{dx}\right|_{x=x_L} = \frac{\alpha\left(F_\star^2+\tau^2\right)+F_1^2\left(\alpha\left(-\psi_2\right)+2(\alpha-1)\psi_1+\alpha+\psi_2\right)}{\psi_1-\psi_2}. \tag{156}$$

*If $\psi_2 = 1$ then* $\left.\dfrac{dL}{dx}\right|_{x=x_R} = +\infty.$ \hfill (157)

*If $\psi_2 < 1$ then* \hfill (158)

$$\left.\frac{dL}{dx}\right|_{x=x_R} = \frac{\left(\alpha\psi_2^2-2(\alpha+1)\psi_2+\alpha+2\psi_1\right)\left(F_\star^2+\tau^2\right)+F_1^2\left(\psi_2-1\right)^2\left(2\alpha\psi_1-(3\alpha+1)\psi_2+\alpha\right)}{\left(\psi_1-\psi_2\right)\left(\psi_2-1\right)^2}. \tag{159}$$

*If $\psi_2 > 1$ then* \hfill (160)

$$\left.\frac{dL}{dx}\right|_{x=x_R} = \frac{F_1^2\left(\psi_2-1\right)^2\left(-2\alpha\psi_1+(\alpha+1)\psi_2+\alpha\right)-\left(\psi_2\left((\alpha-2)\psi_2-2\alpha+2\psi_1\right)+\alpha\right)\left(F_\star^2+\tau^2\right)}{\left(\psi_2-1\right)^2\left(\psi_2-\psi_1\right)}. \tag{161}$$

*If $\psi_2 = 1$ then $L(x_L) - L(x_R) = -\infty$.* \hfill (162)

*If $\psi_2 < 1$ then* \hfill (163)

$$L(x_L) - L(x_R) = \frac{\psi_2\left(F_1^2\left(\psi_2-1\right)\left((1-2\alpha)\psi_1+\alpha\left(2\psi_2-1\right)\right)-\left((\alpha+1)\psi_2-\alpha-\psi_1\right)\left(F_\star^2+\tau^2\right)\right)}{\left(\psi_1-\psi_2\right)\left(\psi_2-1\right)}. \tag{164}$$

*If $\psi_2 > 1$ then* \hfill (165)

$$L(x_L) - L(x_R) = \tag{166}$$

$$\frac{\psi_2\left((\alpha-1)\left(F_\star^2+\tau^2\right)+\alpha F_1^2\right)+\psi_1\left((2\alpha-1)F_1^2\left(\psi_2-1\right)+F_\star^2+\tau^2\right)-\alpha\left(F_\star^2+\tau^2\right)-\alpha F_1^2\psi_2^2}{\left(\psi_2-1\right)\left(\psi_2-\psi_1\right)}. \tag{167}$$

*Also, the following is true.*

*If $\alpha < \alpha_L$ then* $\left.\dfrac{dL}{dx}\right|_{x=x_L} < 0$*; and if $\alpha \geq \alpha_L$ then* $\left.\dfrac{dL}{dx}\right|_{x=x_L} \geq 0.$ \hfill (168)

*Let $\psi_2 \neq 1$. If $E_1$ is true then* $\left.\dfrac{dL}{dx}\right|_{x=x_R} < 0$*; and if $E_2$ is true then* $\left.\dfrac{dL}{dx}\right|_{x=x_R} > 0.$ \hfill (169)

*Conditions $E_1$ and $E_2$ are defined in Theorem 9.*

*Let $\psi_2 \neq 1$. If $\alpha < \alpha_C$ then $L(x_L) - L(x_R) > 0$; and if $\alpha \geq \alpha_C$ then $L(x_L) - L(x_R) \leq 0$* \hfill (170)

*Constant $\alpha_C$ is defined in equation (65) in Theorem 9.*

*Furthermore, we always have that $\alpha_L \in [0,1]$. It also holds that $\alpha_C \in [0,1]$ if and only if $\psi_1 \leq \psi_2/(1 - \rho|1-\psi_2|)$ when $0 < \rho \leq \frac{1}{|\psi_2-1|}$ or $\psi_1 \geq \psi_2/(1 - \rho|1-\psi_2|)$ when $\rho \geq \frac{1}{|\psi_2-1|}$. We have that $\alpha_R \in [0,1]$ if and only if*

$$\psi_1 \leq \psi_2 + \frac{1}{2}\rho(\psi_2 - 1)^2 \min\{\psi_2, 1\} \text{ and} \tag{171}$$

$$\psi_1 \geq \frac{-1 + \rho(\psi_2 - 1)^2(2\psi_2 + \min\{2\psi_2 - 1, 1\}) + \psi_2(2 + \max\{\psi_2, 2 - \psi_2\})}{2(\max\{\psi_2, 1\} + \rho(\psi_2 - 1)^2)}. \tag{172}$$

*Finally, $A \geq B$ always, where $A$ and $B$ are given in (67) and (68). Furthermore, $B < \psi_2$ if and only if $1/\rho > \gamma \triangleq \min\{1, \max\{0, 2\psi_2 - 1\}\}$.*

*Proof.* Except (162) and (157), the derivation of (156)-(167) follows from direct substitution of $x = x_L$ or $x = x_R$ into $L$ – for which we have expressions from Theorems 1 and 4 – or its derivative in equation 155.

Equation (162), for when $\psi_2 = 1$, is obtained by taking the limit of (164) and (167) as $\psi_2 \uparrow 1$ and $\psi_2 \downarrow 1$ respectively. Equation (157) for when $\psi_2 = 1$ is obtained by taking the limit of (159) and (161) as $\psi_2 \uparrow 1$ and $\psi_2 \downarrow 1$ respectively.

The derivation of condition (168) follows from the observation that the equation (156) is linear in $\alpha$ and is always negative for $\alpha = 0$ and positive for $\alpha = 1$. Hence, we can easily compute a value $\alpha_L \in [0, 1]$ at which the expression changes from a negative value to a positive value. The expression for $\alpha_L$ we obtain is (64).

The derivation of condition (169) can be obtained through following the observations. First notice that $\frac{dL}{dx}|_{x=x_R}$ is a linear increasing function of $\alpha$. Now focus on the following three implications.

1. If at $\alpha = 1$ we have $\frac{dL}{dx}|_{x=x_R} < 0$, then $\frac{dL}{dx}|_{x=x_R} < 0$ for all $\alpha \in [0, 1]$;

2. If at $\alpha = 0$ we have $\frac{dL}{dx}|_{x=x_R} > 0$, then $\frac{dL}{dx}|_{x=x_R} > 0$ for all $\alpha \in [0, 1]$;

3. If at $\alpha = 0$ we have $\frac{dL}{dx}|_{x=x_R} < 0$ and at $\alpha = 1$ we have $\frac{dL}{dx}|_{x=x_R} > 0$, then there exists $\alpha_R \in [0, 1]$ such that $(\alpha < \alpha_R \Rightarrow \frac{dL}{dx}|_{x=x_R} < 0)$ and $(\alpha > \alpha_R \Rightarrow \frac{dL}{dx}|_{x=x_R} > 0)$;

A direct computation shows that the sufficient condition in the first implication holds if and only if $\psi_1 < B$, where $B$ is given in (68); the sufficient condition in the second implication holds if and only if $\psi_1 > A$, where $A$ is given in (67); the sufficient condition in the the third implication holds if and only if $B < \psi_1 < A$. Therefore, if $E_1 = (\psi_1 < B) \vee ((\alpha < \alpha_R) \wedge (B < \psi_1 < A))$ is true, we can use the first or third implication to conclude that $\frac{dL}{dx}|_{x=x_R} < 0$. Also, if $E_2 = (\psi_1 > A) \vee ((\alpha > \alpha_R) \wedge (B < \psi_1 < A))$ is true, we can use the second or third implication to conclude that $\frac{dL}{dx}|_{x=x_R} > 0$.

A direct calculation shows that $A \geq B$ and that $B < \psi_2$ if and only if $1/\rho > \gamma \triangleq \min\{1, \max\{0, 2\psi_2 - 1\}\}$. A direct calculation also shows that $\alpha_R \in [0, 1]$ if and only if the conditions in (171) and (172) hold.

Condition (170) can be obtained through following the observations. First, notice that both (164) and (167) are linear decreasing functions of $\alpha$. Second, when $\alpha = 1$ we always have $L(x_L) - L(x_R) < 0$. Therefore, there exists $\alpha_C \leq 1$ such that $(\alpha < \alpha_C \Rightarrow L(x_L) - L(x_R) > 0)$ and $(\alpha > \alpha_C \Rightarrow L(x_L) - L(x_R) < 0)$. The expression for $\alpha_C$ is given by (65). From this expression, a direct calculation shows that $\alpha_C \geq 0$ if and only if $\psi_1 \leq \psi_2/(1 - \rho|1 - \psi_2|)$ when $0 < \rho \leq \frac{1}{|\psi_2-1|}$ or $\psi_1 \geq \psi_2/(1 - \rho|1 - \psi_2|)$ when $\rho \geq \frac{1}{|\psi_2-1|}$. □

To finish the proof of Theorem 9 for $\psi_1 > \psi_2$ we consider the different scenarios in Table 1. These are studied via cases that are exactly the same to the Case 1.1 to Case 1.14 for $\psi_1 < \psi_2$ but with $\alpha < \alpha_R$ replaced by $E_1$ and $\alpha > \alpha_R$ replaced by $E_2$ because when $\psi_1 > \psi_2$ it is $E_1$ and $E_2$ that determine the sign of $\frac{dL}{dx}|_{x=x_R}$. For example, for the top left-most cell in Table 1, both when $\psi_1 < \psi_2$ or $\psi_1 > \psi_2$, we have that $L$ is convex and its decreasing at $x_L$ and $x_R$, so its minimum is at $x = x_R$. As another example, the 4th and 5th rows of the first column are not possible for exactly the same reasons as when $\psi_1 < \psi_2$. Namely, when $\beta_1 < \overline{\psi} = \psi_1$ then $L$ is convex. If $\alpha > \alpha_L$ and $E_1$ holds, then by Lemma M.14 we know that $L$ is increasing at $x = x_L$ and decreasing at $x = x_R$, which is impossible for a convex function. Similarly, the 2nd and 3rd rows of the last column are not possible because when $\overline{\psi} = \psi_1 \leq \beta_3$ then $L$ is concave and if $\alpha < \alpha_L \wedge E_2$ then Lemma M.14 tells us that $L$ is decreasing at $x = x_L$ and increasing at $x = x_R$ which is not possible. We omit repeating the arguments for Case 1.2 to Case 1.14.

Above, for both $\psi_1 < \psi_2$ and $\psi_1 > \psi_2 \neq 1$, Table 1 was derived using the fact that $E_1$ holding implies that the derivative of $L$ at $x = x_R$ is negative and $E_2$ holding implies that the derivative of $L$ at $x = x_R$ is positive. It was also derived using the fact that $\alpha < \alpha_C$ implies that $L(x_L) > L(x_R)$ and that $\alpha > \alpha_C$ implies that $L(x_L) < L(x_R)$. When $\psi_2 = 1$, from Lemma M.14, we know that the derivative of $L$ at $x = x_R$ is $+\infty$, and that $L(x_L) < L(x_R)$. Hence we can keep Table 1 for $\psi_1 > \psi_2 = 1$ unchanged if in this case we set $E_1$ to be false, $E_2$ to be true, and $\alpha_C = -\infty$. □

## M.6 Proof of Theorem 10

This proof involves heavy algebraic computations. To aid the reader, this paper is accompanied by a Mathemetica file that symbolically checks the equations both in the theorem statement as well as in the proof below. This file is in the supplementary zip file, as well as

in the following Github link `https://github.com/Jeffwang87/RFR_AF`. It is called `RunMeToCheckProofOfTheorem10.nb`.

*Proof of Theorem 10.* The proof amounts to a long calculus exercise, which we shorten by some careful observations.

We first notice that $\mathcal{E}_{R_2}^\infty$ and $\mathcal{S}_{R_2}^\infty$ can both be written as function of $\omega_2 = \omega_2(\psi_2, \lambda, \mu_0, \mu_1, \mu_2)$, and we can use this to reduce the optimization problem (22) to an optimization problem over just one variable.

The variable $\omega_2$ is a function of $\mu_0, \mu_1, \mu_2 \geq 0$, which we want to optimize, and has range $[-\infty, 0]$, the value $-\infty$ being achieved when $\mu_\star^2 = \mu_2 - \mu_1^2 - \mu_2^2 = 0$.

To avoid having to deal with infinities, we make use of the Mobius transformation $x = \frac{1+\omega_2}{\omega_2-1}$, and instead work with $x$.

If we substitute $x = \frac{1+\omega_2}{-1+\omega_2}$ into the left hand side (24), and substitute (32) in the resulting expression we confirm that (24) is satisfied.

Furthermore, if we use $x = \frac{1+\omega_2}{-1+\omega_2}$ and (32) to write $x$ as a function of $\mu_0, \mu_1, \mu_2$, we can use the fact that $\mu_0, \mu_1, \mu_2 \geq 0$, to conclude that $x \in [-1, \min\{1, -1 + 2\psi_2\}]$.

Our problem is thus equivalent to solving $\min_x (1-\alpha)\mathcal{E}_{R_2}^\infty + \alpha\mathcal{S}_{R_2}^\infty$ subject to $x \in [-1, \min\{1, -1+2\psi_2\}]$. Once we know $x$, any $\mu_0, \mu_1, \mu_2$ that satisfies (24) will be a minimizer.

At this point we compute $\frac{d}{dx}[(1-\alpha)\mathcal{E}_{R_2}^\infty + \alpha\mathcal{S}_{R_2}^\infty]$ and observe that this is a rational function of $x$. The numerator is, apart from a multiplying constant, $p(x)$, and the denominator is zero if and only if $x = -1 - 2\sqrt{\psi_2}$ or $x = -1 + 2\sqrt{\psi_2}$. Both are outside of the range of $x$ unless $\psi_2 = x = 1$. When $\psi_2 = 1$ the $x = 1$ zeros of the denominator only cancel zeros of the numerator if $\tau = F_\star = 0$. In the remainder of the proof we will assume that $\tau^2 + F_\star^2 > 0$. The optimal AFs' parameters when $\tau = F_\star = 0$ can be obtained as a limit when $\tau, F_\star \to 0$. Since the zeros of the numerator and of the denominator never cancel out (assuming $\tau^2 + F_\star^2 > 0$), all of the critical points are given by $p(x) = 0$.

Now we compute the value of the derivative at the extremes of the range of $x$, namely, $x = -1, 1$, or $-1 + 2\psi_2$. The value of the derivative at $x = -1$ is $F_1^2(-1 + \alpha) < 0$, which implies that $x = -1$ is not a minimizer. The value of the derivative at $x = 1$ is $\alpha F_1^2 + \frac{\psi_2(F_\star^2 + \tau^2)}{(\psi_2 - 1)^2} > 0$, and converges to $+\infty$ when $\psi_2 \to 1$, which implies that $x = 1$ is not a minimizer. The value of the derivative at $x = -1 + 2\psi_2$ is $\alpha F_1^2 + \frac{F_\star^2 + \tau^2}{(\psi_2 - 1)^2} > 0$, and converges to $+\infty$ when $\psi_2 \to 1$, which implies that $x = -1 + 2\psi_2$ is not a minimizer. Another way to see that neither $x = 1$ nor $x = -1 + 2\psi_2$ will be a solution is to see that these choices will not satisfy the equation in (24) unless $\lambda = 0$, which never happens in this regime. This calculation implies that we can assume that $x \in (-1, \min\{1, -1 + 2\psi_2\})$, which we will assume from now on.

Finally, we show that the objective is convex in the domain of $x$, which implies that there is only one critical point – that is $p(x) = 0$ has only one solution in the domain $(-1, \min\{1, -1 + 2\psi_2\})$ – and that this critical point is a global minimum. To show that the objective is convex, we compute its second derivative, which is

$$\frac{8\psi_2\left(F_1^2\left(4\psi_2^2 + \psi_2(3(x-2)x - 5) - x^3 + 3x + 2\right) + \left(F_\star^2 + \tau^2\right)\left(4\psi_2 + 3(x+1)^2\right)\right)}{\left(4\psi_2 - (x+1)^2\right)^3}. \tag{173}$$

The minimum of denominator for $x \in [-1, \min\{1, -1 + 2\psi_2\}]$ is $\begin{cases} 4(\psi_2 - 1) & , \psi_2 > 1 \\ -4(\psi_2 - 1)\psi_2 & , 0 \leq \psi_2 \leq 1 \end{cases}$, which is always non-negative, and is zero only if $x = 1$, or $x = -1 + 2\psi_2$, which have already been excluded because they are not minimizers. Hence, for $x \in (-1, \min\{1, -1 + 2\psi_2\})$, the denominator is strictly positive.

To show that the numerator is non-negative, we only need to show that $4\psi_2^2 + \psi_2(3(x-2)x - 5) - x^3 + 3x + 2 \geq 0$ in the range of $x$. The minimum of $4\psi_2^2 + \psi_2(3(x-2)x - 5) - x^3 + 3x + 2$ for $x \in [-1, \min\{1, -1 + 2\psi_2\}]$ is $\begin{cases} 4(\psi_2 - 1)^2 & , \psi_2 > 1 \\ 4(\psi_2 - 1)^2\psi_2 & , 0 \leq \psi_2 \leq 1, \end{cases}$ which is always non-negative.

$\square$

## M.7 Proof of Theorem 11

This proof involves heavy algebraic computations. To aid the reader, this paper is accompanied by a Mathemetica file that symbolically checks the equations both in the theorem statement as well as in the proof below. This file is in the supplementary zip file, as well as in the following Github link `https://github.com/Jeffwang87/RFR_AF`. It is called `RunMeToCheckProofOfTheorem11.nb`.

*Proof of Theorem 11.* Similar to proof of Theorem 10, the majority of the proof is a long calculus exercise.

We first notice that $\mathcal{E}_{R_3}^\infty$ and $\mathcal{S}_{R_3}^\infty$ can both be written as function of $\omega_1$ and $\zeta^2$.

The variable $\omega_1$ is a function of $\mu_0, \mu_1, \mu_2 \geq 0$, and has range $[-\infty, 0]$, the value $-\infty$ being achieved when $\mu_\star^2 = \mu_2 - \mu_1^2 - \mu_2^2 = 0$.

To avoid having to deal with infinities, we make use of the Mobius transformation $x = \frac{1+\omega_1}{\omega_1 - 1}$, and instead work with $x$. If we use $x = \frac{1+\omega_1}{\omega_1 - 1}$ and (35) to write $x$ as a function of $\mu_0, \mu_1, \mu_2$, we can use the fact that $\mu_0, \mu_1, \mu_2 \geq 0$, to conclude that $x \in [-1, \min\{1, -1 + 2\psi_1\}]$.

Our problem is thus equivalent to solving $\min_x (1-\alpha)\mathcal{E}_{R_3}^\infty + \alpha \mathcal{S}_{R_3}^\infty$ subject to $x \in [-1, \min\{1, -1+2\psi_1\}]$ and $\zeta^2 \geq 0$.

Let $L = (1-\alpha)\mathcal{E}_{R_3}^\infty + \alpha\mathcal{S}_{R_3}^\infty$. With a direct calculation we can check the following. The derivative $\frac{dL}{dx}|_{x=-1}$ is always negative (recall we are assuming $\alpha < 1$) which implies that there is no local minimum at $x = -1$. The derivatives $\frac{dL}{dx}|_{x=1}$ and $\frac{dL}{dx}|_{x=-1+2\psi_1}$ are always positive (recall that we are assuming $\psi_1 > 0$ in addition to $\alpha < 1$), which implies that there is no local minimum at either $x = 1$ or $x = -1 + 2\psi_1$. Therefore, we know that the minimizer has $x \in (-1, \min\{1, -1 + 2\psi_1\})$. For $x \in (-1, \min\{1, -1 + 2\psi_1\})$, the derivative $\frac{dL}{d(\zeta^2)}|_{\zeta^2=0}$ is always negative (when $\alpha < 1$), which implies that there is no local minimum at $\zeta^2 = 0$. We thus know that the minimizer of $L$ is in the interior of the domain for $x$ and $\zeta^2$ and hence it can be found via $\nabla L = 0$, where the gradient is with respect to $x$ and $\zeta^2$.

The remainder of the proof considers a few different cases depending on the value of $\alpha$ and $\psi_1$.

**Case when $\alpha = 0$:**

In this case $L = \mathcal{E}_{R_3}^\infty$ and $\frac{dL}{d(\zeta^2)} = \frac{F_1^2(x+1)^2}{\zeta^4((x+1)^2 - 4\psi_1)}$. The only way that $\frac{dL}{d(\zeta^2)} = 0$ is if $\zeta^2 \to \infty$ (we already know that $x \neq 1$), which corresponds to $\mu_\star \to 0$.

As we noted in the beginning, $L$ is a function of $\omega_1$ and $\zeta^2$, and since $\omega_1$ is a function $\zeta^2$ and $\mu_1^2$ and $\zeta^2$ is a function of $\mu_1^2$ and $\mu_\star^2$, we know that $L$ is a function of $\mu_1^2$ and $\mu_\star^2$. We express $L$ in these variables and compute $\frac{dL}{d(\mu_1^2)}$ when $\mu_\star \to 0$. We get that

$$\frac{dL}{d(\mu_1^2)} = -\frac{2F_1^2\lambda^2\psi_1^3}{\left(2\mu_1^2\psi_1\left(\lambda - \mu_1^2\right) + \psi_1^2\left(\lambda + \mu_1^2\right)^2 + \mu_1^4\right)^{3/2}}. \tag{174}$$

By minimizing the denominator with respect to $\mu_1 \geq 0$, we conclude that its minimum is $\lambda^2\psi_1^2 > 0$, which implies that $\frac{dL}{d(\mu_1^2)} < 0$, which implies that to achieve the minimum $L$ one must have $\mu_1 \to \infty$. In this case, if we express $L$ as a function of $\mu_1$ and $\mu_\star$ and take $\mu_1 \to \infty$ and $\mu_\star \to 0$, we get $L \to F_s^2$.

**Case when $\psi_1 = 1 \wedge 0 < \alpha \leq \frac{1}{4}$:**

In this case $\frac{\mathrm{d}L}{\mathrm{d}(\zeta^2)} = \frac{(\alpha-1)F_1^2(x+1)^2}{(x-1)(x+3)\zeta^4}$. The only way that $\frac{\mathrm{d}L}{\mathrm{d}(\zeta^2)} = 0$ is if $\zeta^2 = \infty$ (recall that $x \neq 1$, $\alpha < 1$ and $F_1 > 0$) which corresponds to $\mu_\star = 0$.

As we noted in the beginning, $L$ is a function of $\omega_1$ and $\zeta^2$, and since $\omega_1$ is a function $\zeta^2$ and $\mu_1^2$ and $\zeta^2$ is a function of $\mu_1^2$ and $\mu_\star^2$, we know that $L$ is a function of $\mu_1^2$ and $\mu_\star^2$. We express $L$ in these variables and compute $\frac{\mathrm{d}L}{\mathrm{d}(\mu_1^2)}$ when $\mu_\star \to 0$. We get that

$$\frac{\mathrm{d}L}{\mathrm{d}(\mu_1^2)} = \frac{2F_1^2\lambda^2\left(\frac{4\alpha\lambda(\lambda+4\mu_1^2)}{\left(\sqrt{\lambda(\lambda+4\mu_1^2)}+\lambda\right)^2} - 1\right)}{\left(\lambda\left(\lambda + 4\mu_1^2\right)\right)^{3/2}}. \tag{175}$$

By maximizing the numerator with respect to $\mu_1^2 \geq 0$, we conclude that its value is always strictly smaller than $2F_1^2\lambda^2(-1+4\alpha) \leq 0$ for any finite $\mu_1$. Hence, $\frac{\mathrm{d}L}{\mathrm{d}(\mu_1^2)} < 0$, which implies that the minimum is achieved only when $\mu_1 \to \infty$. In this case, if we express $L$ as a function of $\mu_1$ and $\mu_\star$ and take $\mu_1 \to \infty$ and $\mu_\star \to 0$, we get $L \to \alpha F_1^2 + (1 - \alpha)F_s^2$.

**Case when $\psi_1 = 1 \wedge \frac{1}{4} < \alpha < 1$:**

This case is very similar to the previous case. The only difference being that, because $\frac{1}{4} < \alpha < 1$, when we solve

$$\frac{\mathrm{d}L}{\mathrm{d}(\mu_1^2)} = \frac{2F_1^2\lambda^2\left(\frac{4\alpha\lambda(\lambda+4\mu_1^2)}{\left(\sqrt{\lambda(\lambda+4\mu_1^2)}+\lambda\right)^2} - 1\right)}{\left(\lambda\left(\lambda + 4\mu_1^2\right)\right)^{3/2}} = 0, \tag{176}$$

we now get two possible solutions, namely,

$$\mu_1^2 \in \left\{\frac{-4\alpha^2\lambda + 3\alpha\lambda + \left(-\sqrt{\alpha}\right)\lambda}{16\alpha^2 - 8\alpha + 1}, \frac{-4\alpha^2\lambda + 3\alpha\lambda + \sqrt{\alpha}\lambda}{16\alpha^2 - 8\alpha + 1}\right\}. \tag{177}$$

First notice that if $\lambda = 0$, both expressions give $\mu_1 = 0$. Let us assume now that $\lambda > 0$. If we maximize the first expression with respect to $\frac{1}{4} < \alpha < 1$, we conclude that its value is always smaller than $-((3\lambda)/16) < 0$, which implies that it is not a valid solution in the range $\mu_1^2 \geq 0$. If we minimize the second expression with respect to $\frac{1}{4} < \alpha < 1$, we conclude that its value is always non-negative, which implies it is a valid solution in the range $\mu_1^2 \geq 0$. We therefore conclude that the second expression is the only stationary point of $L$ in the range $\mu_1^2 \geq 0$ whether $\lambda = 0$ or not.

Given that this is the only stationary point of $L$ in the range $\mu_1^2 \geq 0$, and given that the derivative $\frac{\mathrm{d}L}{\mathrm{d}(\mu_1^2)} \leq 0$ at $\mu_1 = 0$ (which can be checked via substitution), we conclude that the critical point must be a global minimum.

If we substitute the optimal values for $\mu_1^2$ and $\mu_\star^2 = 0$ in $L$, we get $L = \left(4\sqrt{\alpha} - 1 - 3\alpha\right)F_1^2 + (1-\alpha)F_s^2$.

**Case when $\psi_1 \neq 1$:**

In this case $\frac{\mathrm{d}L}{\mathrm{d}(\zeta^2)} = \frac{(\alpha-1)F_1^2(x+1)^2}{\zeta^4\left(4\psi_1-(x+1)^2\right)}$. As before, we start from knowing that the minimizer cannot be at the boundary of the domain. Therefore, since $x \neq 1$, the only way that $\frac{\mathrm{d}L}{\mathrm{d}(\zeta^2)} = 0$ is if $\zeta^2 = \infty$, which corresponds to $\mu_\star = 0$.

If we substitute $x = \frac{1+\omega_1}{\omega_1-1}$ into the left hand side of the last equality in (25), namely, $\mu_1^2(-1 + 2\psi_1 - x)(-1 + x) + 2\lambda\psi_1(1 + x)$, and let $\mu_\star \to 0$, we get $0$, which confirms the condition on $x$ when $\psi_1 \neq 1$.

We take $\zeta \to \infty$ in $L$ to obtain,

$$L = F_1^2 \left( \frac{\psi_1(x-1)^2}{4\psi_1 - (x+1)^2} + \alpha x \right) - (\alpha - 1)F_\star^2, \tag{178}$$

$$\frac{\mathrm{d}L}{\mathrm{d}x} = F_1^2 \left( \alpha + \frac{4\psi_1(x-1)(2\psi_1 - x - 1)}{\left((x+1)^2 - 4\psi_1\right)^2} \right), \tag{179}$$

$$\frac{\mathrm{d}^2L}{\mathrm{d}x^2} = \frac{8F_1^2\psi_1 \left(4\psi_1^2 + \psi_1(3(x-2)x - 5) - x^3 + 3x + 2\right)}{\left(4\psi_1 - (x+1)^2\right)^3}. \tag{180}$$

If we minimize $\frac{\mathrm{d}^2L}{\mathrm{d}x^2}$ over $x \in [-1, \min(1, -1 + 2\psi_1)]$ and $\psi_1 \geq 0$, we obtain $F_1^2/8 > 0$. This shows that our objective $L$ is strictly convex in the range $x \in [-1, \min(1, -1 + 2\psi_1)]$ and hence there is only one solution to $\frac{\mathrm{d}L}{\mathrm{d}x} = 0$.

The rational function $\frac{\mathrm{d}L}{\mathrm{d}x}$ has a denominator which is zero only if $x = -1 - 2\sqrt{\psi_1}$ or $x = -1 + 2\sqrt{\psi_1}$, both of which are outside the valid range for $x$. Hence the unique solution to $\frac{\mathrm{d}L}{\mathrm{d}x} = 0$ comes from the zeros of the numerator of $\frac{\mathrm{d}L}{\mathrm{d}x}$. This numerator is (a constant times) a polynomial in $x$ whose coefficients are described in Theorem 11. □

## N EXPERIMENT ON A REAL DATASET

It is tempting to extrapolate our theory to practice. The scope of validity of our claims is rigorously stated in our theorems' assumptions and one should be cautions not to claim their applicability beyond this scope. In particular, we are not attempting to improve on existing empirical techniques to design AFs but rather seek a better understanding of an already popular model, the RFR model. Namely, we want to understand the effect that using optimal AFs has on the RFR model. Within the context of designing good, or optimal, AFs for practical settings with empirical/heuristic methods we refer the reader to Section C. Nonetheless, here we test some of our more general conclusions on real data. This appendix is referenced in the main text in Section 3.3. In this section, the data, and the fact that we do not work with infinite dimensions, are the only deviations from our theoretical setup. In particular, we work with an RFR model.

We use the MNIST data Deng (2012) to train an RFR model that approximates a function $f$, our ground truth object, defined as follows. For a given digit image $x$ with class $c \in \{0, 1, \ldots, 9\}$, we define $f(x) = -5 + c/9$. Note that in the RFR model we are working with regressions, not classification. The MNIST data set has input dimensions $d = 28 \times 28 = 784$. For the test set we use 10000 random samples.

In Figure 4 we plot the test error $\mathcal{E}$ has a function of $\psi_1/\psi_2 = N/n$ when we have $n = 4000$ train samples and when the number of features $N$ ranges from 1 to 14250. Training is done with $\lambda = 10^{-7}$. We do so in three different settings: (1) the AF is a fixed linear function; (2) the AF is a fixed ReLU; (3) the AF is a numerically optimized linear function. This AF is optimized as follows. For each value of $\psi_1/\psi_2$, we run a Bayesian optimization subroutine that minimizes the test error across all possible linear AFs. We note that, despite the fact that with a linear AF our model is linear, optimizing the test error via linear AFs is different from optimizing the weights in the second layer during training. In this third setting, for each value of $\psi_1/\psi_2$, we are working with a different AF, which is why we use the set notation $\{\cdot\}$ around $\sigma$ in Figure 4.

We observe that we see a double descent curve phenomenon also for MNIST. This was previously known Belkin et al. (2019a), as it was also previously known that double descent curves appear for more complex data sets and neural architectures, e.g. Nakkiran et al. (2021). Unlike for the RFR theory, the interpolation threshold is not at $\psi_1/\psi_2 = 1$, but it around $\psi_1/\psi_2 = 2$. In this practical setting, and consistently with what we stated in our main observations for our theoretical setting, using different AFs affects the double descent curve phenomenon. In particular, using linear optimal AFs (one for each $\psi_1/\psi_2$) can beat using a single ReLU function and seems to destroy the double descent curve phenomenon.

The code to produce Figure 4 is in the following Github link: `https://github.com/Jeffwang87/RFR_AF`. This code is also available in the supplementary zip file provided. To

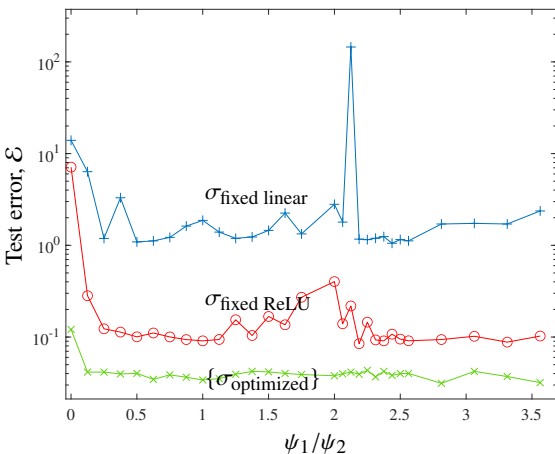

Figure 4: Learning a function from the MNIST data set also produces a double descent curve, i.e. the test error decreases as the model's complexity increases, then it increases until the *interpolation threshold*, which is around $\psi_1/\psi_2 = 2$, and then it decreases against past the *interpolation threshold*. By optimizing the AF this phenomenon disappears. The meaning of this figure is related to the meaning of Figure 1-(A) in the main text.

generate the plot run the file named `RunMeToGenerateFigure_4.m`. It runs using Matlab 2020b. We ran it using a MacBook Pro with 2.6 GHz 6-Core Intel Core i7 and 32 GB 2667 MHz DDR4. In this machine it takes about 10 hours to run.

In Figure 5 we plot the test error $\mathcal{E}$ has a function of $\lambda$ when $\psi_2 = 10$, when we have $n = \psi_2 d$ train samples, and when the number of features is very large, namely, $N = 10000$. We do so in two different settings: (1) the AF is a fixed ReLU; (2) the AF is a numerically optimized quadratic function. This AF is optimized as follows. For each value of $\lambda$, we run a Bayesian optimization subroutine that minimizes the test error across all possible quadratic AFs.

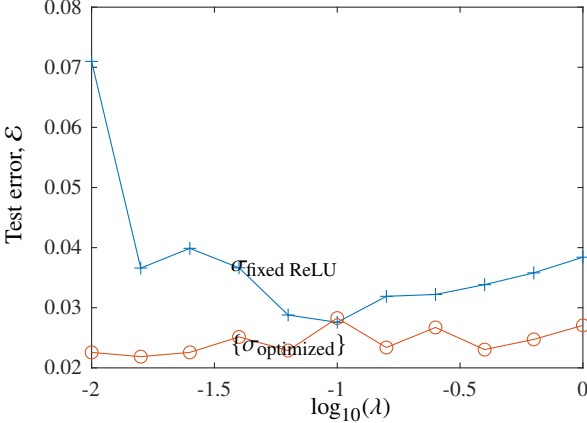

Figure 5: Learning a function from the MNIST data set using the RFR model can be improved by selecting the appropriate ridge regularization parameter $\lambda$, around $10^{-1}$ in the plot. If we are not careful about this choice, but instead we use an optimized AF, we can achieve similarly good performance. The meaning of this figure is related to the meaning of Figure 1-(C) in the main text.

We observe that choosing an optimized AF and being "careless" about the choice of regularization leads to as good results as using a ReLU and optimizing $\lambda$, which is the common practice.

The code to produce Figure 5 is in the following Github link: `https://github.com/Jeffwang87/RFR_AF`. This code is also available in the supplementary zip file provided. To generate the plot run the file named `RunMeToGenerateFigure_5.m`. It runs using Matlab

2020b. We ran it using a MacBook Pro with 2.6 GHz 6-Core Intel Core i7 and 32 GB 2667 MHz DDR4. In this machine it takes about 3 hours to run.

## O  EXPERIMENTAL RESULTS FOR DIFFERENT RANDOM FEATURES INITIALIZATION

Our results assume that the features in the RFR model are sampled i.i.d. uniform on the $(d-1)$-dimensional sphere of radius $\sqrt{d}$.

In this section, we numerically examine if two other initializations of $\Theta$ lead to similar, or different, asymptotic mean squared test error. Specifically, we initialize $\Theta$ with either Xavier initialization (Glorot & Bengio, 2010) or Kaiming initialization (He et al., 2015), and compare the resulting error curve ($L$ when $\alpha = 0$) with the curve for the original initialization for the three different regimes in our paper.

If the new error curves agree with the ones for the original initialization for some regime, we take that as evidence that our conclusion might hold for these initializations and that regime as well.

In Figure 6-(A) and (D), we see that in regime 1 and when $\alpha = 0$, there is a agreement between the three initializations. However, this is not the case for regimes 2 (Plot (B) and (E)) or 3 (Plot (C) and (F)). In regime 3, it is unclear if Xavier and Kaiming initialization agree for large values of $\lambda$.

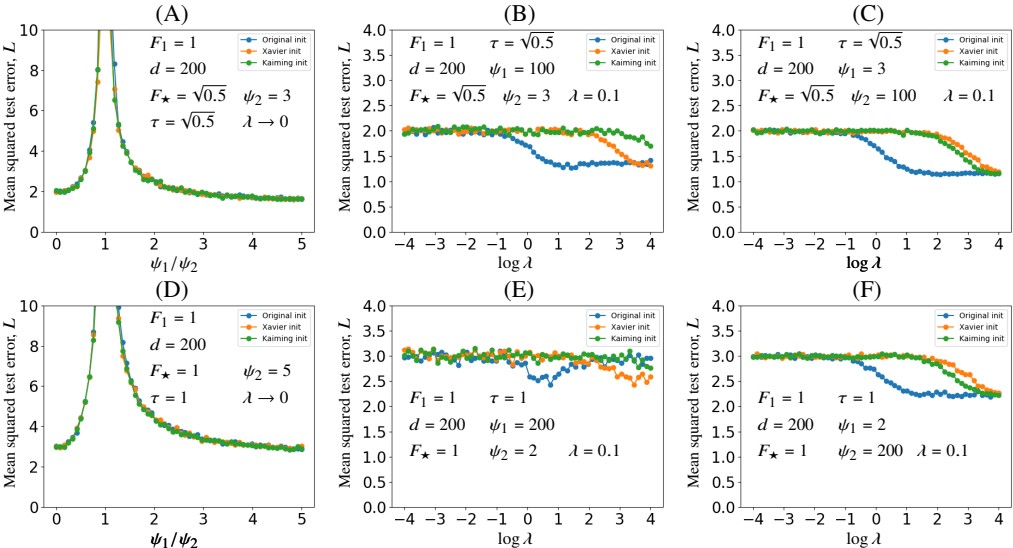

Figure 6: Plot (A), (B), and (C) are generated under the condition when $F_1 = 1$, $d = 200$, $F_\star = \sqrt{0.5}$, and $\tau = \sqrt{0.5}$. Plot (D), (E), and (F) are generated under the condition when $F_1 = 1$, $d = 200$, $F_\star = 1$, and $\tau = 1$. (A) shows the error curve for regime 1 (ridgeless-limit regime) when $\lambda \to 0$ and $\psi_2 = 3$. (B) shows the error curve for regime 2 (over-parameterized regime) when $\psi_1 = 100$, $\psi_2 = 3$, and $\lambda = 0.1$. (C) shows the error curve for regime 3 (large-sample regime) when $\psi_1 = 3$, $\psi_2 = 100$, and $\lambda = 0.1$. (D) shows the error curve for regime 1 (ridgeless-limit regime) when $\lambda \to 0$ and $\psi_2 = 5$. (E) shows the error curve for regime 2 (over-parameterized regime) when $\psi_1 = 200$, $\psi_2 = 2$, and $\lambda = 0.1$. (F) shows the error curve for regime 3 (large-sample regime) when $\psi_1 = 2$, $\psi_2 = 200$, and $\lambda = 0.1$.

