# OpenReview forum: "Optimal Activation Functions for the Random Features Regression Model"
_ICLR.cc/2023/Conference — ICLR 2023 poster_

### Official Review · Reviewer_7nWY · 2022-10-24

**Confidence:** 3
**Correctness:** 3
**Technical Novelty And Significance:** 3
**Empirical Novelty And Significance:** Not applicable
**Recommendation:** 8

**Clarity, Quality, Novelty And Reproducibility:**

The paper is very well written and clear.
I can't really comment much on novelty given that I'm not particularly familiar with the prior work in the area.
It certainly seems to be an original result though.


## Some typos or minor edit recommendations
1. [Page 1, second paragraph of section 1, the line that starts with "optimize AFs have"] missing the word "to" before "optimize"
1. [Across the entire paper] Please use a symbol other than O for the overall loss. How about $L$?
1. [Page 3, just before the bullet points] Why mention the last paper in the list was published at ICLR last year. This is odd to point out, no? The other papers in the list were published at NeurIPS, ICML, and The Annals of Statistics; all prestigious in ML.
1. [Page 3, just before the bullet points] The very last sentence in this paragraph ("Although theoretical...") reads really weirdly. I don't know what you're trying to say, but I don't think it's working. Consider just cutting the line, or rewrite it from scratch?
1. [Page 4, paragraph after equation 12] Can you clarify why $F_{\star}$ is like the norm of the noise? The expectations in the definition of $\Sigma_d$ already looks pretty odd, so I don't understand the intuition you give about what $F_{\star}$ is doing.
1. [Page 5, theorems 2-6] consider pushing all of these polynomials to the appendix, like you do for later theorems. You could even write something like
>$$E_{R_2}^{\infty} = \lim \lim E = F_1^2 p_1 + (\tau^2 + F_\star^2) p_2 + F_{\star}^2$$ where $p_1$ is a rational function in $\zeta, \psi,$ and $\bar\lambda$, and where $p_2$ is a rational function in $\zeta,\psi_2$ and $\bar\lambda$. Both rational functions are shown in Appendix XYZ.
1. [Page 7, equations 27 and 29] The formatting on where the equation number are seems to be broken. Also, you can probably just get rid of these equation numbers? And probably most of the equation markers in this paper could be removed? Y'all's choice though.
1. [Page 7, remark 3] When you say $\psi \in \{A,B\}$, what are $A$ and $B$?
1. [Page 7, last paragraph] Isn't this trivial that the choice of scaling in the AF won't impact the overall error in the unregularized case.... because it's unregularized so the ridge-less regression model will just compensate?
1. [Page 8, proof sketch] "which turns out to be" should be "which *turn* out to be"
1. [Page 9, Figure 1] Isn't $\rho$ the symbol for SNR, and not $\tau$? Why not just actually use the symbol $\rho$ here?


**Strength And Weaknesses:**

The paper is long, and the detailed math is....pedantic, but the key takeaways are really neat and the bulk of the math is buried in the appendix (as it should be).

---

Let me first note that I'm not an expert on anything in this paper, especially the deep learning theory literature. I can't really speak to novelty or significance, so I'll leave that to the other reviewers. I'll just give my impressions, as a relative outsider who can read a nice proof sketch and enjoy a good paper.

This paper was pretty fun to read. It's got a precise statistical model that's been used to study neural nets in the past. The paper's got really precise results on the shape of optimal activation function. The proofs are long, but the proof sketches in the body of the paper are short and sweet and clear. The paper's got really nifty implications of these optimal functions. These are the key strengths of the paper. I recommend publishing the paper.

This paper kinda feels like a shoe-in to get accepted. I only really see two potential weaknesses in this paper:
1. The mathematical choice of norms and objective function and regularization and generative model are all very arbitrary. These are arbitrary, and the authors notes these are arbitrary but have some foundation in recent research (e.g. bottom of page 3, top of page 4; bottom of page 9). It's a theoretical exploration and these choice are arbitrary so they have to analyze _something_. It's a weakness, but not a serious one.
1. There's no way for someone to review all of the math in the appendix in time. This is an issue with the broader peer review process for ML though. Not the fault of the paper in the slightest.

The paper is well written, intuition is regularly given, and for a paper with such mathematically precise results in so many different cases of analysis, the huge list of optimal activation functions is broken down in an impressively not-unapproachable way.

**Summary Of The Paper:**

The paper designs optimal activation functions for the Random Feature Regression model: a special case of a two layer neural network. Here, consider a two-layer neural network with inputs of dimension $d$, a second layer of dimension $N$, and a single output node. The mapping from the input to the hidden layer is generated as a random matrix, passed through a nonlinearity function we can pick, and Ridge regression is used to learn a linear combination of the hidden layer parameters and predict the output value.

This paper studies proves what activation functions are optimal in various limits as the problem grows larger. "Optimal" is characterized in a variety of ways, like balancing mean squared error of the solution against the flatness of the optimization landscape, or by picking regularization criteria on the activation functions (so when there's many possible optimal activation functions, we still typically have a unique solution). The paper always studies the limit as $d \rightarrow \infty$, with analysis specialized to three cases:
1. _Unregularized Case_: The limit as $d \rightarrow \infty$ and $\lambda \rightarrow 0$ (the regularization parameter), but $\frac{d}{n}$ and $\frac{d}{N}$ remain constant
1. _Way Overparameterized Case_: The limit as $d \rightarrow \infty$ and $\frac{d}{N} \rightarrow 0$, but $\frac{d}{n}$ and $\lambda$ remain constant
1. _Large Sample Complexity Case_: The limit as $d \rightarrow \infty$ and $\frac{d}{n} \rightarrow 0$, but $\frac{d}{N}$ and $\lambda$ remain constant

In all cases generated by all combinations of these parameters, exact characterizations of the optimal activation function are given. The details and notation get a bit laborious, but the theory provides a nice model for understanding the value of different activation functions. Like in the limit when we have a huge number of samples, we should just use a linear activation function, making the overall neural net linear. Or in showing that choosing the right activation function can completely avoid double descent, whereas the same data with a ReLU activation function would suffer from the spike in double descent.


**Summary Of The Review:**

Nicely intuitive and very well written paper.
I don't know how significant it is, since I don't especially work in this area.
Still seems like it should be published imo.

---

> ### Author Response · Authors · 2022-11-13
> **Thank you**
>
> We thank you for your time in reading our paper and providing us with feedback.
>
> We will fix all typos and incorporate all edit suggestions.
>
> We appreciate the positive outlook on our paper.  We appreciate the reviewer's honesty in admitting that the reviewer cannot judge the novelty of our work. As reviewer 7nWY can read above, no reviewer questions the novelty of our work.
>
> Reviewer 1hhM has some questions about the motivation/importance of our work given that sometimes a property known as “gaussian equivalence” holds. When this property holds (and it has only been proved to hold only for very specific setups) it makes a few of our conclusions intuitive (or less surprising in the opinion of reviewer 1hhM) but it does lead to any precise mathematical statements and hence does not allow anyone to shortcut our contribution. Furthermore, as we explain to reviewer 1hhM, there are several conclusions from our theory that expose the big limits of these intuitions. We believe and hope that our reply to reviewer 1hhM clarifies the reviewer’s questions.
>
>
> Reviewer riDW asks us to comment on the implications of this “gaussian equivalence”, which we have agreed to do. This will amount to moving a polish version of the discussion with the reviewer on OpenReview to the paper, a trivial tasks. We will make space for this by moving some of the formulas in Theorem 9 to the appendix. This reviewer does not connect the current lack of discussion to a reduction in significance. The other concern of reviewer riDW is our work building on previous work. All papers build on something and we do not think it is fair to be penalized for it. As we explain to reviewer riDW, nothing in our 38 pages of work can be derived trivially from previous work. None of our proofs are technically simple. Although it is true that there is a not single technical “aha” moment on which our proofs rely that could be repurposed by other theoreticians, there are many steps and insights buried in our proofs that make our work more than just turning the crank. We believe and hope that our reply to reviewer riDW addresses these points.
>
> Reviewer X96N only repeats what we readily admit in our paper: that we do not want to focus on empirical results and propose methods to find good AF in practice. Others have done so before us. However, we are the first to study optimal AFs having in mind explicit proven theoretic performance of different choices of AFs. We do so by focusing on the RFR on the asymptotic proportional regime, a very important setup from a theoretical perspective because it accurately captures (compared to empirical observations) phenomena like the double-descent curve that other models, such as linear regression, do not.
>
> We hope that by reading this short summary of other reviewers comments, and perhaps, if time permits, it would best to read their full comments and our full replies, the reviewer 7nWY is convinced that the score of 8 given is indeed a just score.

---

> > ### Comment · Reviewer_7nWY · 2022-11-17
> > **Hey sorry for the late reply**
> >
> > Hi, thanks for the summary of the other reviewers' topics.
> >
> > Unfortunately I have neither the time nor energy to read through those full and robust discussions, but your summary here sounds reasonable I guess.
> >
> > Assuming that you writing is a fair representation of those full and robust discussions (_again, I haven't read them_), your arguments align well with my ideals
> > - Even if some qualitative results follow from prior work, new techniques that circumvent those results is nice and exciting, especially when you further provide new results outside of where the prior work applies
> > - Novelty is hard to comment on, and one slightly lazy way we theory folk tend to measure novelty is by the length of papers (hence part of why major TCS conferences have long papers). 38 pages of technical work that relates to ODEs and orthogonal polynomials sound neat
> > - ML papers need not have experimental sections. Theory papers have value beyond immediately suggesting what a practitioner should do tomorrow. Simply understanding the role and performance of optimal AFs and contrasting optimal AFs in different settings is compelling, because AFs are poorly understood afaik. The four bullet points on the bottom of page 1 are great examples of this.

---

> ### Author Response · Authors · 2022-11-16
> **Follow up with reviewer 7nWY**
>
> Dear reviewer 7nWY,
>
> We have not heard back from you yet. If possible, could you please acknowledge our responses and let us know if there is anything that still needs further clarification?
>
> Thank you

---

### Official Review · Reviewer_X96N · 2022-10-24

**Confidence:** 2
**Correctness:** 4
**Technical Novelty And Significance:** 2
**Empirical Novelty And Significance:** 2
**Recommendation:** 6

**Clarity, Quality, Novelty And Reproducibility:**

Overall, the paper is well-organized and clearly written. The results are technically sound and novel.

**Strength And Weaknesses:**

$\textbf{Strength}$

1. This is a novel contribution that provides insights into the understanding of finding optimal activation functions for random features regression models in the asymptotic regime.

2. This paper provides some interesting observations on the benefit of finding optimal activation functions (and figure out whether they are linear or nonlinear) from the theoretical perspective.

3. Codes are provided.


$\textbf{Concerns}$

1) How to use these theoretical findings in practice? e.g., how to choose $\alpha$;

2) How to combine the optimization problems (2) and (8)?

3) The theoretical results only apply to shallow architecture, which may restrict their significance.

4) All theorems in Section 3.1 are the special cases of Mei&Montanari (2022), which are essential for the main results in Sections 4.1 and 4.2.

Minor typos:
1) Below (23), Guass error should be Gauss error

**Summary Of The Paper:**

The paper studies random features ridge regression where the data $\{(x_i, y_i)\}_{i=1}^n$ is generated from a noisy nonlinear model $y_i=f_d(x_i)+\epsilon_i$ with the covariates $x_i$ i.i.d. uniformly sampled from $\mathbb{S}^{d-1}(\sqrt{d})$.

The estimator is given by $f_{a,\Theta}(x)=\sum_{i=1}^{N}a_i\sigma(\langle\theta_i, x\rangle/\sqrt{d})$ with $\sigma$ being the activation function and $\Theta\in\mathbb{R}^{N\times d}$ with $i$th row $\theta_i\in\mathbb{R}^d$ satisfying $||\theta_i||=\sqrt{d}$, where $\theta_i$ is i.i.d. uniform on $\mathbb{S}^{d-1}(\sqrt{d})$. The authors aim at finding the optimal activation function $\sigma$ in the asymptotic regime by minimizing a linear combination of the test error and sensitivity of the random features regression models.

**Summary Of The Review:**

This work provides some interesting insights on the understanding of finding optimal activation functions in the random features regression models, which should be of interest to the ICLR community. However, I am not quite sure whether these theoretical findings can benefit the design of learning algorithms in practice, esspecially for the deep architecture case.

---

> ### Author Response · Authors · 2022-11-13
> **Response to Concern 1 "How to use these theoretical findings in practice?..."**
>
> Our work focuses on the theoretical understanding of the effect that optimal activation functions have on RFR models. Within this context, our work produces interesting, novel, and non-trivial results. We clearly state in our paper that our work distinguishes itself from empirical work regarding the design of AF (See Introduction, Related Work, and Appendix J). This is not an accident or a limitation but a deliberate choice we made.
>
> We remind the reviewer that per the review guidelines (https://iclr.cc/Conferences/2023/ReviewerGuide) the ICLR community values a much more *diverse* set of papers than simply those that achieve or help achieve empirical state-of-the-art results.
>
> Alpha is a parameter that should be chosen by the user. There is no such thing as “the best alpha”. For each alpha, however, and once the other problem parameters are fixed, there is a “best AF”, which might, or not, be linear.

---

> ### Author Response · Authors · 2022-11-13
> **Response to Concern 2 "How to combine the optimization problems (2) and (8)?"**
>
> Optimization (2) and (8) are two different problems. Problem (2) aims to find the optimal weights for the second layer of the RFR model and problem (8) aims to find the shape of optimal AFs. The parameters found by solving (8) go into the parametric equations found by solving (2).

---

> ### Author Response · Authors · 2022-11-13
> **Response to Concern 3 "The theoretical results only apply to shallow architecture, which may restrict their significance."**
>
> We are aware of this and we comment on this in Section 3, bullet point 4. Given the current state of the theory on neural networks, it is not possible to prove rigorous results of our type for deeper networks. Reviewer riDW agrees that “the authors are right that there are no rigorous multi-layer generalizations”.
>
> We should point out that there is a difference between “practical applicability” and “significance”. As we discuss below, we know our work has limited practical applicability but we also know that, according to the ICLR guidelines, this is not a reason to penalize our paper. As far as theoretical “significance”, we made interesting discoveries about the activation function for this model which no one had done before. Some of these are summarized in the 4 bullet points in the introduction. We also point out that, although a few of these discoveries are intuitive (based on a property known as Gaussian Equivalence , most of them are surprising  - see comment above to reviewers 1hhM and riDW).
>
> It is well known that the RFR model is interesting for the theory community despite its shallow architecture.

---

> ### Author Response · Authors · 2022-11-13
> **Response to Concern 4 "All theorems in Section 3.1 are the special cases of ..."**
>
> We do not understand the point the reviewer is trying to make. It is true that the results in Section 4.2 depend on the results in [[Mei and Montanari 2022](https://onlinelibrary.wiley.com/doi/pdfdirect/10.1002/cpa.22008)]. It is *not* true that the results in Section 4.1  depend on the results in [[Mei and Montanari 2022](https://onlinelibrary.wiley.com/doi/pdfdirect/10.1002/cpa.22008)], although we build our theory in Sec 4.1 to be compatible with the type of AF that [[Mei and Montanari 2022](https://onlinelibrary.wiley.com/doi/pdfdirect/10.1002/cpa.22008)] considers. Regardless, any paper builds on something. We made significant and novel contributions to finding the optimal activations for the RFR model. The fact that there is work before ours that we build on, in no way diminished our contribution to the community. Our contributions cannot be found or derived in any obvious way from past work.

---

> ### Author Response · Authors · 2022-11-13
> **X96N**
>
> Does the reviewer agree with our comments below? If the reviewer agrees, can the reviewer update the paper’s score? If not, can the reviewer point to what remains unanswered?
>
> Thanks

---

> ### Author Response · Authors · 2022-11-13
> **Thank you**
>
> We thank you for your time in reading our paper and providing us with feedback. We recommend you read our responses from the bottom up

---

> ### Author Response · Authors · 2022-11-16
> **Follow up with reviewer X96N**
>
> Dear reviewer X96N,
>
> We have not heard back from you yet. If possible, could you please acknowledge our responses and let us know if there is anything that still needs further clarification?
>
> Thank you

---

> > ### Comment · Reviewer_X96N · 2022-11-18
> > **Response to rebuttal**
> >
> > I would like to thank the authors for their responses and clarifications. After reading other reviewers' insightful comments, I realize that I am not sure about the importance of this specific topic. But since the authors have addressed my concerns, I will raise my score to marginally above the acceptance threshold.

---

### Official Review · Reviewer_riDW · 2022-10-24

**Confidence:** 3
**Correctness:** 3
**Technical Novelty And Significance:** 2
**Empirical Novelty And Significance:** 3
**Recommendation:** 6

**Clarity, Quality, Novelty And Reproducibility:**

- **Clarity**: As discussed above, I found that the heavy use of notation hinders the clarity of the discussion. An intuitive discussion of the results would also add to the paper.

- **Quality**: Clarity aside, the work is sound and technically correct. The algebra is cumbersome, so it is impressive the authors get sharp results in arbitrary asymptotic regime.

- **Novelty**: I am not aware of other works discussing optimal choices of activation for random features regression. Therefore, to my best knowledge the discussion and results are original. However, as noted above from a technical standpoint the results strongly rely on previous literature.

- **Reproducibility**: Code for reproducing some of the derivations is released with the paper.

**Strength And Weaknesses:**

**Strengths**:

- Given the cumbersome asymptotic formulas, getting sharp results for the optimal activation is a quite impressive *tour de force*.
- The authors make an effort to connect their discussion to more realistic settings, despite leaving this to the Appendix F.

**Weaknesses**:

- The work strongly builds on previous theoretical results, with only small novel technical contributions.
- The analysis is limited to ridge regression with Gaussian random features, despite the progress in the literature deriving similar asymptotic results for more general feature matrices, losses and regularisation.
- The discussion is sometimes hard to follow, specially because of the heavy notation which requires the unfamiliar reader to go back and forth for the definitions.

**Comments**:

- **[C1]**: A drawback of this work is that it lacks a discussion of some important intuition underlying the key conclusions. For instance, I believe some of the results in this paper can be directly understood from Gaussian equivalence of random features in the proportional regime. For some context, Gaussian equivalence builds on early spectral universality results in the kernel regression literature [El Karoui '10; Pennington, Warah '17] and is an important ingredient for the derivation of the exact asymptotics in the regression case [Mei, Montanari '22], but it also allow one to extend the exact asymptotic analysis of the RFs model to other loss functions and projection matrices, as shown in [Gerace et al. '20; Goldt et al. '22] and later proven in [Hu, Lu '20].

Gaussian equivalence states that the asymptotic statistics of the random features estimator is equivalent to the asymptotic statistics of an equivalent Gaussian problem. Employing the notation in the paper, one way of writing is:
$$
\sigma(\Theta x_{i}) \asymp \mu_{0} 1 + \mu_{1}\Theta x_{i} + \mu_{\star} z_{i}
$$
where $z_{i} \sim\mathcal{N}(0,I_{N})$ is an uncorrelated noise. Therefore, the problem considered here is asymptotically equivalent to:
$$
y_{i} = f_{d}(x_{i})+\rm{noise} = \beta_{0} + \langle\beta_{1}, x_{i}\rangle + \rm{noise}
$$
$$
\hat{y} = f(x) = \mu_{0}\langle a, 1\rangle + \mu_{1} \langle a, \Theta x\rangle + \mu_{\star}\langle a, z\rangle
$$
where in this regime the non-linear part of the target is also equivalent to additive mismatch noise. From the equivalent model perspective, it is clear that the random features is just introducing a model mismatch that hinders performance. Indeed, since the target is just a linear function in $\mathbb{R}^{d}$, the optimal predictor would just do linear regression in this space, while the projection $\Theta x\in\mathbb{R}^{N}$ is forcing the model to fit a linear function in $\mathbb{R}^{N}$. From this characterisation, it is also clear how $\mu_{\star}$ plays the role of an effective regularisation $\lambda$. However, since it also appears in the effective model mismatch noise this leads to the richer behaviour observed by the authors in Fig 1 (C) & (D).

I am not sure if the authors exploited this intuition, but I believe it would be very important to explicitly discuss it in the manuscript.

- **[C2]**: One important aspect which is missing in the discussion is how much the conclusions depend on the target function. For instance, if the target function itself was given by a random features model with features $\sigma_{\star}(\Theta x)\in\mathbb{R}^{N}$ for some activation $\sigma_{\star}$, it seems an optimal linear predictor would be obtained by matching the features $\sigma=\sigma_{\star}$. Would this substantially change the conclusion?

- **[C3]**: The heavy use of notation in the discussion can be confusing for the unfamiliar reader, who has to go back and forth to definitions in order to follow. Reminding the reader of what the different quantities mean over the discussion and the plots would be very helpful. For instance, here are some suggestions:
  - In Fig. 1, instead of having in the x-axis $\psi_{1}/\psi_{2}$, one could write directly $N/n$ .
  - Using $n/d$ and $N/d$ instead of $\psi_1, \psi_2$ could be helpful also during the discussion. Instead of $\psi_{1}<\psi_{2}$ one can say $N<n$. Of course, this is always understood asymptotically.
  - Instead of saying "In regime $R_2$", one could be redundant and say "In the highly overparametrised regime $\psi_{2}\to\infty$". Same for $\alpha = 0$: "minimising the error" or $\tau=0$: "noiseless".
  - The whole notation in Table 1 is very hard to parse. Unfortunately I don't have a suggestion on how to improve it, but from the reader perspective one just skips it.

- **[C4]**: A few comments concerning the related literature:

  - The sensitivity has been studied in the closely related setting of kernel ridge regression in [Simon et al. '21], where a non-asymptotic formula for this quantity as a function of the kernel spectrum was derived. This should coincide with the expression here in the overparametrised regime $R_{2}$ which corresponds to the kernel limit of random features.

  - In Sec. 3, point 4 the authors mention that:
    > Existing proof techniques make it very hard yet to extend our type of analysis to more than two layers or complex architectures.

    Although the authors are right that there are no rigorous multi-layer generalisations (except for not-so-interesting cases such as $\mu_1 = 0$, c.f. [Pennington, Warah '17]), in [Goldt et al. '22, Loureiro et al. '21] it was empirically shown by comparing the learning curves that Gaussian equivalence holds in multi-layer cases, including in cases when the layers $\Theta$ are pre-trained.


**References**:

[[El Karoui '10]](https://projecteuclid.org/journals/annals-of-statistics/volume-38/issue-1/The-spectrum-of-kernel-random-matrices/10.1214/08-AOS648.full) N El Karoui. *The spectrum of kernel random matrices*.  Ann. Statist. 38(1): 1-50 (February 2010). DOI: 10.1214/08-AOS648.

[[Pennington, Warah '17]](https://papers.nips.cc/paper/2017/hash/0f3d014eead934bbdbacb62a01dc4831-Abstract.html). J Pennington, P Worah. *Nonlinear random matrix theory for deep learning*. Part of Advances in Neural Information Processing Systems 30 (NIPS 2017).

[[Gerace et al. '20]](https://proceedings.mlr.press/v119/gerace20a.html) F Gerace, B Loureiro, F Krzakala, M Mézard, L Zdeborová, "Generalisation error in learning with random features and the hidden manifold model", Proceedings of the 37th International Conference on Machine Learning, PMLR 119:3452-3462, 2020.

[[Goldt et al. '22]](https://proceedings.mlr.press/v145/goldt22a.html) S Goldt, B Loureiro, G Reeves, F Krzakala, M Mezard, L Zdeborova. *The Gaussian equivalence of generative models for learning with shallow neural networks*. Proceedings of the 2nd Mathematical and Scientific Machine Learning Conference, PMLR 145:426-471, 2022.

[[Hu, Lu '20]](https://arxiv.org/abs/2009.07669): H Hu, YM Lu. *Universality Laws for High-Dimensional Learning with Random Features*. arXiv: 2009.07669 [cs.IT]

[[Simon et al. '21]](https://arxiv.org/abs/2110.03922) JB Simon, M Dickens, D Karkada, MR DeWeese. *The Eigenlearning Framework: A Conservation Law Perspective on Kernel Regression and Wide Neural Networks*, arXiv: 2110.03922 [cs.LG]

[[Loureiro et al. '21]](https://proceedings.neurips.cc/paper/2021/hash/9704a4fc48ae88598dcbdcdf57f3fdef-Abstract.html) B Loureiro, C Gerbelot, H Cui, S Goldt, F Krzakala, M Mezard, L Zdeborová. *Learning curves of generic features maps for realistic datasets with a teacher-student model*. Part of Advances in Neural Information Processing Systems 34 (NeurIPS 2021).

**Small typos**:

- Below eq. (23): *"Guass"* -> Gauss.

**Summary Of The Paper:**

This work considers the problem of finding optimal activation functions for a random features regression setting under a performance vs. sensitivity trade-off. Under the same setup as in [Mei, Montanari '22] (spherical input data, linear target function + noise, Gaussian random features projection), the starting point of the analysis is the exact formula derived in [Mei, Montanari '22] for the asymptotic generalisation error of the model in the proportional regime.

The main new technical contribution is an exact asymptotic formula for the sensitivity (squared norm of the model gradient). Combining this with the asymptotic formula for the error allow the authors to study the optimal RF activation for the different regimes of interest by minimising a convex combination of the error and the sensitivity. In particular, the authors discuss when it is better to have a linear vs. non-linear activation, and how the optimal choice of activation compares with optimal cross-validation of the $\ell_2$ penalty.

**Summary Of The Review:**

Overall, although the discussion is novel and the sharp results derived from cumbersome algebra is impressive, I think this manuscript could be considerably improved. See the points raised above for some constructive suggestions.

---

> ### Author Response · Authors · 2022-11-13
> **Response to "The main new technical contribution is an exact asymptotic formula for the sensitivity (squared norm of the model gradient)."**
>
> This is not true. This formula follows from the work of [[D'Amour et al 2021](https://www.jmlr.org/papers/volume23/20-1335/20-1335.pdf)] as we cite in our Theorem 13. Our main contribution is the study of optimal AFs for the RFR model and not the study or derivation of asymptotic performance metrics. To be specific, our contribution is the derivation of *both* the functional form of optimal AFs (Section 4.1) *and* the derivation of optimal parameters for these functional forms (Section 4.2).

---

> ### Author Response · Authors · 2022-11-13
> **Response to Weaknesses 1 "The work strongly builds on previous theoretical results, with only small novel technical contributions."**
>
> We disagree that the ratio between the amount of work that we build on and the amount of novel work advanced by us, the tour de force that the reviewer mentions, is atypical of papers accepted to ICLR or other ML top **conferences**. In our 38 pages and 500 lines of code we: (1)  provided a theory from which one can derive the optimal functional form of AF (Section 4.1). (2) showed how to explicitly extract optimal AF parameters and characterized the cases in which the AF is linear or not linear (Section 4.2). (3) used our theory to discuss several interesting results (Section 4.3). (4) did several numerical experiments (Appendix F). (5) We wrote code to reproduce all figures with a single “click to run”. (6) We wrote code to help the reader symbolically verify the heavy algebra in the theorems in Section 4.2.

---

> ### Author Response · Authors · 2022-11-13
> **Response to Weaknesses 2 "The analysis is limited to ridge regression with Gaussian random features, despite the progress in the literature..."**
>
> Question: Would the reviewer be so kind as to point out results in the literature where one can find explicit test errors and sensitivity formulas for other settings for our *same* model? The review provides a list of several papers but did not link any of these papers to this comment. Please do so if possible.
>
> Perhaps the review is talking about the work of [[Goldt et al. '22](https://proceedings.mlr.press/v145/goldt22a.html)]? This work does consider e.g. more general feature matrices and losses. However, it uses a different setup than ours. In the words of the authors, and comparing their setup with the setup that we use, in section 2.2 of  [[Goldt et al. '22](https://proceedings.mlr.press/v145/goldt22a.html)] they write:
>
> *“The main difference is the class of functions considered. Specifically, Theorem 2 [[Goldt et al. '22](https://proceedings.mlr.press/v145/goldt22a.html)] provides guarantees for any sufficiently smooth function applied to given low-dimensional projections of the features (x, c). This form of approximation is needed to justify the integro-differential equations derived in Sec. 3.1. By contrast, the RMT approach [[Mei and Montanari '22](https://onlinelibrary.wiley.com/doi/pdfdirect/10.1002/cpa.22008)] provides guarantees for a restricted set of functions applied to high-dimensional matrices derived from samples of (x, c). For example, these results provide equivalence of the empirical spectral measures of these random matrices as well as the test error associated with specific learning algorithms. The results in this paper thus neither imply previous works, nor are they, to the best of our knowledge, implied by it.”*
>
> There is also the work of [[Loureiro et al 2021](https://proceedings.neurips.cc/paper/2021/hash/9704a4fc48ae88598dcbdcdf57f3fdef-Abstract.html)]. In this work, there is also a mismatch between their setup (when proofs are provided) and the setup of our work. In particular, the results of Section 3 in [[Loureiro et al 2021](https://proceedings.neurips.cc/paper/2021/hash/9704a4fc48ae88598dcbdcdf57f3fdef-Abstract.html)]., where they apply their formulas to recover the random kitchen sink also studied by [[Song and Montanari 2022](https://onlinelibrary.wiley.com/doi/pdfdirect/10.1002/cpa.22008)] is based on a conjectured very general equivalence that has not been proved.
> We emphasize that regardless of any equivalence between models, the fact remains that extracting optimal AFs for any of these models has not been done prior to our work and constitutes a solid contribution.
>
> Note also that, although [[Goldt et al. '22](https://proceedings.mlr.press/v145/goldt22a.html)] provide conditions under which the gaussian equivalent holds, and although they provide a general procedure to compute asymptotic test errors, they do not apply this procedure to any particular situation where explicit formulas can be derived, which later could be used to compute optimal AFs explicitly like we do.
>
> Even if it is the case that other results existed for which we could follow a similar program as in this paper, the fact remains that there is only so much one can fit in a conference paper, and we believe that the review is setting unrealistic demands on an already extremely complex and lengthy paper.

---

> > ### Comment · Reviewer_riDW · 2022-11-14
> > **References**
> >
> > > *Question: Would the reviewer be so kind as to point out results in the literature where one can find explicit test errors and sensitivity formulas for other settings for our same model? The review provides a list of several papers but did not link any of these papers to this comment. Please do so if possible.*
> >
> > [[Gerace et al. '20]](https://proceedings.mlr.press/v119/gerace20a.html) considers exactly the same setting as [[Mei and Montanari '22]](https://onlinelibrary.wiley.com/doi/pdfdirect/10.1002/cpa.22008) but for a generic convex loss and fixed random features weight $\Theta$ (with well defined asymptotic spectral density in the proportional limit). Although the result in this paper is not rigorous, it was later proven in [[Hu and Lu '20]](https://arxiv.org/abs/2009.07669) and [[Dhifallah and Lu]](https://arxiv.org/abs/2008.11904) with $\Theta$ Gaussian i.i.d.
> >
> > > *Even if it is the case that other results existed for which we could follow a similar program as in this paper, the fact remains that there is only so much one can fit in a conference paper, and we believe that the review is setting unrealistic demands on an already extremely complex and lengthy paper.*
> >
> > By no means I expect you to generalise your analysis to a more general setting at this point. However, I do believe it is a fair constructive criticism to inquire on the limitations of the discussion you propose here and to expect you might have antecipated some of these limitations.
> >
> > Generally speaking, my take on this "exact asymptotics" line of work is that what justifies the study of simple models (Gaussian data, random features, etc.) is not our capacity of proving or computing things, but rather what lessons we learn from the study of these models translate to more general, realistic scenarios. So my question really is: "how much are your conclusions specific to your setting - data distribution, loss function, distribution of features, etc?". Note that this is why I noted that it is a pity you left some of these questions to Appendix F.  From this perspective, I don't expect a precise quantitative answer, but rather a qualitative or numerical discussion, e.g. would changing the distribution of $\Theta$ just change the thresholds for the regimes or have a different phenomenology?

---

> > > ### Author Response · Authors · 2022-11-15
> > > **Response to "[Gerace et al. '20] considers exactly the same setting as [Mei and Montanari '22] but...."**
> > >
> > > The target functions considered by these papers are of the type $f( \beta^T x )$ while our target functions are of the type $\beta^T x + f(x)$. In both cases $f$ can be random but, functionally speaking, these are different spaces. In the first case,  $f$ has one input while in the second case d inputs. For example, consider d = 2 such that $\beta^T x = x_1 - x_2$. In one case we get $f(x_1 - x_2)$ while in the other case we get $x_1 - x_2 + f(x_1, x_2)$. The first function is always invariant to adding a constant to $x$, while the second might not. We do not dispute that it **might** be possible to prove that, asymptotically at least, models for one or the other behave similarly because of some gaussian equivalence property. However, we are not aware of this gaussian equivalence having been proved rigorously for our setting when $f$ is a nonlinear component sampled from a Gaussian process.

---

> > > > ### Comment · Reviewer_riDW · 2022-11-15
> > > > **Response**
> > > >
> > > > I agree that these targets are a priori different. However, I believe this is only cosmetic, and that one can show that the asymptotic formulas in [[Gerace et al '20]](http://proceedings.mlr.press/v119/gerace20a.html) exactly reduce to those in [[Mei and Montanari '22]](https://onlinelibrary.wiley.com/doi/pdfdirect/10.1002/cpa.22008) if one takes the square loss and the Stieltjes transform of the Marchenko-Pastur distribution, up to a mapping in the constants (e.g. $F_{\star}$)

---

> > > ### Author Response · Authors · 2022-11-15
> > > **Response to "By no means I expect you to generalise your analysis to a more general setting at this point..."**
> > >
> > > We agree that adding these comments on the limitations of our work is important. We also agree that **part** of the justification for the study of simple models is to see what lessons learnt from their study translate to more general and realistic settings. We write **part of** because even if one stays within a simplified/conceptual setting, convincing the community that these models can be analyzed analytically to produce explicit results about this or that, and showing how to by doing it, is a contribution in itself.
> > >
> > > More importantly, we disagree that if it takes one group of people to first produce analytical results and then another group of people to extrapolate and compare these results to other settings, then neither group has produced science worth publishing. This being said, we did start this line of inquiry as the reviewer pointed out, in Appendix J, and this effort should be taken into account.
> > >
> > > Given the time we have left to produce a revision (18th Nov), and the amount of change allowed before publication, we can realistically promise to mention in our Future Work a set of new experiments that can and should be done to test how much of our conclusions generalize. Namely,
> > >
> > > 1. Starting from a setting like ours but where two, or more, layers are learnt, even if just by making a single gradient step (similar to [[Ba et al 22](https://arxiv.org/pdf/2205.01445.pdf)]):
> > >
> > > * Is it possible that when learning a (non)-linear function that the best AF is linear?
> > > * Can we find optimal (non)-linear AF (after) beyond the interpolation threshold?
> > > * Does using an optimal AF destroy the double descent curve behavior?
> > > * How big is the difference between tuning regularization and tuning AFs for different (more or less linear) types of target functions
> > >
> > > 2. Starting from a setting like ours but where we change the distribution of Theta such that its entries are sampled for the Xavier distribution [[Glorot and Bengio '10](https://proceedings.mlr.press/v9/glorot10a/glorot10a.pdf)] or the Kaiming He distribution [[He et al. '15](https://arxiv.org/pdf/1502.01852.pdf)]:
> > >
> > > * Answer the same questions as above
> > >
> > > 3. Starting from the more realistic architectures and datasets studied by [[belkin et al '18](https://arxiv.org/pdf/1812.11118.pdf)] and [[Nakkiran et al 2021](https://iopscience.iop.org/article/10.1088/1742-5468/ac3a74/pdf)], that have already been used to inspect theoretical extrapolations:
> > >
> > > * Answer the same questions as above
> > >
> > > We are open to including other specific suggestions for future work if the reviewer can provide them.
> > >
> > > Even leaving the running of these experiments for future work, we can make a few statements with relative confidence, which we will also include in the paper.
> > >
> > > *Conjecture 1*: the key technique in [[Mei and Montanari '22](https://onlinelibrary.wiley.com/doi/pdfdirect/10.1002/cpa.22008)] is the use of random matrix theory. In particular, they show that if one defines the matrix $\Phi = \sigma(X^T \Theta/ \sqrt{d})$ and then computes the spectrum (distribution of eigenvalues) of the matrix $M = \Phi^T \Phi$, that this spectrum will approach the spectrum of $\tilde{M}$ obtained if  $\Phi$ was replaced by $\tilde{\Phi} = X^T \Theta / \sqrt{d} + \mu^* \times G$, where $G$ has gaussian iid entries. Because many universality results exist in random matrix theory,  we do expect that for other choices of $\Theta$ exactly the same asymptotic results will hold. The first thing to try to prove would be similar results for $\Theta$ from well-known random matrix ensembles.
> > >
> > > *Conjecture 2*: As we mentioned to reviewer 1hhM, the work of [[Misiakiewicz '22](https://arxiv.org/pdf/2204.10425.pdf)], has asymptotic error formulas derived for the same setting as in our paper but for a different regime, namely, when n ~ poly(d). In this regime, RFR can learn non-linear target functions with zero MSE see [[Misiakiewicz '22](https://arxiv.org/pdf/2204.10425.pdf)] and [[Ghorbani et al 2021](https://arxiv.org/pdf/1904.12191.pdf)]. The asymptotic formulas derived in [[Misiakiewicz '22](https://arxiv.org/pdf/2204.10425.pdf)] are almost the same as the ones in  [[Mei and Montanari '22](https://onlinelibrary.wiley.com/doi/pdfdirect/10.1002/cpa.22008)] and we believe that all of our theorems will hold exactly with a simple reparametrization of our variables.
> > >
> > > We should clarify that we can realistically mention these experiments in Future Work **in addition to** mentioning the connection with Gaussian equivalence in the Related Work and Background sections.
> > >
> > > **While we hope that the reviewer understands that we cannot, given the time constraints, promise more than this will be added to the paper, a minor but important revision to be sure, we are not idle and we are actively trying to produce new numerical experiments to satisfy the reviewer. We do not know, however, if we can present them before the 18th of November.**

---

> > > > ### Comment · Reviewer_riDW · 2022-11-15
> > > > **Interesting suggestions / conjectures**
> > > >
> > > > I thank the authors for building upon my suggestions. Those would definitively be interesting experiments.
> > > >
> > > > Regarding the conjectures:
> > > >
> > > > - Conjecture 1 is very reasonable. As we have discussed, [[Gerace et al. '20]](http://proceedings.mlr.press/v119/gerace20a.html) provides very strong numerical evidence that this is true for other matrix ensembles, at least in what concerns the equivalence of the error. If the authors could prove this, even for the square loss, it would close a gap in this literature. Note that stronger results are known in what concerns just spectral equivalence, e.g. [[Benigni and Peche '20]](https://arxiv.org/pdf/1904.03090.pdf)
> > > >
> > > > - Conjecture 2: Indeed, the asymptotic formulas in the polynomial regime are equivalent to the ones in the linear regime, but with "renormalised" parameters that depend on the regime considered. But as noted in [[Lu and Yau '22]](https://arxiv.org/pdf/2205.06308.pdf), these reparametrisation depends on the constants in the problem. Therefore, wouldn't you expect a different phenomenology than observed in this work?

---

> > > > > ### Author Response · Authors · 2022-11-17
> > > > > **Response to "Regarding the conjectures:"**
> > > > >
> > > > > We agree that it is likely that the reparametrisation we mentioned will probably real to different thresholds in our theory.
> > > > >
> > > > > We are glad that reviewer riDW appreciates our conjectures. We will include mention in the final version of our paper.

---

> ### Author Response · Authors · 2022-11-13
> **Response to Weaknesses 3 "The discussion is sometimes hard to follow, specially because.."**
>
> Since we are going to moving some more mathematical details (e.g. from Theorem 9) to the appendix to make space for the recommended discussion about gaussian equivalence, the exposition will become simpler.

---

> ### Author Response · Authors · 2022-11-13
> **Response to "Comments"**
>
> *Response to [C1]*
>
> We will provide such discussion about gaussian equivalent models, which we view as a minor revision. Notice that reviewer 1hhM also suggested that we discuss this same topic. We note, however, that the amount of specific information that one can extract from this intuition is limited and in no way allows someone to circumvent the need for our contribution unless one is satisfied with hand waving. We strongly encourage you to see the comments we have made to 1hhM that explain some of the limitations of this intuition.
>
> *Response to [C2]*
>
> We understand that recent papers study target functions of the type proposed. However, the work we build provides asymptotic formulas for a different type of non-linear target function. Basically, a linear function plus a non-linear gaussian process (a random function) whose magnitude is controlled by $F_*$. The value of $F_*$, that is, the amount of nonlinearity in our target function, does affect our conclusions, as one can see directly from Theorem 9 and its dependency on $\rho$ ($\rho$ is a relationship between the relative magnitude of the linear and non-linear components), and in particular, it can affect the optimal AF being linear or not linear. Therefore, the target function does affect our conclusions. It is however outside of the scope of our paper to study the target functions studied by other authors, e.g.  [[Goldt et al. '22](https://proceedings.mlr.press/v145/goldt22a.html)],  [[Loureiro et al 2021](https://proceedings.neurips.cc/paper/2021/hash/9704a4fc48ae88598dcbdcdf57f3fdef-Abstract.html)],  [[Hu and Lu 2020](https://arxiv.org/pdf/2009.07669.pdf)] and [[Ba et al 2022](https://arxiv.org/pdf/2205.01445.pdf)]. Given the limited scope of a conference paper, and the current length of our work, we believe that it is reasonable for the reviewer to accept that leaving this to future work is not a major problem.
>
> *Response to [C3]*
>
> We will incorporate all of your suggestions
>
> *Response to [C4]*
>
> We will refer to and comment on your provided literature and the intuitions that can be derived from gaussian equivalent models

---

> > ### Comment · Reviewer_riDW · 2022-11-14
> > **Role of the target function**
> >
> > > Therefore, the target function does affect our conclusions. It is however outside of the scope of our paper to study the target functions studied by other authors, e.g. [Goldt et al. '22], [Loureiro et al 2021], [Hu and Lu 2020] and [Ba et al 2022]. Given the limited scope of a conference paper, and the current length of our work, we believe that it is reasonable for the reviewer to accept that leaving this to future work is not a major problem.
> >
> > I thank the authors for the clarification, and agree with them that this would require substantial additional work. However, I think it would be important to add a comment on the dependence of the conclusions on the choice of target in the manuscript.

---

> ### Author Response · Authors · 2022-11-13
> **Question from authors to reviewer riDW**
>
> Given what we wrote below, and also the comments to reviewer 1hhM, and given our commitment to discuss the connection between tuning parameters in the gaussian equivalent model and tuning activation functions, can reviewer riDW agree to revise our paper’s score?
>
> Thanks

---

> ### Author Response · Authors · 2022-11-13
> **Thank you**
>
> We thank you for your time in reading our paper and providing us with feedback. We recommend reading the feedback below as well as the feedback to other reviewers. Please read our comments from the bottom up.

---

> ### Comment · Reviewer_riDW · 2022-11-14
> **Thank you for the rebuttal**
>
> Dear authors,
>
> Thank you for taking the time to address my questions and for welcoming some of the suggestions. I will iterate on the questions which I believe might require further discussion below.
>
> Regarding the score, I plan to re-evaluate it after the whole discussion period is over (including the discussion with other reviewers).
>
> Meanwhile, I kindly remind that ICLR allows for the authors to post revised versions of the manuscript during the discussion period if they wish to.

---

> > ### Author Response · Authors · 2022-11-15
> > **Thank you for your response**
> >
> > Thanks for your response. We should also mention that, before we submit a revision to our paper before the 18th of November, it would be great if the reviewer could Ok the proposed changes. For obvious reasons, we do not want to start editing and submitting revisions unless there is some clarity about what to expect.

---

> > > ### Comment · Reviewer_riDW · 2022-11-15
> > > **On the revision**
> > >
> > > I understand the authors request, and I am happy they welcomed some of my suggestions. However, I don't feel comfortable in validating any revision. I just think it is a good practice for authors to propose changes based on the feedback and the discussion, and since ICLR allows for this to happen during the revision process, why not making the best of it. But I also think that ultimately it is up to the authors to weight what suggestions they think are more relevant.

---

> > > > ### Author Response · Authors · 2022-11-17
> > > > **Response to " why not making the best of it. B"**
> > > >
> > > > We agree. We are working on it. Hopefully we can make it before the 18th Nov. Even if we cannot make it on the 18th Nov, for sure we will include these discussions in the final version (if accepted).

---

### Official Review · Reviewer_1hhM · 2022-10-29

**Confidence:** 5
**Correctness:** 4
**Technical Novelty And Significance:** 2
**Empirical Novelty And Significance:** Not applicable
**Recommendation:** 5

**Clarity, Quality, Novelty And Reproducibility:**

**Clarity & Quality:**
The main text is very compact and not easy to follow. The authors presented many theorems on different regimes and objectives, most of which are not immediately interpretable. Personally I am not too interested in how the roots of the polynomial equations are characterized, so I would suggest the authors to remove some technical contents from the main text and highlight a few important messages.

**Novelty:** The asymptotic formulae for both the test loss and the sensitivity have been computed in prior works ([Mei and Montanari 2019] [D’Amour et al. 2020]). The main technical novelty is to solve for the optimal coefficients in the activation function, which is a rather tedious computation.

**Reproducibility:** N/A.

**Strength And Weaknesses:**

## Strength

This paper contributes to the growing literature on the asymptotics of random features regression. Unlike most prior works that focused on the shape of the risk curve (double descent), the current submission considers the design of optimal activation function, which is a new application of the precise error formulae.

## Weaknesses

My main concern is on the motivation and implications of the analysis.

1. Under the setting of [Mei and Montanari 2019], random features model in the proportional regime can only learn linear functions on the input. Specifically, the test error of the RF model only depends on the activation function through the first two Hermite coefficients, and is equivalent to that of a Gaussian linear model -- this is often refereed to as the Gaussian equivalence property [Hu and Lu 2020] [Loureiro et al. 2021] [Montanari and Saeed 2022].
Consequently, designing the activation function is equivalent to tuning the magnitude of Gaussian noise added to the features (this can be interpreted as implicit ridge regularization), which I do not find very meaningful because the resulting model cannot outperform linear regression on the input (see [Ba et al. 2022] Section 4).

2. Given the Gaussian equivalence property, it is not surprising that the optimal activation function can be linear, which reduces the RF model to linear regression on the input in certain regimes. Similarly, due to the self-induced ridge regularization of the nonlinear activation function, it is intuitive that tuning the Hermite coefficients provides similar benefit as tuning $\lambda$.

3. I also do not see the motivation of minimizing the sum of test error and sensitivity; this no doubt makes the computation more involved, but ultimately we are still designing noisy Gaussian linear models. It would be nice if the authors can outline an example where the deigned RF model achieves smaller loss + sensitivity than directly doing ridge regression on the input. Such result can provide some justification of the use of nonlinear random features.

4. It is unclear how the analysis can benefit practitioners in choosing the activation function, since the optimal Hermite coefficients require knowledge of the target function and SNR, which is not known in practice.

Hu and Lu 2020. Universality laws for high-dimensional learning with random features.
Loureiro et al. 2021. Learning curves of generic features maps for realistic datasets with a teacher-student model.
Montanari and Saeed 2022. Universality of empirical risk minimization.
Ba et al. 2022. High-dimensional asymptotics of feature learning: how one gradient step improves the representation.

**Summary Of The Paper:**

This submission studies the optimal activation function in a two-layer random features model that minimizes a weighted sum of the test error and the model sensitivity measured by a Sobolev norm. The analysis assumes the proportional limit and spherical data; in this setting both the test error and the sensitivity have been analytically derived in prior works. Using these analytic formulae, the authors presented a few cases where the optimal activation can be either linear or nonlinear.

**Summary Of The Review:**

In my opinion this submission requires some major revision to be relevant to the ICLR community. Hence I cannot recommend acceptance.
I am willing to update my evaluation if the authors can address my concerns and elaborate on the motivation / importance of the studied problem.

---

> ### Author Response · Authors · 2022-11-13
> **Response to Weaknesses 1 "Under the setting of [Mei and Montanari 2019], random features model ...."**
>
> It is true that zero MSE can only be achieved for linear target functions under the setting we study. However, our work considers more than linear target functions. It considers non-linear target functions of the type target = linear function + nonlinear gaussian process (a function), which appear in the model and our theorems via the parameter ${F_*}^2$, which is non-zero in general. Different AFs affect the degree to which the non-linear component affects MSE and sensitivity. E.g. even though we can never get the MSE to be smaller than ${F_*}^2$, different AFs will result in a greater, or smaller extra error beyond ${F_*}^2$.  These extra changes are worth investigating in the context, e.g. of the double descent curve. Furthermore, the magnitude of the non-linear component, ${F_*}^2$, plays a crucial role in some of our analyses. In particular, it can affect the optimal AF being linear, or not. See Theorem 9 and its non-trivial dependency on ${F_*}^2$ via the $\rho$ variable.
>
> We should note however that the RFR model can learn non-linear functions in other regimes. For example, when n ~ poly(d), see e.g. [[Misiakiewicz '22](https://arxiv.org/pdf/2204.10425.pdf)], where asymptotic error formulas are also derived. In fact, the asymptotic formulas derived there are almost the same as the ones in [[Mei & Montanari 2022](https://onlinelibrary.wiley.com/doi/pdfdirect/10.1002/cpa.22008)] that we use. In future work, we plan to derive optimal AFs for the RFR model for this setting as well (infinite number of features and n ~ poly(d), d-> inf). The statement that the RFR model cannot learn (with zero MSE) non-linear functions is only valid for the specific classes of target functions for which the different papers advance actual proofs: for the case of  [[Mei & Montanari 2022](https://onlinelibrary.wiley.com/doi/pdfdirect/10.1002/cpa.22008)]  target =linear functions + gaussian processes, and for the case of e.g. [[Ba et al 2022](https://arxiv.org/pdf/2205.01445.pdf)] target = linear functions composed with nonlinearity.  Nothing can be concluded about the inability of the RFR model to learn non-linear functions in general.
>
> The equivalence to the Gaussian covariate model was only proved in [[Mei & Montanari 2022](https://onlinelibrary.wiley.com/doi/pdfdirect/10.1002/cpa.22008)]   for the setting of linear target functions, and not for the more general setting of non-linear target functions.  We are not claiming that it might not hold, but we are not aware of proof. Other work, e.g. [[Hu and Lu 2020](https://arxiv.org/pdf/2009.07669.pdf)] and [[Ba et al 2022](https://arxiv.org/pdf/2205.01445.pdf)], proved the gaussian equivalence for target functions of the single-index type, that is, a linear function composed with a non-linear function, which is different from the set of nonlinear target functions considered by [[Mei & Montanari 2022](https://onlinelibrary.wiley.com/doi/pdfdirect/10.1002/cpa.22008)]. Furthermore,[[Hu and Lu 2020](https://arxiv.org/pdf/2009.07669.pdf)] and [[Ba et al 2022](https://arxiv.org/pdf/2205.01445.pdf)] assume that the AF must be odd, which excludes ReLu, quadratic functions, and even linear functions with non-zero intercept coefficients. In particular, odd activation functions exclude all of our optimal non-linear AFs which are quadratic functions or ReLU-like functions.  We should note that the consequences of going, or not, beyond odd activation functions have been discussed in [[Goldt et al. '22](https://proceedings.mlr.press/v145/goldt22a.html)], section 2.2.  Note also that [[Ba et al 2022](https://arxiv.org/pdf/2205.01445.pdf)]  only provides a gaussian equivalence and an asymptotic risk formula for very small gradient steps, which results in a model that also cannot beat the best linear estimator of the input. They provide no such equivalence or asymptotic formulas in the case when the gradient step is “large” and the resulting model can have an error smaller than the magnitude of the non-linear component of the target function. In short, since the gaussian equivalence has not been rigorously proved (to the best of our knowledge) for the whole family of functions that we are considering, we cannot rigorously state that, in the asymptotic proportional regime, the statements in  [[Ba et al 2022](https://arxiv.org/pdf/2205.01445.pdf)] can be converted to statements for [[Mei & Montanari 2022](https://onlinelibrary.wiley.com/doi/pdfdirect/10.1002/cpa.22008)]. Even if this equivalence would hold, it would not change the fact that it does not solve the problem of finding optimal AFs in these settings, which is our goal and which we are the first to investigate. We recommend the reviewer 1hhM to read the comments we made to reviewer riDW regarding gaussian equivalence and a few other recent papers.

---

> ### Author Response · Authors · 2022-11-13
> **(continued) Response to Weaknesses 1 "Under the setting of [Mei and Montanari 2019], random features model ...."**
>
> Assuming gaussian equivalence, finding the optimal activation function parameters requires tuning two parameters, $\mu_1$ and $\mu_*$ (we can assume $\mu_0 = 0$ w.l.o.g.), while tuning the magnitude of the Gaussian noise added to the features involves only one of such parameter, $\mu_*$. Therefore, strictly speaking, it is not true that “designing the activation function is equivalent to tuning the magnitude of a Gaussian” or that it is effectively the same as regularization. What is true assuming gaussian equivalence is that tuning an AF is the same as tuning parameters in a gaussian model, some of which are noise-related, and others which are not. This has been also correctly pointed out by reviewer riDW that writes that “However since $\mu_*$ also appears in the effective model mismatch noise this leads to the richer behavior observed by the authors in Fig 1 (C) & (D).” This richer behavior is very interesting, worth reporting, and constitutes a significant contribution.
>
> Regardless of the learning ability of the RFR in either our setting (in which it cannot get an error smaller than $F_*$) or in some other setting, in which it can learn non-linear functions, it remains a fact that the RFR is a powerful model to study phenomena like the double descent curve that neither [[Ba et al 2022](https://arxiv.org/pdf/2205.01445.pdf)] nor [[Hu and Lu 2020](https://arxiv.org/pdf/2009.07669.pdf)] study. In particular, this fact clearly disproves the notion that our setup is less interesting than simply studying a linear regression. Linear regression can also exhibit a double descent curve in an asymptotic proportional regime, see e.g. [[Tibshirani et al 2020](https://arxiv.org/pdf/1903.08560.pdf)]. However, for linear regression, and for the overparameterized regime, the double descent curve exhibits a minimizer at a finite ratio of $d/n$, while empirical evidence for real networks shows that the error decreases monotonically as $d/n$ increases. Therefore, linear regression is not a good model for explaining observed double-descent phenomena.  Our work allows us to understand how this and other phenomena are affected by the choice of AF, which is both novel and important. This cannot be achieved starting from a linear regression.
>
> In short, although the existence of the gaussian equivalence property, even setting aside the fact that it does not apply to the full generality of the model we use, allows one to relate tuning AFs to tuning parameters in a gaussian model, it does not allow one to make any precise general mathematical statements about optimal AFs. It does not help in any way to bypass the herculean effort that we have put into the second part (Section 4.2) of our contribution nor does it reduce its merit. We note that the reviewer seems to have glanced at the first part of our contribution (Section 4.1), finding the functional form of activation functions, also a non-trivial and interesting problem.

---

> ### Author Response · Authors · 2022-11-13
> **Response to weaknesses 2 "Given the Gaussian equivalence property,...."**
>
> We agree that *some* of our results, e.g. the existence of optimal linear AF and the fact that tuning an AF is related to (but not the same as) tuning $\lambda$, can be intuitively expected. However:
>
> 1. These few intuitions do not lead to any precise mathematical statements and hence do not reduce the magnitude of our contribution. Someone needs to prove things carefully, this should be rewarded, these people are us.
>
> 2. These few intuitions do not remove the surprise of the ability to actually prove even the intuitive statements explicitly
>
> 3. The degree to which a result is intuitive is a subjective matter and is not a reason to reject a paper. Furthermore, several of our results are not intuitive. For example:
> We tell you “There is a model that exhibits a double descent curve for parameter x with an interpolation threshold at x=y for which the test error is maximal”. We also tell you that “We know that for part of the range of x the optimal AF is linear and for part of the range of x the optimal AF is non-linear”. Then we ask you “What is your intuition about where in the parameter space the optimal AF changes from linear to non-linear ?” The answer to this question is not intuitive. Perhaps an intuitive answer would be  “the switch happens at x=y”. But this is not the case. See Fig. 1-(b).
> We tell you “Optimally tuning lambda is *sometimes* the same as optimally tuning the AF”. Then we ask you “What is your intuition about the generality with which the previous statement holds?” We argue that the answer to this question is also not intuitive. As we can see from Fig. 1-(c), or Fig. 1-(d), sometimes tuning lambda never achieves as good a result as tuning the AF, other times it does.

---

> ### Author Response · Authors · 2022-11-13
> **Response to weaknesses 3 "I also do not see the motivation of minimizing the sum of test error and sensitivity..."**
>
> Thanks for the suggestion. We are currently working on a numerical example for this, and other numerical examples suggested by reviewer riDW, which we hope to include in the final version of our paper.

---

> ### Author Response · Authors · 2022-11-13
> **Response to weaknesses 4 "It is unclear how the analysis can benefit practitioners..."**
>
> This comment seems incompatible with the reviewer's understanding that our paper is theoretical and the reviewer's acceptance that the “Empirical Novelty And Significance” is “Not applicable” to our paper. “Not applicable” is not the same thing as “Poor/bad”. We ask the reviewer to reconsider this comment and any of its implications for our paper’s score.

---

> ### Author Response · Authors · 2022-11-13
> **Response to Clarity, Quality, Novelty, Reproducibility and Summary of the reviewers**
>
> *This submission requires some major revision*
>
> We will include all of the points discussed below in our revision and cite all provided, and other references. We will also include in our revision the comments resulting from conversing with reviewer riDW below. We disagree that the task of including such observations constitutes a major revision of the paper. It is a rather small one in our opinion. Essentially moving some of the formulas in Theorem 9 into the appendix, and copying a polished version of this discussion in OpenReview to the related work section.
>
> *The main technical novelty is to solve for the optimal coefficients in the activation function, which is a rather tedious computation.*
>
> The reviewer seems to miss the fact that half of our work is devoted to Section 4.1, the derivation of the functional form of optimal AF. This, to the best of our knowledge, is also new and non-trivial. Although the resulting AF are simple functions, and their derivation is based on an assumption that could be arguably different (we agree with reviewer 7nWY that the need for these assumptions is not a bit issue), and someone else could also, perhaps, have done our derivations, or similar ones, the fact remains that they did not. We did. This should be recognized.

---

> ### Author Response · Authors · 2022-11-13
> **Question from authors to reviewer 1hhM**
>
> Finally, we appreciate the reviewer stating that “I am willing to update my evaluation if the authors can address my concerns and elaborate on the motivation/importance of the studied problem”.
>
> Question: does the reviewer agree that our comments below, if included in the paper, address the fact that “intuition of some predictions” $\leq$ “intuition on *all* predictions” $\leq$ “rigorously prove all predictions” = “our contribution”? If the inclusion of the points below does not satisfy the reviewer, can the reviewer explain why and perhaps give attainable actionable points that we can focus on? If the inclusion of the points below does satisfy the reviewer, can the reviewer update the paper’s score?
>
> Thanks

---

> ### Author Response · Authors · 2022-11-14
> **Thank you**
>
> We thank you for your time in reading our paper and providing us with feedback. We recommend reading the feedback below as well as the feedback to other reviewers. Please read our comments from the bottom up.

---

> ### Comment · Reviewer_1hhM · 2022-11-16
> **Reply to authors**
>
> I have read the authors' reply, which unfortunately does not address my concerns. Due to the limited time, I will only comment on some important points from the rebuttal.
>
> 1. *"the Gaussian equivalence has not been rigorously proved for the whole family of functions that we are considering."*
> This is not true. If the authors simply derive the test error of the Gaussian equivalent model under their linear + isotropic Gaussian process target function (which is straightforward to do), it will be transparent that the formula is the same as the RF case given by the partial Stieltjes transforms. The same set of equations also appeared in [Gerace et al 2020], [Hu and Lu 2020], [Loureiro et al 2021], [Ba et al 2022]; note that in all these results, the nonlinear component of the target cannot be learned in the proportional regime.
>
> 2. The authors failed to convince me that their analysis is "both novel and important" (quoted from the rebuttal). Indeed the judgment of what counts as important/novel/intuitive can be somewhat subjective, but given the Gaussian equivalent interpretation, I still believe the results here have limited theoretical and practical impact. The fact that no one has done the same computation rigorously does not entail that the studied problem is well-motivated.

---

> > ### Author Response · Authors · 2022-11-16
> > **Response to "can be somewhat subjective," and "fact that no one has done the same computation rigorously does not entail that"**
> >
> > We agree that the opinion of what is important is subjective.
> > However, we disagree that personal subjective preferences should be the main reason for rejecting a paper.
> > Perhaps a secondary reason, but not a main reason.
> >
> > We also disagree that the issue of novelty is subjective. We feel pretty safe in claiming no one has done what we are doing before. If we are wrong, please do provide us with a reference.
> >
> > We agree that "The fact that no one has done the same computation rigorously does not entail that the studied problem is well-motivated.". However, it is indisputable that there is a widespread interest in understanding the effect of AFs on learning with NNs (See Related Work). It is also indisputable that nothing as been done as rigorously on this end apart from our advances. It is also indisputable, as we expanded in our reply, and to which reviewer 1hhM has not replied yet, that the analogies/intuitions that can be extracted from the gaussian equivalence are very limited. Therefore, several of our results are objectively non-intuitive. For example, in Fig. c and d reviewer 1hhM will see that tuning AF and tuning lambda can lead to very different results.
> >
> > We recommend reviewer 1hhM to see our reply to reviewer riDW where we acknowledge that, at least in future work, someone should investigates how much of our conclusions extrapolate to other settings. As reviewer 1hhM understands, time is limited, so we might not be able to produce these investigations before the 18th of November. We do not think that just because one group does a first investigation and another group a further investigation, that neither is worth publishing.

---

> > ### Author Response · Authors · 2022-11-16
> > **Response to "This is not true. "**
> >
> > The review is 100% right that one can easily show that there is some gaussian model with the same equations as those in [Mei & Montanari 22], which by the way is not the point of our paper but its starting point.
> >
> > We were not very clear (our fault) and now clarify that, in the context of our reply, what we were trying to mean by "has not been rigorously proved for the whole family of functions" was something slightly different.
> >
> > Assume that:
> >
> > A) For model 1, with target functions in family C1, one computes an asymptotic test error formula.
> >
> > B) For model 2, with target functions in family C2, one computes an asymptotic test error formula.
> >
> > C) One proves that there is one Guassian covariate model that leads to the same test error formulas as in A) and B), which hence agree.
> >
> > We were overloading the term model 1 and model 2 are connected via the same "Gaussian equivalence" to mean that, **if C1 = C2**, then A, B and C hold.
> >
> > However, if C1 and C2 are different, then, despite the fact that both models can be explained via the same formulas, which can also be explained via a Gaussian model, there are setups for models 1 and model 2 that have not been proved to behave the same way, namely, if we choose a target function in C1 that is not in C2. We would thus not use the term "model 1 and model 2 are connected via the same Gaussian equivalence".
> >
> > This is why we were trying to point to the fact that the papers cited consider setups, and their derivations are only valid within these setups, and in particular target functions, that **do not** cover in full generality the set of functions considered in [Mei & Montanari 22].
> >
> > E.g.  [Hu and Lu 2020] and [Ba et al 2022] require odd activation functions and bounded derivatives. [Mei & Montanari 22] does not.
> >
> > E.g. [Gerace et al 2020] have no conditions on the activation function, which obviously is not possible if a rigorous proof was being provided. Indeed , they use replica theory.
> >
> > E.g. [Loureiro et al 2021] consider a training procedure where the is no activation function. The activation function can be considered absorbed in g, the cost function, but since g must be convex, their asymptotic formulas are not derived for the full family of functions considered by [Mei & Montanari 22].
> >
> > Just because two models 1 and 2 have been proved to lead to the same formulas for a few common target functions, and these formulas match the formulas of a third simpler model 3 (gaussian), it does not mean that the models 1 and 2 are the same for all target functions.
> >
> > Also, the similarity of model 3 and model 1 for target functions in C1, and the similarity of model 3 and model 1 for target functions in C2 is a **consequence**, a **product**, of the analysis. Model 3, by itself, without connection to model 1 and model 2 is not so meaningful because if lacks more explicit realistic features. For example, the gaussian model does not have any AFs. It has constants that can model AFs in other models, but we need the other models for we to be able to make this interpretation. Perhaps not everyone appreciates these ontological differences.
> >
> > '*Note that in all these results, the nonlinear component of the target cannot be learned in the proportional regime.*'
> >
> > We agree, we are aware of this, and we have also argued in our initial reply why this does not detract from the usefulness in studying our problem. See below
> >
> > ''**Regardless of the learning ability of the RFR in either our setting (in which it cannot get an error smaller than $F_*$) or in some other setting, in which it can learn non-linear functions, it remains a fact that the RFR is a powerful model to study phenomena like the double descent curve that neither [[Ba et al 2022](https://arxiv.org/pdf/2205.01445.pdf)] nor [[Hu and Lu 2020](https://arxiv.org/pdf/2009.07669.pdf)] study. In particular, this fact clearly disproves the notion that our setup is less interesting than simply studying a linear regression. Linear regression can also exhibit a double descent curve in an asymptotic proportional regime, see e.g. [[Tibshirani et al 2020](https://arxiv.org/pdf/1903.08560.pdf)]. However, for linear regression, and for the overparameterized regime, the double descent curve exhibits a minimizer at a finite ratio of $n/d$, while empirical evidence for real networks shows that the error decreases monotonically as $n/d$ increases. Therefore, linear regression is not a good model for explaining observed double-descent phenomena.  Our work allows us to understand how this and other phenomena are affected by the choice of AF, which is both novel and important. This cannot be achieved starting from a linear regression.**''

---

> > ### Author Response · Authors · 2022-11-16
> > **Thank you**
> >
> > We understand the reviewer is extremely busy and we really appreciate the time taken to clarify our points.

---

### Author Response · Authors · 2022-11-18
**Dear reviewers, a great "thank you" for your active discussion and your help to improve our paper !**

We have incorporated into our revised version the points discussed in Openreview.

The main changes are as follows.

0) We have fixed typos, made small notation changes (e.g. O -> L) and made suggested moves of content to the appendices.

1) We now reference Gaussian equivalent models in our paper (Section 2.3)

1.1) We explain their partial connection with implicit regularization

1.2) We emphasize the difference between tuning AFs and tuning regularization

1.2.1) We have updated the caption of Fig 1 to highlight this difference when talking about Fig1-C and Fig1-D

2) We explain the importance and limitations of studying the RFR model (Section 2.4)

2.1) We emphasize why linear regression is not as good of a model to study, even when it surpasses the RFR model's performance for certain classes of non-linear functions in certain regimes

3) We have extended the Future Work section (Section 5)

3.1) We highlight the importance of testing how general our conclusions hold

3.1.1) We mention several works for which an analysis similar to the one in this paper can be applied

3.1.2) We mention several numerical experiments that would be good to perform in the near future

3.2) We state some of our expectations about these generalizations

We remain very much open to suggestion on how to improve the paper. However, we do hope that the reviewers agree that it is impossible to include in a single conference paper (already a very rich paper) all connections with all possible topics, perform all experiments, analyze all models, etc.

We are actively working on producing several numerical examples discussed in Openreview. We did not have time to include them in our revised version during the time window in which OpenReview allows authors to upload new versions. We hope to include them in the appendix of our final version, if the paper is accepted. Any experiment that is currently mentioned as "Future Work" that we are able to complete in the next weeks, and include in final version, will be removed from the "Future Work" in the final version.

We hope that reviewers are satisfied with our reposes in Openreview and our changes to the paper, and that they are willing to raise our paper's score.

---

> ### Comment · Reviewer_riDW · 2022-11-27
> **Post-discussion update**
>
> Dear authors,
>
> Thank you for the revised version, which I believe addresses most of the points in our discussion and improve several aspects of the manuscript. For this reason, I am raising my score to a 6. I look forward to seeing a final version with the numerical experiments mentioned above.

---

### Decision · Program_Chairs · 2023-01-20

**Decision:**

Accept: poster

**Justification For Why Not Higher Score:**



**Justification For Why Not Lower Score:**



**Metareview: Summary, Strengths And Weaknesses:**

This submission is centered around the random feature (RF) technique in regression. More particularly, the author consider a specific two-layer neural network (1) with weights initialized randomly in the 1st layer, and optimized in the 2nd layer using a ridge regression objective (2). The authors' primary goal is to identify the optimal activation function (\sigma in (1), see (8)) where the goodness of the activation function is defined as the convex combination of the mean squared error and the sensitivity w.r.t. the input arguments (5)-(7) while keeping the activation function simple in the sense of (4). The work operates in the asymptotic proportional regime (Assumption 2), with focus on 3 regimes (last paragraph of Section 2): (i) ridgeless limit, (ii) highly overparameterized limit, and (iii) large sample limit. The main results of the paper are the sensitivity values (7) in the considered regimes (Theorem 4-6), alongside with the optimal activation functions while keeping the mean parameters \mu_0, \mu_1 and \mu_2 specified in (9) fixed (Theorem 7-8 for the different simplicity measures on \sigma), and the determination of the optimal mean parameters in the 3 regimes (Theorem 9-11).

Random features are without doubt among the most popular methods in machine learning; the focus and motivation of the submission are excellent fit to ICLR. The submission is well-organized and clearly written, the authors deliver a nice story underpinned by the proved solid theoretical results as it was evaluated by the reviewers. The work also inspired lively discussion: the raised questions were successfully resolved by the authors' rebuttal, and the discussion shed light on various potential future research directions and extensions.

My personal recommendation is to tone down the discussion on the future work in the paper (Section 4) slightly; it is more typical to grant applications.

**Note From Pc:**

if the above contains the word "oral" or "spotlight" please see: "oral" presentation means -> notable-top-5% and "spotlight" means -> notable-top-25%. As stated in our emails, we are disassociating presentation type from AC recommendations